# Dimension-free deterministic equivalents and scaling laws for random feature regression

**Leonardo Defilippis**
Département d'Informatique
École Normale Supérieure - PSL & CNRS
`leonardo.defilippis@ens.psl.eu`

**Bruno Loureiro**
Département d'Informatique
École Normale Supérieure - PSL & CNRS
`bruno.loureiro@ens.psl.eu`

**Theodor Misiakiewicz**
Department of Statistics and Data Science
Yale University
`theodor.misiakiewicz@yale.edu`

## Abstract

In this work we investigate the generalization performance of random feature ridge regression (RFRR). Our main contribution is a general deterministic equivalent for the test error of RFRR. Specifically, under a certain concentration property, we show that the test error is well approximated by a closed-form expression that only depends on the feature map eigenvalues. Notably, our approximation guarantee is non-asymptotic, multiplicative, and independent of the feature map dimension—allowing for infinite-dimensional features. We expect this deterministic equivalent to hold broadly beyond our theoretical analysis, and we empirically validate its predictions on various real and synthetic datasets. As an application, we derive sharp excess error rates under standard power-law assumptions of the spectrum and target decay. In particular, we provide a tight result for the smallest number of features achieving optimal minimax error rate.

## 1 Introduction

At odds with classical statistical intuition, overparametrized neural networks are able to generalize while perfectly interpolating the training data. This observation, which defies the canonical mathematical understanding of generalization based on complexity measures and uniform convergence, appears surprising at first [Zhang et al., 2017]. However, recent progress in our mathematical understanding of generalization has taught us that this *benign overfitting* property of overparametrized neural networks is shared by a plethora of simpler learning tasks [Bartlett et al., 2021, Belkin, 2021]. Among them, the investigation of the following class of *random feature models* has been at the forefront of this progress:

$$\mathcal{F}_{\mathrm{RF}} = \left\{ \hat{f}(\boldsymbol{x}; \boldsymbol{a}) = \frac{1}{\sqrt{p}} \sum_{j \in [p]} a_j \varphi(\boldsymbol{x}, \boldsymbol{w}_j) \ : \ \boldsymbol{a} = (a_j)_{j \in [p]} \in \mathbb{R}^p \right\}. \tag{1}$$

Here $\boldsymbol{x} \in \mathcal{X}$ denotes the inputs and $\boldsymbol{W} = (\boldsymbol{w}_j)_{j \in [p]}$ a set of weight vectors which are taken to be random $\boldsymbol{w}_j \in \mathcal{W} \subseteq \mathbb{R}^d \sim_{\mathrm{iid}} \mu_w$. Hence, as the name suggests the feature map $\varphi : \mathcal{X} \times \mathcal{W} \to \mathbb{R}$ defines a random function. A few examples of random feature maps include the fully connected neural network features $\varphi(\boldsymbol{x}, \boldsymbol{w}) = \sigma(\langle \boldsymbol{w}, \boldsymbol{x} \rangle)$, $\sigma : \mathbb{R} \to \mathbb{R}$, and the convolutional features with global average pooling $\varphi(\boldsymbol{x}, \boldsymbol{w}) = 1/d \sum_{\ell=1}^{d} \sigma(\langle \boldsymbol{w}, g_\ell \cdot \boldsymbol{x} \rangle)$ where $\sigma : \mathbb{R} \to \mathbb{R}$ and

38th Conference on Neural Information Processing Systems (NeurIPS 2024).

$g_\ell \cdot \boldsymbol{x} = (x_{\ell+1}, \ldots, x_d, x_1, \ldots, x_\ell)$ is the $\ell$-shift operator with cyclic boundary conditions (both with $\mathcal{X} = \mathbb{R}^d$).

Random features [Balcan et al., 2006, Rahimi and Recht, 2007] were originally introduced as a computationally efficient approximation for the limiting kernel:

$$K(\boldsymbol{x}, \boldsymbol{x}') = \mathbb{E}_{\boldsymbol{w} \sim \mu_w} [\varphi(\boldsymbol{x}, \boldsymbol{w}) \varphi(\boldsymbol{x}', \boldsymbol{w})]. \tag{2}$$

Although this can reduce the computational cost of kernel methods, it introduces an approximation error. Rahimi and Recht [2008] showed that in a supervised setting with $n$ samples, $p = O(n)$ features are sufficient to achieve an excess risk $O(n^{-1/2})$. Rudi and Rosasco [2017] improved this result under standard power-law assumptions on the asymptotic kernel spectrum, showing that for fast decays, less features are needed to achieve the minimax rate. In Section 4 we will revisit this question, where we will derive a tight result for the minimum number of features.

More recently, the random feature model has gained in popularity as a proxy model for studying the generalization properties of two-layer neural networks in the lazy regime of training [Jacot et al., 2018, Chizat et al., 2019]. Indeed, for particular choices of feature maps such as $\varphi(\boldsymbol{x}, \boldsymbol{w}) = \sigma(\langle \boldsymbol{w}, \boldsymbol{x} \rangle)$, it can also be seen as a two-layer neural network with fixed first-layer weights. Exact asymptotic results for the generalization error of eq. (1) were derived for different supervised learning tasks under the proportional scaling regime $n, p = \Theta(d)$ in [Mei and Montanari, 2022, Gerace et al., 2021, Dhifallah and Lu, 2020, Hu and Lu, 2023, Goldt et al., 2022, Loureiro et al., 2022, Bosch et al., 2023b,a, Schröder et al., 2023, 2024] and under more general polynomial scaling $n, p = \Theta(d^\kappa)$ in [Simon et al., 2023b, Aguirre-López et al., 2024, Hu et al., 2024]. As discussed above, these works played a fundamental role in our current mathematical understanding of the relationship between overparametrization and generalization, demystifying different phenomena such as double descent Belkin et al. [2019] and benign overfitting Bartlett et al. [2020]. It also led to fundamental separation results between lazy and trained networks [Ghorbani et al., 2019, 2020, Mei et al., 2022], recently motivating the investigation of corrections to the random limit [Ba et al., 2022, Dandi et al., 2023, Moniri et al., 2024, Cui et al., 2024].

With the exception of [Simon et al., 2023b], which is based on non-rigorous arguments, the results in all the works cited above are derived in the asymptotic limit of large data dimension. However, the relative scaling of the $n, p, d$ is fundamentally artificial, and in practice it is hard to unambiguously define the regime of interest. Our main goal in this manuscript is to provide a dimension-free characterization of the generalization error allowing us to give tight answers to questions which cannot be addressed asymptotically. More precisely, our main contributions are:

(i) Under a concentration assumption on the feature map eigenfunctions, we prove a non-asymptotic deterministic approximation for the RFRR risk $\mathcal{R}_{\text{test}} \approx \mathsf{R}_{n,p}$ which is independent of the feature map dimension. More precisely, with high-probability over the input data and random weights:

$$|\mathcal{R}_{\text{test}} - \mathsf{R}_{n,p}| \leq \tilde{O}(p^{-1/2} + n^{-1/2}) \cdot \mathsf{R}_{n,p}. \tag{3}$$

where the *deterministic equivalent* $\mathsf{R}_{n,p}$ can be computed by solving a set of self-consistent equations of the type $x = f(x)$, with $f$ a contractive map. This result unifies the long list of asymptotic formulas in the RFRR literature, and proves a conjecture by Simon et al. [2023b]. We numerically validate the results on various real and synthetic datasets. The precise statement of the theorem and the assumptions are discussed in Section 3.

(ii) Leveraging our formula, we investigate the error scaling laws in a setting where the target function and feature spectrum decay as a power-law, also known as *source and capacity* conditions. We provide a full picture of the different scaling regimes and the cross-overs between them, summarized in Figure 2. Our result is closely related to the neural scaling laws literature [Kaplan et al., 2020], and provides the first rigorous, non-linear extension of [Bahri et al., 2024, Maloney et al., 2022].

(iii) We provide a sharp expression for the minimum number of features required to achieve the minimax optimal decay rate of Caponnetto and De Vito [2007], closing the gap of previous lower-bounds in the literature [Rudi and Rosasco, 2017].

**Further related works —** Deterministic equivalents have been derived for a wide range of learning problems, such as ridge regression [Dobriban and Wager, 2018, Hastie et al., 2022, Cheng and Montanari, 2022, Wei et al., 2022], kernel regression [Misiakiewicz and Saeed, 2024], shallow [Liao

and Couillet, 2018, Mei and Montanari, 2022, Chouard, 2022, Bach, 2024, Atanasov et al., 2024] and deep random feature regression [Fan and Wang, 2020, Schröder et al., 2023, 2024, Chouard, 2023] and spiked random features [Wang et al., 2024]. Scaling laws under source and capacity conditions were studied by several authors in the context of kernel ridge regression [Bordelon et al., 2020, Spigler et al., 2020, Cui et al., 2022, Simon et al., 2023a, Li et al., 2023, Misiakiewicz and Mei, 2022, Favero et al., 2021, Cagnetta et al., 2023, Dohmatob et al., 2024] and classification [Cui et al., 2023].

## 2 Setting

In this work, we focus on the generalization properties of the random feature class $\mathcal{F}_{\text{RF}}$ defined in eq. (1) in a supervised regression setting. More precisely, consider a data set $\mathcal{D} = \{(\boldsymbol{x}_i, y_i)_{i \in [n]}\}$ composed of $n$ independent and identically distributed samples from a joint distribution $\mu_{x,y}$ on $\mathcal{X} \times \mathbb{R}$. Let $f_\star(\boldsymbol{x}) = \mathbb{E}[y|\boldsymbol{x}]$ denote the target function. We assume $f_\star \in L_2(\mu_x)$, where $\mu_x$ is the marginal distribution over $\mathcal{X}$. Moreover, we assume the noise $\varepsilon := y - f_\star(\boldsymbol{x})$ has zero mean and finite variance $\mathbb{E}[\varepsilon^2] = \sigma_\varepsilon^2 < \infty$. Note this is equivalent to:

$$y_i = f_\star(\boldsymbol{x}_i) + \varepsilon_i, \qquad f_\star \in L_2(\mu_x). \tag{4}$$

Given the training data, we are interested in the properties of the minimiser:

$$\hat{\boldsymbol{a}}_\lambda(\boldsymbol{Z}, \boldsymbol{y}) := \underset{\boldsymbol{a} \in \mathbb{R}^p}{\arg\min} \left\{ \sum_{i \in [n]} \left( y_i - \hat{f}(\boldsymbol{x}_i; \boldsymbol{a}) \right)^2 + \lambda \|\boldsymbol{a}\|_2^2 \right\} = (\boldsymbol{Z}^\top \boldsymbol{Z} + \lambda \boldsymbol{I}_p)^{-1} \boldsymbol{Z}^\top \boldsymbol{y}, \tag{5}$$

where we have defined the feature matrix $\boldsymbol{Z}_{ij} = p^{-1/2} \varphi(\boldsymbol{x}_i; \boldsymbol{w}_j)$ and the label vectors $\boldsymbol{y} = (y_i)_{i \in [n]}$. In particular, we are interested in its capacity of generalising to unseen data, as quantified by the *excess population risk*:

$$\mathcal{R}(f_\star, \boldsymbol{X}, \boldsymbol{W}, \boldsymbol{\varepsilon}, \lambda) := \mathbb{E}_{\boldsymbol{x} \sim \mu_x} \left[ \left( f_\star(\boldsymbol{x}) - \hat{f}(\boldsymbol{x}; \hat{\boldsymbol{a}}_\lambda) \right)^2 \right]. \tag{6}$$

It will be convenient to decompose the excess risk above in terms of the standard bias and variance:

$$\mathcal{R}(f_\star; \boldsymbol{X}, \boldsymbol{W}, \lambda) := \mathbb{E}_{\boldsymbol{\varepsilon}} \left[ \mathcal{R}(f_\star; \boldsymbol{X}, \boldsymbol{W}, \boldsymbol{\varepsilon}, \lambda) \right] = \mathcal{B}(f_\star; \boldsymbol{X}, \boldsymbol{W}, \lambda) + \mathcal{V}(\boldsymbol{X}, \boldsymbol{W}, \lambda), \tag{7}$$

where:

$$\mathcal{B}(f_\star; \boldsymbol{X}, \boldsymbol{W}, \lambda) := \mathbb{E}_{\boldsymbol{x} \sim \mu_x} \left[ \left( f_\star(\boldsymbol{x}) - \mathbb{E}_{\boldsymbol{\varepsilon}}[\hat{f}(\boldsymbol{x}; \hat{\boldsymbol{a}}_\lambda)] \right)^2 \right], \tag{8}$$

$$\mathcal{V}(\boldsymbol{X}, \boldsymbol{W}, \lambda) := \mathbb{E}_{\boldsymbol{x} \sim \mu_x} \left[ \text{Var}_{\boldsymbol{\varepsilon}}(\hat{f}(\boldsymbol{x}; \hat{\boldsymbol{a}}_\lambda)) \right]. \tag{9}$$

Note that to simplify the exposition, we have explicitly taken an expectation over the training data noise $\boldsymbol{\varepsilon} = (\varepsilon_i)_{i \in [n]}$. Indeed, it can be shown that the excess risk eq. (6) concentrates on its expectation over $\varepsilon$ under mild assumptions (see for example Misiakiewicz and Saeed [2024]).

## 3 Deterministic equivalents

The excess risk eq. (6) is a function of the covariates $\boldsymbol{X}$ and the weights $\boldsymbol{W}$, and therefore it is a random quantity. Our main result in what follows is a sharp characterization of the bias and variance in terms of a *deterministic equivalent* depending only on the model parameters and spectral properties of the features. Consider a square-integrable $\varphi \in L_2(\mathcal{X} \times \mathcal{W})$, and define the Fredholm integral operator $\mathbb{T} : L_2(\mathcal{X}) \to \mathcal{V} \subseteq L_2(\mathcal{W})$:

$$\mathbb{T}h(\boldsymbol{w}) := \int_{\mathcal{X}} \varphi(\boldsymbol{x}; \boldsymbol{w}) h(\boldsymbol{x}) \mu_x(\mathrm{d}\boldsymbol{x}), \qquad \forall h \in L_2(\mathcal{X}), \tag{10}$$

where we define $\mathcal{V} = \text{Im}(\mathbb{T})$. This is a compact operator, and therefore can be diagonalized:

$$\mathbb{T} = \sum_{k=1}^\infty \xi_k \psi_k \phi_k^*, \tag{11}$$

where $(\xi_k)_{k\geq 1} \subseteq \mathbb{R}$ are the eigenvalues and $(\psi_k)_{k\geq 1}$ and $(\phi_k)_{k\geq 1}$ are orthonormal bases of $L_2(\mathcal{X})$ and $\mathcal{V}$ respectively:

$$\langle \psi_k, \psi_{k'}\rangle_{L_2(\mathcal{X})} = \delta_{kk'}, \qquad \langle \phi_k, \phi_{k'}\rangle_{L_2(\mathcal{W})} = \delta_{kk'}. \tag{12}$$

Without loss of generality, we assume the eigenvalues are ordered in non-increasing absolute values $|\xi_1| \geq |\xi_2| \geq \ldots$, and for simplicity of presentation we assume that all eigenvalues are non-zero, i.e., $\ker(\mathbb{T}) = \{0\}$. Denote $\boldsymbol{\Sigma} = \mathrm{diag}(\xi_1^2, \xi_2^2, \ldots) \in \mathbb{R}^{\infty \times \infty}$ the diagonal matrix of the squared eigenvalues. Similarly, since $f_\star \in L_2(\mu_x)$, it admits the following decomposition in $(\psi)_{k\geq 1}$:

$$f_\star = \sum_{k\geq 1} \boldsymbol{\beta}_{\star,k} \psi_k \tag{13}$$

Our formal results will assume the following concentration property over the eigenfunctions.

**Assumption 3.1** (Concentration of the eigenfunctions). *Denote the (infinite-dimensional) random vectors[1] $\boldsymbol{\psi} := (\xi_k \psi_k(\boldsymbol{x}))_{k\geq 1}$ and $\boldsymbol{\phi} := (\xi_k \phi_k(\boldsymbol{w}))_{k\geq 1}$. There exists a constant $\mathsf{C}_x > 0$ such that for any deterministic p.s.d. matrix $\boldsymbol{A} \in \mathbb{R}^{\infty \times \infty}$, i.e. a linear operator acting on an infinite-dimensional Hilbert space, with $\mathrm{Tr}(\boldsymbol{\Sigma A}) < \infty$, we have*

$$\mathbb{P}\left(\left|\boldsymbol{\psi}^\mathsf{T} \boldsymbol{A} \boldsymbol{\psi} - \mathrm{Tr}(\boldsymbol{\Sigma A})\right| \geq t \cdot \|\boldsymbol{\Sigma}^{1/2} \boldsymbol{A} \boldsymbol{\Sigma}^{1/2}\|_F\right) \leq \mathsf{C}_x \exp\left\{-t/\mathsf{C}_x\right\}, \tag{14}$$

$$\mathbb{P}\left(\left|\boldsymbol{\phi}^\mathsf{T} \boldsymbol{A} \boldsymbol{\phi} - \mathrm{Tr}(\boldsymbol{\Sigma A})\right| \geq t \cdot \|\boldsymbol{\Sigma}^{1/2} \boldsymbol{A} \boldsymbol{\Sigma}^{1/2}\|_F\right) \leq \mathsf{C}_x \exp\left\{-t/\mathsf{C}_x\right\}. \tag{15}$$

While Assumption 3.1 is restrictive and will not be satisfied by many non-linear settings, it covers a number of popular theoretical models studied in the literature: 1) independent sub-Gaussian entries, 2) verifying a log-Sobolev inequality or convex Lipschitz concentration (see Cheng and Montanari [2022]). We expect that Assumption 3.1 can be relaxed using the same procedure as in Misiakiewicz and Saeed [2024] to cover classical examples such as data and weights uniformly distributed on the sphere or hypercube. Such a relaxation is involved and we leave it to future work. We will further assume that:

**Assumption 3.2.** *There exists $\mathsf{m} \in \mathbb{N}$ such that*

$$p\xi_{\mathsf{m}+1}^2 \leq \frac{\lambda}{n} \sum_{k=\mathsf{m}+1}^{\infty} \xi_k^2. \tag{16}$$

*Furthermore, we will assume that for some $\mathsf{C}_* > 0$ that we have*

$$\frac{\mathrm{Tr}(\boldsymbol{\Sigma}(\boldsymbol{\Sigma} + \nu_2)^{-1})}{\mathrm{Tr}(\boldsymbol{\Sigma}^2(\boldsymbol{\Sigma} + \nu_2)^{-2})} \leq \mathsf{C}_*, \qquad \frac{\langle \boldsymbol{\beta}_\star, (\boldsymbol{\Sigma} + \nu_2)^{-1} \boldsymbol{\beta}_\star\rangle}{\nu_2 \langle \boldsymbol{\beta}_\star, (\boldsymbol{\Sigma} + \nu_2)^{-2} \boldsymbol{\beta}_\star\rangle} \leq \mathsf{C}_*. \tag{17}$$

Assumption 3.2 is technical, and we believe it can be removed at the cost of a more involved analysis. For instance, eq. (16) is always satisfied for $\xi_k^2 \propto k^{-\alpha}$ if we take $\mathsf{m} = O(p^2)$. Condition (17) was also considered in Cheng and Montanari [2022], and is satisfied in many settings of interest, for example under source and capacity conditions $\beta_k \asymp k^{-\beta}$ and $\xi_k^2 \asymp k^{-\alpha}$ considered in Section 4.

**Main result —** Our main result concerns a dimension-free characterization of the risk eq. (6) in terms of deterministic equivalents. We start by defining them.

**Definition 1** (Deterministic equivalents). *Given integers $n, p$, covariance matrix $\boldsymbol{\Sigma}$ and regularization parameter $\lambda \geq 0$. Consider the parameter $\nu_2 \in \mathbb{R}_{>0}$ defined as the unique solution of the self-consistent equation:*

$$1 + \frac{n}{p} - \sqrt{\left(1 - \frac{n}{p}\right)^2 + 4\frac{\lambda}{p\nu_2}} = \frac{2}{p}\mathrm{Tr}\left(\boldsymbol{\Sigma}(\boldsymbol{\Sigma} + \nu_2)^{-1}\right), \tag{18}$$

*and $\nu_1 \in \mathbb{R}_{>0}$ is given by:*

$$\nu_1 := \frac{\nu_2}{2}\left[1 - \frac{n}{p} + \sqrt{\left(1 - \frac{n}{p}\right)^2 + 4\frac{\lambda}{p\nu_2}}\right]. \tag{19}$$

---

[1]Note that we can consider both $\boldsymbol{\psi}$ and $\boldsymbol{\phi}$ random elements of the Hilbert space $\ell_2$ with distribution induced by $\boldsymbol{x} \sim \mu_x$ and $\boldsymbol{w} \sim \mu_w$, where $\mathbb{E}[\boldsymbol{\psi}\boldsymbol{\psi}^\mathsf{T}] = \mathbb{E}[\boldsymbol{\phi}\boldsymbol{\phi}^\mathsf{T}] = \boldsymbol{\Sigma}$ and $\mathrm{Tr}(\boldsymbol{\Sigma}) < \infty$.

*We introduce the short-hand:*

$$\Upsilon(\nu_1, \nu_2) := \frac{p}{n}\left[\left(1 - \frac{\nu_1}{\nu_2}\right)^2 + \left(\frac{\nu_1}{\nu_2}\right)^2 \frac{\mathrm{Tr}(\mathbf{\Sigma}^2(\mathbf{\Sigma}+\nu_2)^{-2})}{p - \mathrm{Tr}(\mathbf{\Sigma}^2(\mathbf{\Sigma}+\nu_2)^{-2})}\right], \tag{20}$$

$$\chi(\nu_2) := \frac{\mathrm{Tr}(\mathbf{\Sigma}(\mathbf{\Sigma}+\nu_2)^{-2})}{p - \mathrm{Tr}(\mathbf{\Sigma}^2(\mathbf{\Sigma}+\nu_2)^{-2})}. \tag{21}$$

*Then, the deterministic equivalents for the bias, variance and test error are defined as:*

$$\mathsf{B}_{n,p}(\boldsymbol{\beta}_*, \lambda) := \frac{\nu_2^2}{1 - \Upsilon(\nu_1, \nu_2)}\left[\langle\boldsymbol{\beta}_*, (\mathbf{\Sigma}+\nu_2)^{-2}\boldsymbol{\beta}_*\rangle + \chi(\nu_2)\langle\boldsymbol{\beta}_*, \mathbf{\Sigma}(\mathbf{\Sigma}+\nu_2)^{-2}\boldsymbol{\beta}_*\rangle\right], \tag{22}$$

$$\mathsf{V}_{n,p}(\lambda) := \sigma_\varepsilon^2 \frac{\Upsilon(\nu_1, \nu_2)}{1 - \Upsilon(\nu_1, \nu_2)}, \tag{23}$$

$$\mathsf{R}_{n,p}(\boldsymbol{\beta}_*, \lambda) := \mathsf{B}_{n,p}(\boldsymbol{\beta}_*, \lambda) + \mathsf{V}_{n,p}(\lambda). \tag{24}$$

Our main result provides precise conditions for when the deterministic equivalents defined in definition 1 are a good approximation for the test error eq. (6), as a function of the dimensions $n, p$, feature covariance $\mathbf{\Sigma}$, and regularization $\lambda > 0$. More precisely, the approximation rates will depend on them through the following quantities:

$$r_{\mathbf{\Sigma}}(k) := \frac{\sum_{j\geq k}\xi_j^2}{\xi_k^2}, \quad M_{\mathbf{\Sigma}}(k) := 1 + \frac{r_{\mathbf{\Sigma}}(\lfloor\eta_* \cdot k\rfloor) \vee k}{k}\log\left(r_{\mathbf{\Sigma}}(\lfloor\eta_* \cdot k\rfloor) \vee k\right), \tag{25}$$

$$\rho_\kappa(p) := 1 + \frac{p \cdot \xi_{\lfloor\eta_* \cdot p\rfloor}^2}{\kappa}M_{\mathbf{\Sigma}}(p), \tag{26}$$

$$\widetilde{\rho}_\kappa(n, p) := 1 + \mathbb{1}[n \leq p/\eta_*] \cdot \left\{\frac{n\xi_{\lfloor\eta_* \cdot n\rfloor}^2}{\kappa} + \frac{n}{p} \cdot \rho_\kappa(p)\right\}M_{\mathbf{\Sigma}}(n), \tag{27}$$

Below we denote $C_{a_1,\ldots,a_k}$ constants that only depend on the values of $\{a_i\}_{i\in[k]}$. We use $a_i = $ '$*$' to denote the dependency on the constants in Assumptions 3.1 and 3.2.

**Theorem 3.3** (Test error of RFRR). *Under Assumptions 3.1, 3.2 and for any $D, K > 0$, there exist constants $\eta_* \in (0, 1/2)$ and $C_{*,D,K} > 0$ such that the following holds. For any $n, p \geq C_{*,D,K}$, regularization $\lambda > 0$, and target function $f_\star \in L_2(\mu_x)$, if*

$$\lambda \geq n^{-K}, \qquad \gamma_\lambda \geq p^{-K}, \qquad \widetilde{\rho}_\lambda(n,p)^{5/2} \cdot \log^{3/2}(n) \leq K\sqrt{n}, \tag{28}$$

$$\widetilde{\rho}_\lambda(n,p)^2 \cdot \rho_{\gamma_+}(p)^8 \cdot \log^4(p) \leq K\sqrt{p}, \tag{29}$$

*then with probability at least $1 - n^{-D} - p^{-D}$, we have*

$$|\mathcal{R}(f_\star; \mathbf{X}, \mathbf{W}, \lambda) - \mathsf{R}_{n,p}(\boldsymbol{\beta}_*, \lambda)| \leq C_{*,D,K} \cdot \mathcal{E}(n, p) \cdot \mathsf{R}_{n,p}(\boldsymbol{\beta}_*, \lambda), \tag{30}$$

*where $\mathsf{R}_{n,p}(\boldsymbol{\beta}_*, \lambda)$ has been defined in eq. (24) and:*

$$\gamma_\lambda = \frac{p\lambda}{n} + \sum_{k=\mathsf{m}+1}^{\infty}\xi_k^2, \qquad \gamma_+ = p\nu_1 + \sum_{k=\mathsf{m}+1}^{\infty}\xi_k^2. \tag{31}$$

*and the approximation rate is given by*

$$\mathcal{E}(n, p) := \frac{\widetilde{\rho}_\lambda(n,p)^6 \log^{7/2}(n)}{\sqrt{n}} + \frac{\widetilde{\rho}_\lambda(n,p)^2 \cdot \rho_{\gamma_+}(p)^8 \log^{7/2}(p)}{\sqrt{p}}. \tag{32}$$

For typical settings, with regularly varying spectrum, $\rho_\kappa(p) \lesssim \log(p)^C/\kappa$ and $\widetilde{\rho}_\kappa(n, p) \lesssim \log(n \wedge p)^C/\kappa$. In this case, the approximation rate scales as $\mathcal{E}(n, p) = \tilde{O}(n^{-1/2} + p^{-1/2})$, which matches the optimal rates expected from local law fluctuations. A few remarks on this theorem are in order:

(a) Theorem 3.3 provides fully non-asymptotic approximation bounds for the population risk and its deterministic equivalent. They hold pointwise and for a large class of functions. In particular, they do not require probabilistic assumptions over the target function coefficients $\boldsymbol{\beta}_\star$, as for instance in [Dobriban and Wager, 2018, Richards et al., 2021, Wu and Xu, 2020].

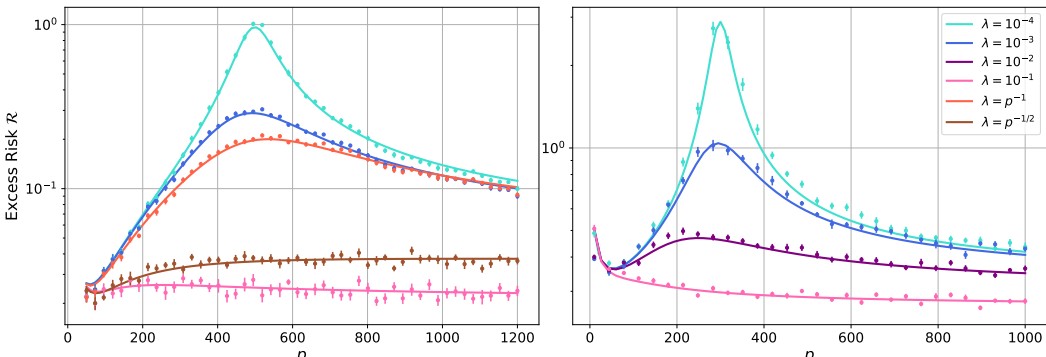

Figure 1: Excess risk eq. (6) of RFRR as a function of the number of features $p$ for a fixed number of samples $n$. Solid lines are obtained from the deterministic equivalent in Theorem 3.3, and points are numerical simulations, with the different curves denoting different regularization strengths $\lambda \geq 0$. **(Left)** Training data $(\boldsymbol{x}_i, y_i)_{i \in [n]}$, $n = 500$, sampled from a teacher-student model $y_i = \text{erf}(\langle \boldsymbol{\beta}, \boldsymbol{x}_i \rangle) + \varepsilon_i$, $\sigma_\varepsilon^2 = 0.1$, $\boldsymbol{x}_i \sim_{\text{i.i.d.}} \mathcal{N}(0, \boldsymbol{I}_d)$, with a spiked random feature map $\varphi(\boldsymbol{x}, \boldsymbol{w}) = \tanh(\langle \boldsymbol{w} + u\boldsymbol{v}, \boldsymbol{x} \rangle)$ where $\boldsymbol{v} \in \mathbb{R}^d$ has a fixed overlap $\gamma = \langle \boldsymbol{v}, \boldsymbol{\beta} \rangle$ with the teacher vector, $\boldsymbol{w} \sim \mathcal{N}(0, d^{-1}\boldsymbol{I}_d)$, $u \sim \mathcal{N}(0,1)$. **(Right)** Training data $(\boldsymbol{x}_i, y_i)_{i \in [n]}$, $n = 300$, sub-sampled from the FashionMNIST data set [Xiao et al., 2017], with feature map given by $\varphi(\boldsymbol{x}; \boldsymbol{w}) = \text{erf}(\langle \boldsymbol{w}, \boldsymbol{x} \rangle)$ and $\mu_w = \mathcal{N}(0, d^{-1}\boldsymbol{I}_d)$.

(b) They are not explicitly dependent on the feature map dimension.

(c) They are multiplicative, and therefore relative to the scale of the risk. In particular, they hold even if $\mathcal{R} \asymp n^{-\gamma}$, which will be crucial to the discussion in section 4.

(d) Theorem 3.3 is considerably more general than previous results. First, it extends the dimension-free results of Cheng and Montanari [2022] for well-specified ridge regression and Misiakiewicz and Saeed [2024] for KRR (see $p \to \infty$ discussion below) to the case of feature maps $\varphi : \mathcal{X} \times \mathcal{W} \to \mathbb{R}$, which, as discussed in Section 2, comprises several cases of interest in machine learning. Moreover, the deterministic equivalent recovers as particular cases the asymptotic results derived under proportional $n, p = \Theta(d)$ [Mei and Montanari, 2022, Loureiro et al., 2022, Schröder et al., 2023] and polynomial $n, p = \Theta(d^\kappa)$ [Xiao et al., 2022, Hu et al., 2024, Aguirre-López et al., 2024] scaling.

(e) The bounds depend on $\lambda^{-1}$ and $\lambda_{>\mathsf{m}}^{-1}$. Following similar arguments as in Cheng and Montanari [2022], Misiakiewicz and Saeed [2024], this assumption could be removed at the cost of a lengthier analysis and worse rates $n^{-C} + p^{-C}$ with $C < 1/2$.

Figure 1 illustrates Theorem 3.3 in two different settings with real and synthetic data. On the left, we show the population risk of learning a single-index target function with a spiked random features model. This model was recently shown to be equivalent to the first-step of training in a fully-connected two-layer network [Ba et al., 2022], and it was recently studied by several authors [Moniri et al., 2024, Cui et al., 2024, Wang et al., 2024]. On the right, we apply our formulas directly to a real data set. In both cases, the theoretical curves show excellent agreement with the numerical simulations. In Appendix C we present additional plots, together with a discussion of how these plots were generated.

**Particular limits —** We now discuss some particular limits of interest of the deterministic equivalent eq. (24). First, note that at the interpolation threshold $n = p$, we have $1 - \Upsilon(\nu_1, \nu_2) \sim \sqrt{\lambda}$. Therefore, the risk $\mathsf{R}_{n,p} \sim \lambda^{-1/2}$ diverges as $\lambda \to 0^+$, a well-known behaviour known as the *interpolation peak* in the random feature literature [Hastie et al., 2022, Mei and Montanari, 2022, Gerace et al., 2021] and observed in neural networks [Spigler et al., 2019, Nakkiran et al., 2021].

Another limit of interest is $p \to \infty$ where, in the generic case, the features span an infinite-dimensional RKHS. Typically, the resulting kernel will be universal, implying it can approximate any function in $L_2(\mu_x)$. In this limit, the risk bottleneck is given by the finite amount of data $n$.

**Corollary 3.4** (Kernel limit). *In the $p \to \infty$ limit both $\nu_1$ and $\nu_2$ converge to a single $\nu_K$ which is the unique positive solution to the following self-consistent equation*

$$n - \frac{\lambda}{\nu_K} = \text{Tr}\big(\boldsymbol{\Sigma}(\boldsymbol{\Sigma} + \nu_K)^{-1}\big). \tag{33}$$

*Moreover, the bias eq. (22) and variance eq. (23) terms simplify to:*

$$\mathsf{B}_{K,n}(\boldsymbol{\beta}_*, \lambda) = \frac{\nu_K^2 \langle \boldsymbol{\beta}_*, (\boldsymbol{\Sigma} + \nu_K)^{-2} \boldsymbol{\beta}_* \rangle}{1 - \frac{1}{n}\text{Tr}(\boldsymbol{\Sigma}^2(\boldsymbol{\Sigma} + \nu_K)^{-2})}, \qquad \mathsf{V}_{K,n}(\lambda) = \sigma_\varepsilon^2 \frac{\text{Tr}(\boldsymbol{\Sigma}^2(\boldsymbol{\Sigma} + \nu_K)^{-2})}{n - \text{Tr}(\boldsymbol{\Sigma}^2(\boldsymbol{\Sigma} + \nu_K)^{-2})}. \tag{34}$$

*We denote the corresponding test error $\mathsf{R}_{K,n}(\boldsymbol{\beta}_*, \lambda) = \mathsf{B}_{K,n}(\boldsymbol{\beta}_*, \lambda) + \mathsf{V}_{K,n}(\lambda)$.*

Note that eq. (23) exactly agrees with the dimension-free deterministic equivalents for kernel methods in Cheng and Montanari [2022], Misiakiewicz and Saeed [2024]. Finally, the third limit of interest is the $n \to \infty$ where data is abundant. In this case, the empirical risk eq. (5) converge to the population risk, and therefore the bottleneck in the risk is given by the capacity of the random feature class $\mathcal{F}_{RF}$ eq. (1) to approximate the target $f_\star$.

**Corollary 3.5** (Approximation limit). *In the $n \to \infty$ limit, we have $\nu_1 \to 0$ and $\nu_2 \to \nu_A$ satisfying the following simplified self-consistent equation:*

$$p = \text{Tr}\big(\boldsymbol{\Sigma}(\boldsymbol{\Sigma} + \nu_A)^{-1}\big). \tag{35}$$

*Moreover, the bias eq. (22) and variance eq. (23) terms simplify to:*

$$\mathsf{B}_{A,p}(\boldsymbol{\beta}_*) = \nu_A \langle \boldsymbol{\beta}_*, (\boldsymbol{\Sigma} + \nu_A)^{-1} \boldsymbol{\beta}_* \rangle, \qquad\qquad \mathsf{V}_{A,n} = 0. \tag{36}$$

*We denote the risk in this case $\mathsf{R}_{A,p}(\boldsymbol{\beta}_*) = \mathsf{B}_{A,p}(\boldsymbol{\beta}_*)$, which as expected does not depend on $\lambda$.*

## 4 Scaling laws

Our exact characterization of the excess risk in Theorem 3.3 shows that the bottleneck in the model performance stems either from its approximation capacity (as measured by the "width" $p$) and the availability of data (as measured by the number of samples $n$). In other words, for a fixed data budget $n$, increasing $p$ might not improve the error besides a certain point, yielding a waste of computational resources. This raises an important question: *given a fixed data budged $n$, what is the optimal choice of model size $p_\star$?*

**Context —** This is a fundamental question in the random feature literature, and was investigated already in the pioneering works of Rahimi and Recht [2007, 2008], who showed that to achieve an excess risk of $O(n^{-1/2})$ requires at most $p = O(n)$ features. This upper bound was considerably refined by Rudi and Rosasco [2017] under classical power law scaling assumptions, also known as *source* and *capacity* conditions in the kernel literature:

$$\text{Tr}\boldsymbol{\Sigma}^{1/\alpha} < \infty, \qquad\qquad ||\boldsymbol{\Sigma}^{-r}\boldsymbol{\beta}_\star||_2 < \infty. \tag{37}$$

where $\alpha \in (1, \infty)$ and $r \in (0, \infty)$, with the case $r = 1/2$ corresponding to $f_\star$ belonging to the RKHS of the asymptotic random feature kernel eq. (2). The optimal minmax rate $O\left(n^{-\frac{2\alpha r}{2\alpha r + 1}}\right)$ for ridge regression under source and capacity conditions were obtained by Caponnetto and De Vito [2007]. Rudi and Rosasco [2017] showed that this optimal rate can be attained by the random feature hypothesis eq. (1) with $p > p_0 = O\left(n^{\frac{\alpha - 1 + 2r}{1 + 2\alpha r}}\right)$ features. However, this is only an upper bound, and understanding how tight it is, as well as the full picture in the hard regime $r \in (0, 1/2)$, remains an open question. In this section, we leverage our tight characterization of the excess risk in Theorem 3.3 to provide a sharp answer to this question.

**Results —** Without loss of generality, we can assume the covariance is a diagonal matrix $\boldsymbol{\Sigma} = \text{diag}(\xi_k^2)_{k \geq 1}$, and we consider the case where the exponents exactly saturate the source and capacity conditions eq. (37):

$$\xi_k^2 = k^{-\alpha}, \qquad\qquad \beta_{*,k} = k^{-\frac{1+2\alpha r}{2}}. \tag{38}$$

Further, we assume a relative scaling of the number of features $p$ and the regularization $\lambda$ with the number of samples $n$:

$$p = n^q, \qquad\qquad \lambda = n^{-(\ell-1)}. \tag{39}$$

with $q \geq 0$ and $\ell \geq 0$.

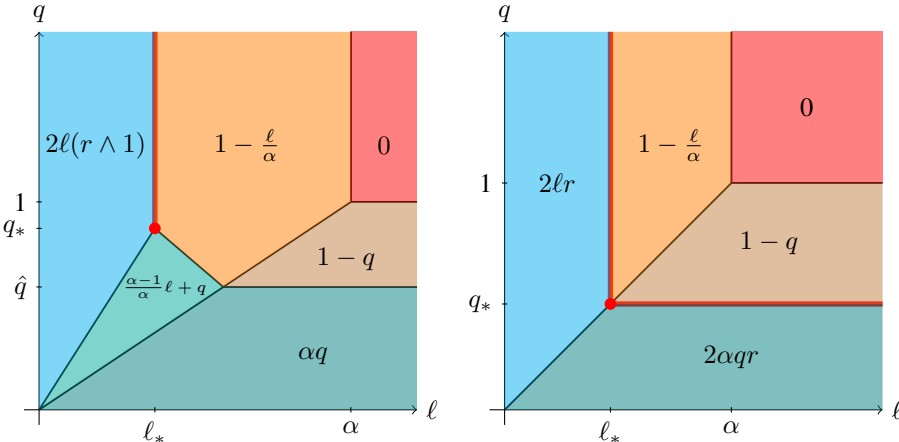

Figure 2: Excess error rate $\gamma$ in the regime $n \gg \sigma_\varepsilon^{-1/(\gamma_\mathcal{B}(\ell,q) - \gamma_\mathcal{V}(\ell,q))}$ as a function of $(\ell, q)$, defined in eq. (40) and eq. (39) for $r \geq 1/2$ (**Left**) and $r \in [0, 1/2)$ (**Right**). The explicit crossover points $\ell_\star, q_\star, \hat{q}$ are defined in eq. (43) as a function of the source $r$ and capacity $\alpha$ exponents.

**Theorem 4.1** (Excess risk rates). *Under source and capacity conditions eq. (38) and scaling assumptions eq. (39), the deterministic equivalent eq. (24) rate is given by:*

$$\mathsf{R}_{n,p}(\beta_\star, \lambda) = \Theta\left(n^{-\gamma_\mathcal{B}(\ell,q)} + \sigma_\varepsilon^2 n^{-\gamma_\mathcal{V}(\ell,q)}\right) = \Theta\left(n^{-\gamma(\ell,q)}\right), \tag{40}$$

*where $\gamma(\ell, q) := \gamma_\mathcal{B}(\ell,q) \wedge \gamma_\mathcal{V}(\ell,q)$ for non-zero noise variance $\sigma_\varepsilon^2 \neq 0$, otherwise $\gamma(\ell,q) = \gamma_\mathcal{B}(\ell,q)$. The exponents $\gamma_\mathcal{B}$ and $\gamma_\mathcal{V}$ are respectively the decay rates of the bias and variance terms eqs. (22) and (23), and are explicitly given by*

$$\gamma_\mathcal{B} := \left[2\alpha\left(\frac{\ell}{\alpha} \wedge q \wedge 1\right)(r \wedge 1)\right] \wedge \left[\left(2\alpha\left(r \wedge \frac{1}{2}\right) - 1\right)\left(\frac{\ell}{\alpha} \wedge q \wedge 1\right) + q\right], \tag{41}$$

$$\gamma_\mathcal{V} := 1 - \left(\frac{\ell}{\alpha} \wedge q \wedge 1\right). \tag{42}$$

**Remark 4.1.** *Under the scaling in eqs. (38) and (39), one can check that the approximation rates $\mathcal{E}(n, p)$ in Theorem 3.3 are vanishing for $\ell \leq \alpha + 1/12$ if $q \geq 1$, and for $\ell \leq q((\alpha + 1/16) \vee 1/16(\alpha - 1))$ if $q < 1$, which includes the optimal vertical line $\ell = \ell_\star$. Hence, for these regions of scaling, Theorem 3.3 readily implies that the excess risk eq. (6) indeed has the decay rates described in Theorem 4.1. As discussed in the previous section, we expect that these approximation guarantees can be improved to include a larger region of decay rates, but we leave it to future work.*

A detailed derivation of the result above from the deterministic equivalent characterization from Theorem 3.3 is discussed in Appendix D. The expressions in eq. (41) are easier to visualise in a diagram. Figure 2 shows the excess risk exponent $\gamma(\ell, q)$ as a function of the parameters $\ell$ and $q$, in the case where $\sigma_\varepsilon^2 \neq 0$ for $r \geq 1/2$ (left) and $r < 1/2$ (right). Note that the key difference between the diagrams is the presence of an additional region for $r \geq 1/2$.[2] Defining the following shorthand:

$$\ell_\star := \frac{\alpha}{2\alpha(r \wedge 1) + 1}, \qquad q_\star := 1 - \ell_\star(2r \wedge 1), \qquad \hat{q} := \frac{1}{\alpha(2r \wedge 1) + 1} = q_\star \vee \frac{1}{\alpha + 1} \tag{43}$$

we can identify two main regions in the $(\ell, q)$ plane, corresponding to a trade-off between the bias $\gamma_\mathcal{B}$ and variance $\gamma_\mathcal{V}$ terms:

(a) **Variance dominated region** ($\gamma_\mathcal{V} < \gamma_\mathcal{B}$): if $\ell > \ell_\star$, $q > \hat{q}$ and $p > \lambda$, the excess risk is dominated by the variance term, provided the number of samples is large enough $n \gg \sigma_\varepsilon^{-1/(\gamma_\mathcal{B}(\ell,q) - \gamma_\mathcal{V}(\ell,q))}$.[3] Inside this region it is possible to further distinguish between two regimes:

---

[2]Recall that these two cases correspond to the target function $f_\star$ belonging ($r \geq 1/2$) or not ($r < 1/2$) to the RKHS spanned by the asymptotic kernel

[3]The noise dominated regime where $n < \sigma_\varepsilon^{-1/(\gamma_\mathcal{B}(\ell,q) - \gamma_\mathcal{V}(\ell,q))}$ and the corresponding cross-over was studied by Cui et al. [2022]. A similar phenomenology hold here, but for simplicity we focus the discussion on the data dominated regime.

- **slow decay regime** (orange and brown): for $\ell < \alpha$ and $q < 1$ ($p \ll n$), $\gamma_\mathcal{V} = 1 - (\ell/\alpha \wedge q)$, hence the decay depends on the interplay between regularization strength and number of random features and it is slower as $(\ell/\alpha \wedge q)$ increases;
- **plateau regime** (red): for $\ell \geq \alpha$ and $q \geq 1$ ($p \geq n$) the excess risk converges to a constant value and does not decay as $n$ increases.

(b) **Bias dominated region** ($\gamma_\mathcal{V} > \gamma_\mathcal{B}$): if $\ell < \ell_\star$, $q < \hat{q}$ and $p < \lambda$, the excess risk is dominated by the bias term, whose decay is faster as $(\ell/\alpha \wedge q)$ increases (cyan, emerald and teal).

Note that in the limit of large number of random features $p \to \infty$, we recover the same rates found by Cui et al. [2022] for kernel ridge regression. Of particular interest is the rate for which the excess risk decays the fastest with the number of samples $n$, and what is the minimum number of random features $p_\star$ required to achieve this rate.

**Corollary 4.2** (Optimal rates). *The optimal excess risk rate achieved by the random features hypothesis eq. (1) under source and capacity conditions eq. (38) and scaling assumptions eq. (39):*

$$\gamma_\star = \max_{\ell,q} \gamma(\ell, q) = \frac{2\alpha(r \wedge 1)}{2\alpha(r \wedge 1) + 1}, \tag{44}$$

*and it is attained for:*

$$\begin{cases} \lambda = \lambda_\star := n^{-(\ell_\star - 1)} \\ p \geq p_\star := n^{q*} = \lambda_\star \end{cases} \qquad \textit{for } r \geq 1/2, \tag{45}$$

$$\begin{cases} \lambda = \lambda_\star \\ p \geq p_\star = \left(\lambda_\star^{-1} n\right)^{1/\alpha} \end{cases} \quad or \quad \begin{cases} \lambda \leq \lambda_\star \\ p = p_\star = \left(\lambda_\star^{-1} n\right)^{1/\alpha} \end{cases} \qquad \textit{for } r < 1/2 \tag{46}$$

*corresponding to the bold red line (—) in Fig. 2. In particular, the minimal number of random features $p_\star = n^{q_\star}$ required to achieve the optimal rate $\gamma_\star$ is given by:*

$$q_\star = 1 - \frac{\alpha(2r \wedge 1)}{2\alpha(r \wedge 1) + 1} \tag{47}$$

*and corresponds to the bold red dot (•) in Fig. 2.*

A few comments on Corollary 4.2 are in place. 1) The optimal excess error rate eq. (44) is consistent with the minimax optimal rates for ridge regression from Caponnetto and De Vito [2007], as also discussed by Rudi and Rosasco [2017]. 2) The minimal number of random features $p_\star = n^{q_\star}$ in eq. (47) achieving the optimal rate eq. (44) in the $r \geq 1/2$ regime is strictly smaller than the lower bound $p > p_0$ of Rudi and Rosasco [2017]. More precisely, letting $p_0 = n^{q_0}$, for $r \in [1/2, 1)$:

$$q_0 - q_\star = \frac{2(1-r)(\alpha - 1)}{2\alpha r + 1} > 0, \qquad \text{for all } \alpha > 1. \tag{48}$$

**Relationship to scaling laws —** The empirical observation that the performance of large scale neural networks decreases as a power law with respect to the number of samples, parameter and computing time has sparked a renewed wave of interest in the theoretical investigation of power laws [Kaplan et al., 2020]. Despite being a mature topic in the statistical learning literature, different recent works have turned to the study of linear models under source and capacity conditions as a playground to understand the emergence of different bottlenecks in the excess error rates [Bahri et al., 2024, Maloney et al., 2022].

The model studied in these works is given by ridge regression on data $y_i = \langle \boldsymbol{\beta}_\star, \boldsymbol{x}_i \rangle$ with $\boldsymbol{x} \sim \mathcal{N}(\mathbf{0}, \mathrm{diag}((d/k)^\alpha))$ and $\boldsymbol{\beta}_\star \sim \mathcal{N}(\mathbf{0}, 1/d \boldsymbol{I}_d)$ with a linear projection model $\hat{f}(\boldsymbol{x}, \boldsymbol{a}) = \langle \boldsymbol{a}, \boldsymbol{W} \boldsymbol{x} \rangle$, where $\boldsymbol{W}$ is an i.i.d. Gaussian matrix. Note this model is a particular case of the one studied here, corresponding to a linear feature map and random target function. Moreover, since the variance of the target is constant, the source is entirely determined by the capacity $\alpha$ of the asymptotic kernel, here controlled by the decay of the covariance of the input data.

The approximation limit from Corollary 3.5 and the kernel limit from Corollary 3.4 are known in this literature as *Variance* and *Resolution limited regimes*, respectively [Bahri et al., 2024]. They correspond precisely to the bottlenecks in the excess risk arising from the limited approximation

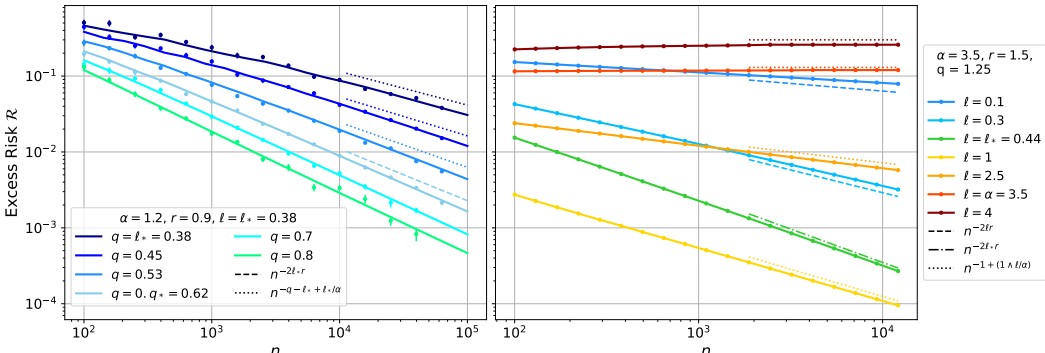

Figure 3: Excess risk eq. (6) of RFRR as a function of the number of samples $n$ under source and capacity conditions eq. (37) and power-law assumptions $\lambda = n^{-(\ell-1)}$, $p = n^q$, with noise variance $\sigma_\varepsilon^2 = 0.1$. Solid lines are obtained from the deterministic equivalent Theorem 3.3. In the figure on the left, points are finite size numerical experiments. Dashed and dotted lines are the analytical rates from Theorem 4.1, stated in the legend. The colour scheme corresponds to the regions of Fig. 2.

capacity of the random feature model or the limited availability of training data. As this model is a particular case of ours, the rates in the variance limited regime can also be obtained from Theorem 4.1, and correspond to particular cases in Fig. 2, see Appendix E for a detailed discussion. Contemporary to our work, Atanasov et al. [2024] has extended the analysis in this linear model to the case where $\beta_\star$ also has a power-law decay, and provided a comprehensive discussion of the different scaling regimes for this model. Their rates can be put in a one-to-one correspondence with the rates derived in section 4. We refer the interested reader to Section VI.6 of Atanasov et al. [2024] for a detailed discussion of this relationship. We stress, however, that beside being rigorous, our results hold for features in infinite-dimensional Hilbert spaces and are not restricted to a particular asymptotic limit in the dimensions.

Complementary to the sample and model complexity bottlenecks, Kaplan et al. [2020] also observed the emergence of computational scaling laws in the risk as a function of flops used in training. A recent line of work has investigated this question on the aforementioned linear random feature model under different training algorithms, such as gradient flow [Bordelon et al., 2024] and SGD [Paquette et al., 2024, Lin et al., 2024]. Due to the simplicity of this setting, the risk of ridge regression with a particular choice of regularization $\lambda$ is closely related to the risk of different descent algorithms for least-squares at a fixed running horizon [Ali et al., 2019, 2020, Sonthalia et al., 2024]. A similar analogy allows us to compare our results to the ones obtained in [Paquette et al., 2024, Lin et al., 2024]. In particular, our setting cover three of the phases identified by Paquette et al. [2024], and correspond to the result in Theorem 4.1 with $\lambda = 1$ ($\ell = 1$). Similarly, the rates of Lin et al. [2024] are obtained by taking $\lambda$ to be the inverse of the learning rate. A detailed connection to this line of work is discussed in Appendix E.

## 5   Conclusion

In this paper, we have investigated the generalization properties of random feature models, deriving a non-asymptotic deterministic equivalent for the risk of random feature ridge regression—which recovers (and unifies) previous asymptotic findings as special limits. Our results provide a rigorous multiplicative approximation rate, enabling us to analyze error scaling laws under source and capacity conditions, and offers a complete view of the different scaling regimes and their cross-overs. Our analysis relies on Assumption 3.1 which, while popular in theoretical investigations, excludes more realistic random feature models, such as $\varphi(\boldsymbol{x}, \boldsymbol{w}) = \sigma(\boldsymbol{x}^\top \boldsymbol{w})$ with $\boldsymbol{x}, \boldsymbol{w}$ Gaussian vectors and non-linear $\sigma$. Although restrictive, this assumption allowed us to derive tight multiplicative approximation bounds for a generic random feature model with infinite-dimensional features—which was essential for obtaining the rigorous excess risk rates that are the primary motivation of our work. We further note that numerical simulations in Figure 1 and Appendix C suggest that the predictions of Theorem 3.3 remain accurate much beyond Assumption 3.1. We consider lifting this technical condition—e.g., by following the approach in Misiakiewicz and Saeed [2024]—to be an important direction for future research.

## Acknowledgements

We would like to thank Yasaman Bahri, Hugo Cui and Florent Krzakala for stimulating discussions. BL & LD acknowledges funding from the *Choose France - CNRS AI Rising Talents* program.

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

# Appendix

## A  Background on deterministic equivalents

We consider a feature vector $\boldsymbol{f} \in \mathbb{R}^q$, $q \in \mathbb{N} \cup \{\infty\}$, with covariance matrix $\boldsymbol{\Sigma} = \mathbb{E}[\boldsymbol{f}\boldsymbol{f}^\mathsf{T}]$. We denote $\gamma_1^2 \geq \gamma_2^2 \geq \gamma_3^2 \geq \cdots$ the eigenvalues of $\boldsymbol{\Sigma}$ in non-increasing order. In the case of infinite-dimensional features $q = \infty$, we will further assume that $\mathrm{Tr}(\boldsymbol{\Sigma}) < \infty$, i.e., we consider $\boldsymbol{\Sigma}$ to be a trace-class self-adjoint operator.

We assume that the feature $\boldsymbol{f}$ satisfies the following assumption.

**Assumption A.1** (Feature concentration). *There exist* $\mathsf{c}_*, \mathsf{C}_* > 0$ *such that for any p.s.d. matrix* $\boldsymbol{A} \in \mathbb{R}^{q \times q}$ *with* $\mathrm{Tr}(\boldsymbol{\Sigma}\boldsymbol{A}) < \infty$, *we have*

$$\mathbb{P}\left(\left|\boldsymbol{f}^\mathsf{T}\boldsymbol{A}\boldsymbol{f} - \mathrm{Tr}(\boldsymbol{\Sigma}\boldsymbol{A})\right| \geq t \cdot \|\boldsymbol{\Sigma}^{1/2}\boldsymbol{A}\boldsymbol{\Sigma}^{1/2}\|_F\right) \leq \mathsf{C}_* \exp\left\{-\mathsf{c}_* t\right\}. \tag{49}$$

Recall that we denote $C_{a_1,\ldots,a_k}$ constants that only depend on the values of $\{a_i\}_{i \in [k]}$. We use $a_i = \text{`}*\text{'}$ to denote the dependency on the constants $\mathsf{c}_*, \mathsf{C}_*$ from Assumption A.1.

We will further define the following two quantities.

**Definition 2** (Effective regularization). *For an integer $n$, covariance $\boldsymbol{\Sigma}$, and regularization $\lambda \geq 0$, we define the* effective regularization $\lambda_*$ *associated to model* $(n, \boldsymbol{\Sigma}, \lambda)$ *to be the unique non-negative solution to the equation*

$$n - \frac{\lambda}{\lambda_*} = \mathrm{Tr}\big(\boldsymbol{\Sigma}(\boldsymbol{\Sigma} + \lambda_*)^{-1}\big). \tag{50}$$

Throughout this appendix, we assume that $\lambda > 0$. The existence and uniqueness follow from noticing that the left-hand side is monotonically increasing in $\lambda_*$ while the right-hand side is monotonically decreasing. We consider the change of variable $\mu_* := \mu_*(\lambda) = \lambda/\lambda_*$, such that $\mu_*$ is the unique non-negative solution of

$$\mu_* = \frac{n}{1 + \mathrm{Tr}(\boldsymbol{\Sigma}(\mu_*\boldsymbol{\Sigma} + \lambda)^{-1})}. \tag{51}$$

Both $\mu_*$ and $\lambda_*$ are increasing functions with $\lambda$.

**Definition 3** (Intrinsic dimension). *For a covariance matrix $\boldsymbol{\Sigma} \in \mathbb{R}^{q \times q}$ with eigenvalues in nonincreasing order $\gamma_1^2 \geq \gamma_2^2 \geq \gamma_3^2 \geq \cdots$, we define the* intrinsic dimension $r_{\boldsymbol{\Sigma}}(k)$ *at level $k \in \mathbb{N}$ of $\boldsymbol{\Sigma}$ to be the intrinsic dimension of the covariance matrix $\boldsymbol{\Sigma}_{\geq k} = \mathrm{diag}(\gamma_k^2, \gamma_{k+1}^2, \ldots)$, i.e., the covariance matrix projected orthogonally to the top $k - 1$ eigenspaces, which is given by*

$$r_{\boldsymbol{\Sigma}}(k) := \frac{\mathrm{Tr}(\boldsymbol{\Sigma}_{\geq k})}{\|\boldsymbol{\Sigma}_{\geq k}\|_{\mathrm{op}}} = \frac{\sum_{j=k}^q \gamma_j^2}{\gamma_k^2}.$$

The intrinsic dimension of $\boldsymbol{\Sigma}_{\geq k}$ captures the number of dimensions of $\boldsymbol{\Sigma}_{\geq k}$ that have significant spectral content, i.e., $\gamma_j^2 \approx \|\boldsymbol{\Sigma}_{\geq k}\|_{\mathrm{op}}$ (see [Tropp et al., 2015, Chapter 7] for further background).

We are given $n$ i.i.d. features $(\boldsymbol{f}_i)_{i \in [n]}$, and we denote $\boldsymbol{F} = [\boldsymbol{f}_1, \ldots, \boldsymbol{f}_n]^\mathsf{T} \in \mathbb{R}^{n \times q}$ the feature matrix. The train and test errors are functionals of the feature matrix $\boldsymbol{F}$. In particular, they depend on the following resolvent matrix

$$\boldsymbol{R} = (\boldsymbol{F}^\mathsf{T}\boldsymbol{F} + \lambda \mathbf{I}_q)^{-1}.$$

In this section we consider functionals that depend on products of $\boldsymbol{F}$, $\boldsymbol{R}$ and deterministic matrices.

For a general p.s.d. matrix $\boldsymbol{A} \in \mathbb{R}^{q \times q}$, define the functionals

$$\Phi_1(\boldsymbol{F}; \boldsymbol{A}, \lambda) := \mathrm{Tr}\left(\boldsymbol{A}\boldsymbol{\Sigma}^{1/2}(\boldsymbol{F}^\mathsf{T}\boldsymbol{F} + \lambda)^{-1}\boldsymbol{\Sigma}^{1/2}\right),$$

$$\Phi_2(\boldsymbol{F}; \lambda) := \mathrm{Tr}\left(\frac{\boldsymbol{F}^\mathsf{T}\boldsymbol{F}}{n}(\boldsymbol{F}^\mathsf{T}\boldsymbol{F} + \lambda)^{-1}\right),$$

$$\Phi_3(\boldsymbol{F}; \boldsymbol{A}, \lambda) := \mathrm{Tr}\left(\boldsymbol{A}\boldsymbol{\Sigma}^{1/2}(\boldsymbol{F}^\mathsf{T}\boldsymbol{F} + \lambda)^{-1}\boldsymbol{\Sigma}(\boldsymbol{F}^\mathsf{T}\boldsymbol{F} + \lambda)^{-1}\boldsymbol{\Sigma}^{1/2}\right),$$

$$\Phi_4(\boldsymbol{F}; \boldsymbol{A}, \lambda) := \mathrm{Tr}\left(\boldsymbol{A}\boldsymbol{\Sigma}^{1/2}(\boldsymbol{F}^\mathsf{T}\boldsymbol{F} + \lambda)^{-1}\frac{\boldsymbol{F}^\mathsf{T}\boldsymbol{F}}{n}(\boldsymbol{F}^\mathsf{T}\boldsymbol{F} + \lambda)^{-1}\boldsymbol{\Sigma}^{1/2}\right).$$

These functionals are well approximated by quantities proportional to

$$\Psi_1(\lambda_*; \boldsymbol{A}) := \mathrm{Tr}\left(\boldsymbol{A}\boldsymbol{\Sigma}(\boldsymbol{\Sigma} + \lambda_*)^{-1}\right),$$

$$\Psi_2(\lambda_*) := \frac{1}{n}\mathrm{Tr}\left(\boldsymbol{\Sigma}(\boldsymbol{\Sigma} + \lambda_*)^{-1}\right),$$

$$\Psi_3(\lambda_*; \boldsymbol{A}) := \frac{1}{n} \cdot \frac{\mathrm{Tr}(\boldsymbol{A}\boldsymbol{\Sigma}^2(\boldsymbol{\Sigma} + \lambda_*)^{-2})}{n - \mathrm{Tr}(\boldsymbol{\Sigma}^2(\boldsymbol{\Sigma} + \lambda_*)^{-2})}.$$

Without loss of generality, we can assume that $\mathrm{Tr}(\boldsymbol{A}\boldsymbol{\Sigma}) < \infty$ for $\Phi_1$, as otherwise $\Phi_1(\boldsymbol{F}; \boldsymbol{A}, \lambda) = \Psi_1(\mu_*; \boldsymbol{A}, \lambda) = \infty$ almost surely, and $\mathrm{Tr}(\boldsymbol{A}\boldsymbol{\Sigma}^2) < \infty$ for $\Phi_3$ and $\Phi_4$, as otherwise $\Phi_j(\boldsymbol{F}; \boldsymbol{A}, \lambda) = \Psi_j(\mu_*; \boldsymbol{A}, \lambda) = \infty$, $j = 3, 4$, almost surely.

Our relative approximation bound will depend on the covariance matrix $\boldsymbol{\Sigma}$ through

$$\rho_\lambda(n) = 1 + \frac{n\gamma_{\lfloor \eta_* \cdot n \rfloor}^2}{\lambda}\left\{1 + \frac{r_{\boldsymbol{\Sigma}}(\lfloor \eta_* \cdot n \rfloor) \vee n}{n}\log\left(r_{\boldsymbol{\Sigma}}(\lfloor \eta_* \cdot n \rfloor) \vee n\right)\right\}, \tag{52}$$

where $\eta_* \in (0, 1/2)$ is a constant that will only depend on $\mathsf{c}_*, \mathsf{C}_*$ and we used the convention that $\gamma_{\lfloor \eta_* \cdot n \rfloor}^2 = 0$ if $\lfloor \eta_* \cdot n \rfloor > q$.

The following theorem gathers the approximation guarantees for the different functionals stated above, and is obtained by modifying [Misiakiewicz and Saeed, 2024, Theorem 4].

**Theorem A.2** (Dimension-free deterministic equivalents). *Assume the features $(\boldsymbol{f}_i)_{i\in[n]}$ satisfy Assumption A.1 with some constants $\mathsf{c}_x, \mathsf{C}_x, \beta > 0$. For any $D, K > 0$, there exist constants $\eta := \eta_x \in (0, 1/2)$ (only depending on $\mathsf{c}_x, \mathsf{C}_x, \beta$), $C_{D,K} > 0$ (only depending on $K, D$), and $C_{x,D,K} > 0$ (only depending on $\mathsf{c}_x, \mathsf{C}_x, \beta, D, K$), such that the following holds. For all $n \geq C_{D,K}$ and $\lambda > 0$, if it holds that*

$$\lambda \cdot \rho_\lambda(n) \geq \|\boldsymbol{\Sigma}\|_{\mathrm{op}} \cdot n^{-K}, \qquad \rho_\lambda(n)^{5/2} \log^{3/2}(n) \leq K\sqrt{n}, \tag{53}$$

*then for any p.s.d. matrix $\boldsymbol{A}$, we have with probability at least $1 - n^{-D}$ that*

$$\left|\Phi_1(\boldsymbol{F}; \boldsymbol{A}, \lambda) - \frac{\lambda_*}{\lambda}\Psi_1(\lambda_*; \boldsymbol{A})\right| \leq C_{x,D,K}\frac{\rho_\lambda(n)^{5/2}\log^{3/2}(n)}{\sqrt{n}} \cdot \frac{\lambda_*}{\lambda}\Psi_1(\lambda_*; \boldsymbol{A}), \tag{54}$$

$$\left|\Phi_2(\boldsymbol{F}; \lambda) - \Psi_2(\lambda_*)\right| \leq C_{x,D,K}\frac{\rho_\lambda(n)^{5/2}\log^{3/2}(n)}{\sqrt{n}}\Psi_2(\lambda_*), \tag{55}$$

$$\left|\Phi_3(\boldsymbol{F}; \boldsymbol{A}, \lambda) - \left(\frac{n\lambda_*}{\lambda}\right)^2\Psi_3(\mu_*; \boldsymbol{A}, \lambda)\right| \leq C_{x,D,K}\frac{\rho_\lambda(n)^6\log^{5/2}(n)}{\sqrt{n}} \cdot \left(\frac{n\lambda_*}{\lambda}\right)^2\Psi_3(\mu_*; \boldsymbol{A}, \lambda), \tag{56}$$

$$\left|\Phi_4(\boldsymbol{F}; \boldsymbol{A}, \lambda) - \Psi_3(\lambda_*; \boldsymbol{A})\right| \leq C_{x,D,K}\frac{\rho_\lambda(n)^6\log^{3/2}(n)}{\sqrt{n}}\Psi_3(\lambda_*; \boldsymbol{A}). \tag{57}$$

*Proof of Theorem A.2.* The only difference between this theorem and [Misiakiewicz and Saeed, 2024, Theorem 4] comes from the definition of $\rho_\lambda(n)$. This new definition is obtained by slightly modifying the proof bounding the operator norm of $\boldsymbol{\Sigma}^{1/2}\boldsymbol{R}\boldsymbol{\Sigma}^{1/2}$ from [Cheng and Montanari, 2022, Lemma 7.2] and [Misiakiewicz and Saeed, 2024, Lemma 1]. In particular, we will simply modify step 2 in the proof of [Misiakiewicz and Saeed, 2024, Lemma 1]. Consider $\boldsymbol{F}_+ = [\boldsymbol{f}_{+,1}, \ldots, \boldsymbol{f}_{+,n}]^\mathsf{T} \in \mathbb{R}^{n\times(q-k_*)}$ where $\boldsymbol{f}_{+,i}$ correspond to the projection orthogonal to the top $k_* := \lfloor \eta_* n \rfloor - 1$ eigenspaces with covariance matrix

$$\boldsymbol{\Sigma}_+ := \mathbb{E}[\boldsymbol{f}_{+,i}\boldsymbol{f}_{+,i}^\mathsf{T}] = \mathrm{diag}(\gamma_{k_*}^2, \gamma_{k_*+1}^2, \ldots, \gamma_q^2).$$

Then, denoting $\boldsymbol{S} = \sum_{i\in[n]}\boldsymbol{S}_i$ with $\boldsymbol{S}_i := \boldsymbol{f}_{+,i}\boldsymbol{f}_{i,+}^\mathsf{T}$, we have with probability at least $1 - n^{-D}$, for any $i \in [n]$,

$$\|\boldsymbol{S}_i\|_{\mathrm{op}} \leq \mathrm{Tr}(\boldsymbol{\Sigma}_+) + C_{*,D} \cdot \log(n)\sqrt{\gamma_{k_*}^2\,\mathrm{Tr}(\boldsymbol{\Sigma}_+)} \leq \mathrm{Tr}(\boldsymbol{\Sigma}_+)\left(1 + C_{*,D}\frac{\log(n)}{\sqrt{r_{\boldsymbol{\Sigma}}(k_*)}}\right) =: L_n.$$

Denoting $\widetilde{\boldsymbol{S}} = \sum_{i \in [n]} \widetilde{\boldsymbol{S}}_i$ with $\widetilde{\boldsymbol{S}}_i := \boldsymbol{S}_i \mathbb{1}_{\|\boldsymbol{S}_i\|_{\mathrm{op}} \leq L_n}$, so that $\widetilde{\boldsymbol{S}} = \boldsymbol{S}$ with probability at least $1 - n^{-D}$, we obtain

$$\|\widetilde{\boldsymbol{S}}\|_{\mathrm{op}} \leq n L_n \gamma_{k_*}^2 =: v_n, \qquad \|\mathbb{E}[\widetilde{\boldsymbol{S}}]\|_{\mathrm{op}} \leq n \|\mathbb{E}[\boldsymbol{S}_i]\|_{\mathrm{op}} = n \gamma_{k_*}^2.$$

Therefore, applying the matrix Bernstein's inequality with intrinsic dimension [Tropp et al., 2015, Theorem 7.3.1] to $\widetilde{\boldsymbol{S}}$ gives that with probability at least $1 - n^{-D}$,

$$
\begin{aligned}
\|\widetilde{\boldsymbol{S}}\|_{\mathrm{op}} &\leq n \gamma_{k_*}^2 + C_D \left( \sqrt{v_n} + L_n \right) \sqrt{\log(r_{\boldsymbol{\Sigma}}(k_*)n)} \\
&\leq n \gamma_{k_*}^2 + C_D L_n \log(r_{\boldsymbol{\Sigma}}(k_*)n) \\
&\leq n \gamma_{k_*}^2 \left\{ 1 + \frac{r_{\boldsymbol{\Sigma}}(k_*)}{n} \left( 1 + C_{*,D} \frac{\log(n)}{\sqrt{r_{\boldsymbol{\Sigma}}(k_*)}} \right) \log(r_{\boldsymbol{\Sigma}}(k_*)n) \right\}.
\end{aligned}
$$

Note that by the condition of our theorem, $\log(n) \leq K\sqrt{n}$, and therefore

$$
\begin{aligned}
\|\widetilde{\boldsymbol{S}}\|_{\mathrm{op}} &\leq C_{*,D,K} \cdot n \gamma_{k_*}^2 \left\{ 1 + \frac{r_{\boldsymbol{\Sigma}}(k_*)}{n} \left( 1 + \sqrt{\frac{n}{r_{\boldsymbol{\Sigma}}(k_*)}} \right) \log(r_{\boldsymbol{\Sigma}}(k_*)n) \right\} \\
&\leq C_{*,D,K} \cdot n \gamma_{k_*}^2 \left\{ 1 + \frac{r_{\boldsymbol{\Sigma}}(k_*) \vee n}{n} \log(r_{\boldsymbol{\Sigma}}(k_*) \vee n) \right\}.
\end{aligned}
$$

Following the rest of the argument in [Misiakiewicz and Saeed, 2024, Lemma 1] we obtain $\rho_\lambda(n)$ in Eq. (52). $\qquad\square$

## B  Proof of the deterministic equivalent for RFRR

In this appendix, we prove the approximation guarantees stated in Theorem 3.3 between the test error of RFRR and its deterministic equivalent. We start in Section B.1 by introducing background and notations that we will use throughout the proof. Section B.2 introduces key results on the covariance matrix and the fixed points. We then leverage these results to prove deterministic equivalents for different functionals of $\boldsymbol{Z} = (\sigma(\langle \boldsymbol{x}_i, \boldsymbol{w}_j \rangle))_{i \in [n], j \in [p]} \in \mathbb{R}^{n \times p}$ conditional on $(\boldsymbol{w}_j)_{j \in [p]}$ in Section B.3, and functionals of $\boldsymbol{F} = (\xi_k \phi_k(\boldsymbol{w}_j))_{j \in [p], k \geq 1} \in \mathbb{R}^{p \times \infty}$ in Section B.4. Given these deterministic equivalents, we prove our approximation guarantees for the variance term in Section B.5, and for the bias term in B.6. Finally, we deffer the proof of some technical results to Section B.7.

### B.1  Preliminaries

Recall that throughout the paper, we will keep track of the parameters of the problem $(n, p, \boldsymbol{\Sigma}, \lambda, \sigma_\varepsilon^2)$. For the other constants $C_*, K, D$, we will denote $C_{a_1, a_2, \ldots, a_k}$ constants that only depend on the values of $\{a_i\}_{i \in [k]}$. We use $a_i = $ '$*$' to denote the dependency on the constant $C_*$ appearing in Assumption 3.1 and Assumption 3.2.

Throughout this appendix, we will directly work in the 'feature space'

$$\boldsymbol{g}_i := (\psi_k(\boldsymbol{x}_i))_{k \geq 1}, \qquad \text{and} \qquad \boldsymbol{f}_j := (\xi_k \phi_k(\boldsymbol{w}_j))_{k \geq 1},$$

with distribution induced by $\boldsymbol{x}_i \sim \mu_x$ and $\boldsymbol{w}_j \sim \mu_w$. We will denote the covariate feature and weight feature matrices by

$$\boldsymbol{G} := [\boldsymbol{g}_1, \ldots, \boldsymbol{g}_n]^{\mathsf{T}} \in \mathbb{R}^{n \times \infty}, \qquad \boldsymbol{F} := [\boldsymbol{f}_1, \ldots, \boldsymbol{f}_p]^{\mathsf{T}} \in \mathbb{R}^{p \times \infty}.$$

We denote the random feature weight vector

$$\boldsymbol{z}_i := \frac{1}{\sqrt{p}} [\sigma(\langle \boldsymbol{w}_1, \boldsymbol{x}_i \rangle), \ldots, \sigma(\langle \boldsymbol{w}_p, \boldsymbol{x}_i \rangle)] = \frac{1}{\sqrt{p}} \boldsymbol{F} \boldsymbol{g}_i \in \mathbb{R}^p,$$

and the associated feature matrix

$$\boldsymbol{Z} = [\boldsymbol{z}_1, \ldots, \boldsymbol{z}_n]^{\mathsf{T}} = \frac{1}{\sqrt{p}} \boldsymbol{G} \boldsymbol{F}^{\mathsf{T}} \in \mathbb{R}^{n \times p}.$$

Note that $\boldsymbol{f}$ has covariance matrix

$$\boldsymbol{\Sigma} := \mathbb{E}[\boldsymbol{f} \boldsymbol{f}^{\mathsf{T}}] = \mathrm{diag}(\xi_1^2, \xi_2^2, \xi_3^2, \ldots).$$

We will further introduce the covariance matrix of $z$ conditional on the weight feature matrix $F$ (i.e., conditional on $(w_j)_{j \in [p]}$)

$$\widehat{\Sigma}_F := \mathbb{E}_z \left[ zz^\mathsf{T} \Big| F \right] = \frac{1}{p} FF^\mathsf{T} \in \mathbb{R}^{p \times p}. \tag{58}$$

Note that under Assumption 3.1, the features $z$ and $f$ satisfy the following assumption.

**Assumption B.1** (Concentration of the features $z$ and $f$). *There exists a constant $\mathsf{C}_* > 0$ such that for any weight feature matrix $F \in \mathbb{R}^{p \times \infty}$ and deterministic p.s.d. matrix $A \in \mathbb{R}^{p \times p}$ with $\mathrm{Tr}(\widehat{\Sigma}_F A) < \infty$, we have*

$$\mathbb{P}_{z|F} \left( \left| z^\mathsf{T} A z - \mathrm{Tr}(\widehat{\Sigma}_F A) \right| \geq t \cdot \left\| \widehat{\Sigma}_F^{1/2} A \widehat{\Sigma}_F^{1/2} \right\|_F \right) \leq \mathsf{C}_* \exp \left\{ -t/\mathsf{C}_x \right\}, \tag{59}$$

*and for any deterministic p.s.d. matrix $B \in \mathbb{R}^{\infty \times \infty}$ with $\mathrm{Tr}(\Sigma B) < \infty$,*

$$\mathbb{P}_f \left( \left| f^\mathsf{T} B f - \mathrm{Tr}(\Sigma B) \right| \geq t \cdot \left\| \Sigma^{1/2} B \Sigma^{1/2} \right\|_F \right) \leq \mathsf{C}_* \exp \left\{ -t/\mathsf{C}_x \right\}. \tag{60}$$

We will assume in the rest of this appendix that Assumption B.1 holds. Using the notations introduced above, we restate our setting below. Recall that we consider learning a target function $h_*(g) := g^\mathsf{T} \beta_*$ from i.i.d. samples $(y_i, g_i)_{i \in [n]}$ with

$$y_i = g_i^\mathsf{T} \beta_* + \varepsilon_i,$$

where $\varepsilon_i$ are independent noise with $\mathbb{E}[\varepsilon_i] = 0$ and $\mathbb{E}[\varepsilon_i^2] = \sigma_\varepsilon^2$. Denote $y = (y_1, \ldots, y_n)$ the vector containing the labels. We fit this data using a random feature model with i.i.d. random weight features $(f_j)_{j \in [p]}$

$$\hat{f}(g) = \frac{1}{\sqrt{p}} g^\mathsf{T} F^\mathsf{T} a, \qquad a \in \mathbb{R}^p. \tag{61}$$

We fit the parameter $a$ using random feature ridge regression (RFRR)

$$\hat{a}_\lambda = \arg \min_{a \in \mathbb{R}^p} \left\{ \| y - Za \|_2^2 + \lambda \| a \|_2^2 \right\} = (Z^\mathsf{T} Z + \lambda)^{-1} Z^\mathsf{T} y. \tag{62}$$

The test error is then given by

$$\mathcal{R}_{\text{test}}(h_*; G, F, \lambda) := \mathbb{E}_\varepsilon \left\{ \mathbb{E}_g \left[ \left( g^\mathsf{T} \beta_* - \frac{1}{\sqrt{p}} g^\mathsf{T} F^\mathsf{T} \hat{a}_\lambda \right)^2 \right] \right\}$$
$$= \mathcal{B}(\beta_*; G, F, \lambda) + \mathcal{V}(G, F, \lambda), \tag{63}$$

where the bias and variance terms are given explicitly by

$$\mathcal{B}(\beta_*; G, F, \lambda) = \| \beta_* - p^{-1/2} F^\mathsf{T} (Z^\mathsf{T} Z + \lambda)^{-1} Z^\mathsf{T} G \beta_* \|_2^2, \tag{64}$$

$$\mathcal{V}(G, F, \lambda) = \sigma_\varepsilon^2 \cdot \mathrm{Tr}\left( \widehat{\Sigma}_F Z^\mathsf{T} Z (Z^\mathsf{T} Z + \lambda)^{-2} \right). \tag{65}$$

Note that both the bias and variance terms are random quantities that depend on the random matrices $G, F$. The goal of this appendix is to prove *non-asymptotic* and *multiplicative* approximation guarantees between these two terms and deterministic quantities that only depend on the parameters of the model $(n, p, \Sigma, \beta_*, \lambda, \sigma_\varepsilon^2)$, i.e., we will show that with high probability

$$|\mathcal{B}(\beta_*; G, F, \lambda) - \mathsf{B}_{n,p}(\beta_*, \lambda)| = \widetilde{O}\left( n^{-1/2} + p^{-1/2} \right) \cdot \mathsf{B}_{n,p}(\beta_*, \lambda),$$

$$|\mathcal{V}(G, F, \lambda) - \mathsf{V}_{n,p}(\lambda)| = \widetilde{O}\left( n^{-1/2} + p^{-1/2} \right) \cdot \mathsf{V}_{n,p}(\lambda),$$

where $\mathsf{B}_{n,p}(\beta_*, \lambda)$ and $\mathsf{V}_{n,p}(\lambda)$ are defined in Eqs (22) and (23), and the approximation rates $\widetilde{O}(\cdot)$ are explicit in terms of the model parameters.

The proof of these approximation guarantees will proceed in two steps. We first show that the bias and variance terms conditional on $F$ are well approximated by functionals that only depend on $F$, i.e.,

$$\left| \mathcal{B}(\beta_*; G, F, \lambda) - \widetilde{\mathcal{B}}(\beta_*; F, \lambda) \right| = \widetilde{O}\left( n^{-1/2} \right) \cdot \widetilde{\mathcal{B}}(\beta_*; F, \lambda),$$

$$\left| \mathcal{V}(G, F, \lambda) - \widetilde{\mathcal{V}}(F, \lambda) \right| = \widetilde{O}\left( n^{-1/2} \right) \cdot \widetilde{\mathcal{V}}(F, \lambda). \tag{66}$$

We then show that $\widetilde{\mathcal{B}}(\boldsymbol{\beta}_*; \boldsymbol{F}, \lambda)$ and $\widetilde{\mathcal{V}}(\boldsymbol{F}, \lambda)$ are well approximated by $\mathsf{B}_{n,p}(\boldsymbol{\beta}_*, \lambda)$ and $\mathsf{V}_{n,p}(\lambda)$ with

$$
\begin{aligned}
\left| \widetilde{\mathcal{B}}(\boldsymbol{\beta}_*; \boldsymbol{F}, \lambda) - \mathsf{B}_{n,p}(\boldsymbol{\beta}_*, \lambda) \right| &= \widetilde{O}\left(p^{-1/2}\right) \cdot \mathsf{B}_{n,p}(\boldsymbol{\beta}_*, \lambda), \\
\left| \widetilde{\mathcal{V}}(\boldsymbol{F}, \lambda) - \mathsf{V}_{n,p}(\lambda) \right| &= \widetilde{O}\left(p^{-1/2}\right) \cdot \mathsf{V}_{n,p}(\lambda).
\end{aligned}
\tag{67}
$$

For each of these two steps, we will apply general results showing deterministic equivalents for functionals of (possibly infinite-dimensional) random matrices proved in Misiakiewicz and Saeed [2024] (see Appendix A and in particular Theorem A.2 for some background). Note that when writing the proof, we will directly show Eq. (66) assuming that $\boldsymbol{F}$ is in some good event $\boldsymbol{F} \in \mathcal{A}_{\mathcal{F}}$, where $\mathbb{P}_{\boldsymbol{F}}(\mathcal{A}_{\mathcal{F}}) \geq 1 - p^{-D}$, so that we can immediately write functionals with regularization parameter that does not depend on $\widehat{\boldsymbol{\Sigma}}_{\boldsymbol{F}}$, and therefore the rate will be directly $\widetilde{O}\left(n^{-1/2} + p^{-1/2}\right)$. However, we can reorganize the proof to indeed get the separate contributions (66) and (67) to the approximation error rate.

The rest of Appendix B is devoted to implementing this proof strategy. We start in the next three sections by introducing key technical results which we will use in the analysis of the bias and variance terms.

## B.2    Fixed points, feature covariance matrix, and tail rank

Recall that our deterministic equivalents will depend on the fixed points $(\nu_1, \nu_2) \in \mathbb{R}_{>0}^2$ stated in Definition 1. Furthermore, our approximation guarantees will depend on the covariance matrix $\boldsymbol{\Sigma}$ through $\rho_\kappa(p)$ and $\widetilde{\rho}_\kappa(n, p)$ which we restate below for convenience

$$
M_{\boldsymbol{\Sigma}}(k) = 1 + \frac{r_{\boldsymbol{\Sigma}}(\lfloor \eta_* \cdot k \rfloor) \vee k}{k} \log\left(r_{\boldsymbol{\Sigma}}(\lfloor \eta_* \cdot k \rfloor) \vee k\right),
\tag{68}
$$

$$
\rho_\kappa(p) = 1 + \frac{p \cdot \xi_{\lfloor \eta_* \cdot p \rfloor}^2}{\kappa} M_{\boldsymbol{\Sigma}}(p),
\tag{69}
$$

$$
\widetilde{\rho}_\kappa(n, p) = 1 + \mathbb{1}[n \leq p/\eta_*] \cdot \left\{ \frac{n\xi_{\lfloor \eta_* \cdot n \rfloor}^2}{\kappa} + \frac{n}{p} \cdot \rho_\kappa(p) \right\} M_{\boldsymbol{\Sigma}}(n),
\tag{70}
$$

where $\eta_* \in (0, 1/4)$ is a constant that only depends on $\mathsf{C}_*$ appearing in Assumption B.1, and $r_{\boldsymbol{\Sigma}}(k)$ is the intrinsic dimension of $\boldsymbol{\Sigma}$ at level $k$ (see Definition 3)

$$
r_{\boldsymbol{\Sigma}}(k) = \frac{\sum_{j=k}^p \xi_j^2}{\xi_k^2}.
$$

Observe that $\widetilde{\rho}_\kappa(n, p)$ is well defined for $n \to \infty$ while $p$ stays constant with $\widetilde{\rho}_\kappa(\infty, p) = 1$, and for $p \to \infty$ while $n$ stays constant with $\widetilde{\rho}_\kappa(n, \infty) = \rho_\kappa(n)$ (under the conditions in our setting that $\rho_\kappa(p) \leq K\sqrt{p}$ for some constant $K$).

In this section, we introduce and prove properties on the fixed point $(\nu_1, \nu_2) \in \mathbb{R}_{>0}^2$ and the weight feature matrix $\boldsymbol{F}$ which we will use to prove deterministic equivalents.

**Feature covariance matrix.**    The features $\boldsymbol{z}_i \in \mathbb{R}^p$ conditional on $\boldsymbol{F}$ are i.i.d. random vectors with covariance $\widehat{\boldsymbol{\Sigma}}_{\boldsymbol{F}} = \boldsymbol{F}\boldsymbol{F}^\mathsf{T}/p$. We first show that with high probability over $\boldsymbol{F}$, this feature covariance matrix has eigenvalues and intrinsic dimensions bounded by the ones of $\boldsymbol{\Sigma}$.

Denote $(\hat{\xi}_1^2, \hat{\xi}_2^2, \ldots \hat{\xi}_p^2)$ the $p$ eigenvalues of $\widehat{\boldsymbol{\Sigma}}_{\boldsymbol{F}}$ in nonincreasing order. Applying the definition of the intrinsic dimension at level $k$ to $\widehat{\boldsymbol{\Sigma}}_{\boldsymbol{F}}$, we have for any $k = 1, \ldots, p$,

$$
r_{\widehat{\boldsymbol{\Sigma}}_{\boldsymbol{F}}}(k) = \frac{\sum_{j=k}^p \hat{\xi}_j^2}{\hat{\xi}_k^2}.
\tag{71}
$$

Applying Theorem A.2 directly to functionals of $\boldsymbol{Z}$ conditional on $\boldsymbol{F}$, the approximation guarantees depend on

$$
\widehat{\rho}_\lambda(n) = 1 + \frac{n \cdot \hat{\xi}_{\lfloor \eta_* \cdot n \rfloor}^2}{\lambda} \left\{ 1 + \frac{r_{\widehat{\boldsymbol{\Sigma}}_{\boldsymbol{F}}}(\lfloor \eta_* \cdot n \rfloor) \vee n}{n} \log\left(r_{\widehat{\boldsymbol{\Sigma}}_{\boldsymbol{F}}}(\lfloor \eta_* \cdot n \rfloor) \vee n\right) \right\},
\tag{72}
$$

which simply corresponds to $\rho_\lambda$ defined in Eq. (52) applied to $\widehat{\Sigma}_{\boldsymbol{F}}$ where we recall that $\hat{\xi}^2_{\lfloor \eta_* \cdot n \rfloor} = 0$ if $\lfloor \eta_* \cdot n \rfloor > p$. The next lemma shows that with high probability over $\boldsymbol{F}$, we have $\widehat{\rho}_\lambda(n) \lesssim \widetilde{\rho}_\lambda(n,p)$ for all $n \in \mathbb{N}$.

**Lemma B.2** (Feature covariance matrix). *Assume the feature vectors $\{\boldsymbol{f}_j\}_{j \in [p]}$ satisfy Assumption B.1. Then for any $D, K > 0$, there exist constants $\eta_* \in (0, 1/4)$ and $C_{*,D,K} > 0$ such that the following holds. For any $p \geq C_{*,D,K}$, the event*

$$\widetilde{\mathcal{A}}_{\mathcal{F}} = \left\{ \boldsymbol{F} \in \mathbb{R}^{p \times \infty} \; : \; \|\widehat{\Sigma}_{\boldsymbol{F}}\|_{\mathrm{op}} \geq \frac{1}{2}, \quad \widehat{\rho}_\lambda(n) \leq C_{*,D,K} \cdot \widetilde{\rho}_\lambda(n,p), \;\; \forall n \in \mathbb{N}, \lambda \in \mathbb{R} \right\} \quad (73)$$

*holds with probability at least $1 - p^{-D}$.*

We defer the proof of this lemma to Section B.7.1.

**High-degree part of the feature matrix $\boldsymbol{F}$.** Recall that $(\nu_1, \nu_2) \in \mathbb{R}^2_{>0}$ are the solutions to the fixed point equations stated in Definition 1. In order to get approximation guarantees when $p\nu_1 \to 0$ as $n \to \infty$, we will analyze separately the top eigenspaces of $\boldsymbol{F}$ from the rest. For an integer $\mathsf{m} \in \mathbb{N}$, we split the feature vector $\boldsymbol{f}_j = [\boldsymbol{f}_{0,j}, \boldsymbol{f}_{+,j}]$ where $\boldsymbol{f}_{0,j} \in \mathbb{R}^{\mathsf{m}}$ corresponds to the top $\mathsf{m}$ coordinates with covariance

$$\Sigma_0 := \mathbb{E}[\boldsymbol{f}_{0,j}\boldsymbol{f}_{0,j}^\mathsf{T}] = \mathrm{diag}(\xi_1^2, \xi_2^2, \ldots, \xi_{\mathsf{m}}^2),$$

and $\boldsymbol{f}_{+,j} \in \mathbb{R}^\infty$ corresponds to the high degree features orthogonal to the top $\mathsf{m}$ eigenspaces. We denote their covariance

$$\Sigma_+ := \mathbb{E}[\boldsymbol{f}_{+,j}\boldsymbol{f}_{+,j}^\mathsf{T}] = \mathrm{diag}(\xi_{\mathsf{m}+1}^2, \xi_{\mathsf{m}+2}^2, \ldots).$$

We split the weight feature matrix intro $\boldsymbol{F} = [\boldsymbol{F}_0, \boldsymbol{F}_+]$, where

$$\boldsymbol{F}_0 = [\boldsymbol{f}_{0,1}, \ldots, \boldsymbol{f}_{0,p}]^\mathsf{T} \in \mathbb{R}^{p \times \mathsf{m}}, \qquad \boldsymbol{F}_+ = [\boldsymbol{f}_{+,1}, \ldots, \boldsymbol{f}_{+,p}]^\mathsf{T} \in \mathbb{R}^{p \times \infty}.$$

We will use that for $\mathsf{m}$ chosen such that $p \cdot \xi_{\mathsf{m}+1} \ll \mathrm{Tr}(\Sigma_+)$, we have with high probability

$$\|\boldsymbol{F}_+\boldsymbol{F}_+^\mathsf{T} - \mathrm{Tr}(\Sigma_+) \cdot \mathbf{I}_p\|_{\mathrm{op}} \lesssim \sqrt{p \cdot \xi_{\mathsf{m}+1}\mathrm{Tr}(\Sigma_+)} \ll \mathrm{Tr}(\Sigma_+),$$

and therefore

$$\boldsymbol{F}\boldsymbol{F}^\mathsf{T} + \kappa \approx \boldsymbol{F}_0\boldsymbol{F}_0^\mathsf{T} + \gamma(\kappa),$$

where we defined the function

$$\gamma(\kappa) := \kappa + \mathrm{Tr}(\Sigma_+).$$

To simplify the final statement of our results, we assume that we can choose $\mathsf{m}$ such that $p^2\xi_{\mathsf{m}+1}^2 \leq \gamma(p\lambda/n)$. Note that $\nu_1 \geq \lambda/n$ from the fixed point equations and $\gamma(p\lambda/n) \leq \gamma(p\nu_1)$ (e.g., see Equation (76)). For convenience, we will further denote

$$\gamma_+ := \gamma(p\nu_1), \qquad \gamma_\lambda := \gamma(p\lambda/n). \quad (74)$$

**Lemma B.3** (Concentration of high-degree part of $\boldsymbol{F}$). *Assume that $(\boldsymbol{f}_{+,j})_{j \in [p]}$ satisfy Assumption B.1 and $\mathsf{m} \in \mathbb{N}$ is chosen such that $p^2\xi_{\mathsf{m}+1}^2 \leq \gamma(p\lambda/n)$. Then for any $D > 0$, there exists a constant $C_{*,D} > 0$ such that with probability at least $1 - p^{-D}$,*

$$\|\boldsymbol{F}_+\boldsymbol{F}_+^\mathsf{T} - \mathrm{Tr}(\Sigma_+) \cdot \mathbf{I}_p\|_{\mathrm{op}} \leq C_{*,D}\frac{\log^3(p)}{\sqrt{p}}\gamma_\lambda \leq C_{*,D}\frac{\log^3(p)}{\sqrt{p}}\gamma_+.$$

This lemma follows directly from [Misiakiewicz and Saeed, 2024, Proposition 9].

**Effective regularization and fixed points.** Conditional on $\boldsymbol{F}$, the bias and variance are functionals of the random matrix $\boldsymbol{Z}$ which has $n$ i.i.d. rows with regularization parameter $\lambda$ and covariance $\widehat{\Sigma}_{\boldsymbol{F}}$. The deterministic approximations to these functionals depend on an "effective regularization" $\tilde{\nu}_1$ associated to $(n, \widehat{\Sigma}_{\boldsymbol{F}}, \lambda)$ (see Definition 2 in Appendix A for background) which is given as the unique non-negative solution to the equation

$$n - \frac{\lambda}{\tilde{\nu}_1} = \mathrm{Tr}\big(\widehat{\Sigma}_{\boldsymbol{F}}\big(\widehat{\Sigma}_{\boldsymbol{F}} + \tilde{\nu}_1\big)^{-1}\big).$$

Note that the right-hand side can be rewritten as

$$\text{Tr}\big(\widehat{\boldsymbol{\Sigma}}_{\boldsymbol{F}}\big(\widehat{\boldsymbol{\Sigma}}_{\boldsymbol{F}} + \tilde{\nu}_1\big)^{-1}\big) = \text{Tr}\big(\boldsymbol{F}\boldsymbol{F}^{\mathsf{T}}(\boldsymbol{F}\boldsymbol{F}^{\mathsf{T}} + p\tilde{\nu}_1)^{-1}\big), \tag{75}$$

which is itself of functional of the random matrix $\boldsymbol{F}$ which has $p$ i.i.d. rows with regularization parameter $p\tilde{\nu}_1$ and covariance $\boldsymbol{\Sigma}$. Note that $\tilde{\nu}_1$ is a random variable depending itself on $\boldsymbol{F}$. However we will show that it concentrates on a deterministic value $\nu_1$. We therefore introduce a second effective regularization $\nu_2$ associated to $(p, \boldsymbol{\Sigma}, p\nu_1)$ given as the unique non-negatice solution of the equation

$$p - \frac{p\nu_1}{\nu_2} = \text{Tr}\left(\boldsymbol{\Sigma}(\boldsymbol{\Sigma} + \nu_2)^{-1}\right).$$

The functional (75) is then well approximated by

$$\text{Tr}\big(\boldsymbol{F}\boldsymbol{F}^{\mathsf{T}}(\boldsymbol{F}\boldsymbol{F}^{\mathsf{T}} + p\tilde{\nu}_1)^{-1}\big) = (1 + o_{p,\mathbb{P}}(1)) \cdot \text{Tr}\big(\boldsymbol{\Sigma}(\boldsymbol{\Sigma} + \nu_2)^{-1}\big).$$

This motivates to define $(\nu_1, \nu_2) \in \mathbb{R}^2_{>0}$ as the unique non-negative solutions to the coupled fixed point equations

$$\begin{aligned}
n - \frac{\lambda}{\nu_1} &= \text{Tr}\left(\boldsymbol{\Sigma}(\boldsymbol{\Sigma} + \nu_2)^{-1}\right), \\
p - \frac{p\nu_1}{\nu_2} &= \text{Tr}\left(\boldsymbol{\Sigma}(\boldsymbol{\Sigma} + \nu_2)^{-1}\right).
\end{aligned} \tag{76}$$

Writing $\nu_1$ as a function of $\nu_2$ indeed produces the equations stated in Definition 1.

To show that $\tilde{\nu}_1$ concentrates on $\nu_1$, we define the following fixed points $(\tilde{\nu}_1, \tilde{\nu}_2) \in \mathbb{R}^2_{>0}$ to be the unique positive solutions to the random equations

$$\begin{aligned}
n - \frac{\lambda}{\tilde{\nu}_1} &= \text{Tr}\big(\widehat{\boldsymbol{\Sigma}}_{\boldsymbol{F}}\big(\widehat{\boldsymbol{\Sigma}}_{\boldsymbol{F}} + \tilde{\nu}_1\big)^{-1}\big), \\
p - \frac{p\tilde{\nu}_1}{\tilde{\nu}_2} &= \text{Tr}\left(\boldsymbol{\Sigma}(\boldsymbol{\Sigma} + \tilde{\nu}_2)^{-1}\right).
\end{aligned} \tag{77}$$

The following proposition shows that $(\tilde{\nu}_1, \tilde{\nu}_2)$ is well approximated by $(\nu_1, \nu_2)$ with high probability.

**Proposition B.4** (Concentration of the fixed points)**.** *Assume that $(\boldsymbol{f}_j)_{j \in [p]}$ satisfy Assumption B.1. Then for any $D, K > 0$, there exist constants $\eta_* \in (0, 1/4)$ and $C_{*,D,K} > 0$ such that the following holds. Let $\rho_\kappa(p)$ and $\widetilde{\rho}_\kappa(n, p)$ be defined as per Eqs. (26) and (25), and $\gamma_\lambda$ and $\gamma_+$ as per Eq. (74). For any $p \geq C_{*,D,K}$ and $\lambda > 0$, if it holds that*

$$\gamma_\lambda \geq p^{-K}, \qquad \widetilde{\rho}_\lambda(n, p) \cdot \rho_{\gamma_\lambda}(p)^{5/2} \log^4(p) \leq K\sqrt{p}, \tag{78}$$

*then with probability at least $1 - p^{-D}$, we have*

$$\max\left\{ \frac{|\tilde{\nu}_2 - \nu_2|}{\nu_2}, \frac{|\tilde{\nu}_1 - \nu_1|}{\nu_1} \right\} \leq C_{*,D,K} \cdot \mathcal{E}_\nu(p),$$

*where we defined*

$$\mathcal{E}_\nu(p) := \frac{\widetilde{\rho}_\lambda(n, p) \cdot \rho_{\gamma_+}(p)^{5/2} \log^3(p)}{\sqrt{p}}.$$

The proof of this proposition can be found in Section B.7.2.

To study functionals of $\boldsymbol{Z}$ conditional on $\boldsymbol{F}$, we will assume that $\boldsymbol{F}$ belongs to the good event

$$\mathcal{A}_{\mathcal{F}} := \widetilde{\mathcal{A}}_{\mathcal{F}} \cap \left\{ \boldsymbol{F} \in \mathbb{R}^{p \times \infty} \ : \ |\tilde{\nu}_1 - \nu_1| \leq C_{*,D,K} \cdot \mathcal{E}_\nu(p) \cdot \nu_1 \right\}, \tag{79}$$

where $\widetilde{\mathcal{A}}_{\mathcal{F}}$ is defined in Lemma B.2. In particular, as long as $p \geq C_{*,D,K}$ and condition (78) hold, then $\mathbb{P}(\mathcal{A}_{\mathcal{F}}) \geq 1 - 2p^{-D}$ by Lemma B.2 and Proposition B.4.

**Truncated fixed point.** As mentioned above, we will separate the analysis of the low-degree part of the feature matrix $\boldsymbol{F}_0$ and the high-degree part $\boldsymbol{F}_+$. For the high-degree part, we will simply use the concentration stated in Lemma B.3. For the low degree part, we will study functional of $\boldsymbol{F}_0$ with regularization $\gamma_+ = p\nu_1 + \mathrm{Tr}(\boldsymbol{\Sigma}_+)$. We therefore introduce an effective regularization $\nu_{2,0}$ associated to the model $(p, \boldsymbol{\Sigma}_0, \gamma_+)$, i.e., the unique positive solution to the equation

$$p - \frac{\gamma_+}{\nu_{2,0}} = \mathrm{Tr}\big(\boldsymbol{\Sigma}_0(\boldsymbol{\Sigma}_0 + \nu_{2,0})^{-1}\big). \tag{80}$$

Intuitively, $\nu_{2,0}$ will be closed to $\nu_2$ as soon as $\lambda_{\max}(\boldsymbol{\Sigma}_+) \ll \nu_2$ as

$$n - \frac{p\nu_1}{\nu_2} \approx \mathrm{Tr}\big(\boldsymbol{\Sigma}_0(\boldsymbol{\Sigma}_0 + \nu_2)^{-1}\big) + \frac{\mathrm{Tr}(\boldsymbol{\Sigma}_+)}{\nu_2},$$

and uniqueness of the positive solution $\nu_{2,0}$. This is formalized in the following lemma.

**Lemma B.5** (Truncated fixed point). *Let* $\mathsf{m}$ *be chosen such that* $p^2\xi_{\mathsf{m}+1}^2 \le \gamma_\lambda$, *then*

$$\frac{|\nu_{2,0} - \nu_2|}{\nu_2} \le \frac{1}{p}. \tag{81}$$

*Furthermore, there exists an absolute constant* $C > 0$ *such that*

$$\left|\mathrm{Tr}\big(\boldsymbol{\Sigma}_0(\boldsymbol{\Sigma}_0 + \nu_{2,0})^{-1}\big) + \frac{\mathrm{Tr}(\boldsymbol{\Sigma}_+)}{\nu_{2,0}} - \mathrm{Tr}\big(\boldsymbol{\Sigma}(\boldsymbol{\Sigma} + \nu_2)^{-1}\big)\right| \le \frac{C}{p}\mathrm{Tr}\big(\boldsymbol{\Sigma}(\boldsymbol{\Sigma} + \nu_2)^{-1}\big). \tag{82}$$

*Proof of Lemma B.5.* The first bound (81) follows directly from [Misiakiewicz and Saeed, 2024, Lemma 6]. For the second inequality, we decompose this difference into

$$\left|\mathrm{Tr}\big(\boldsymbol{\Sigma}_0(\boldsymbol{\Sigma}_0 + \nu_{2,0})^{-1}\big) + \frac{\mathrm{Tr}(\boldsymbol{\Sigma}_+)}{\nu_{2,0}} - \mathrm{Tr}\big(\boldsymbol{\Sigma}(\boldsymbol{\Sigma} + \nu_2)^{-1}\big)\right|$$

$$\le |\nu_{2,0} - \nu_2|\,\mathrm{Tr}\big(\boldsymbol{\Sigma}_0(\boldsymbol{\Sigma}_0 + \nu_{2,0})^{-1}(\boldsymbol{\Sigma}_0 + \nu_2)^{-1}\big) + \frac{\xi_{\mathsf{m}+1}^2 + |\nu_{2,0} - \nu_2|}{\nu_{2,0}}\mathrm{Tr}\big(\boldsymbol{\Sigma}_+(\boldsymbol{\Sigma}_+ + \nu_2)^{-1}\big)$$

$$\le \left\{\frac{\xi_{\mathsf{m}+1}^2}{\nu_{2,0}} + \frac{|\nu_{2,0} - \nu_2|}{\nu_{2,0}}\right\}\mathrm{Tr}\big(\boldsymbol{\Sigma}(\boldsymbol{\Sigma} + \nu_2)^{-1}\big)$$

$$\le \frac{3}{p}\mathrm{Tr}\big(\boldsymbol{\Sigma}(\boldsymbol{\Sigma} + \nu_2)^{-1}\big),$$

where we used that $\xi_{\mathsf{m}+1}^2/\nu_{2,0} \le \gamma_+/(p^2\nu_{2,0}) \le 1/p$ by the assumption on $\mathsf{m}$ and identity (80). $\square$

## B.3 Deterministic equivalents for functionals of Z conditional on F

As mentioned in Section B.1, we will first show that the test error concentrates over $\boldsymbol{Z}$ conditional on $\boldsymbol{F}$ on some quantity that only depends on $\widehat{\boldsymbol{\Sigma}}_{\boldsymbol{F}}$. The bias and variance terms can be written in terms of the following three functionals of the feature matrix $\boldsymbol{Z}$: for a general p.s.d. matrix $\boldsymbol{A} \in \mathbb{R}^{p \times p}$ and positive scalar $\kappa > 0$, define

$$\Phi_2(\boldsymbol{Z}; \kappa) := \mathrm{Tr}\left(\frac{\boldsymbol{Z}^\mathsf{T}\boldsymbol{Z}}{n}(\boldsymbol{Z}^\mathsf{T}\boldsymbol{Z} + \kappa)^{-1}\right),$$

$$\Phi_3(\boldsymbol{Z}; \boldsymbol{A}, \kappa) := \mathrm{Tr}\left(\boldsymbol{A}\widehat{\boldsymbol{\Sigma}}_{\boldsymbol{F}}^{1/2}(\boldsymbol{Z}^\mathsf{T}\boldsymbol{Z} + \kappa)^{-1}\widehat{\boldsymbol{\Sigma}}_{\boldsymbol{F}}(\boldsymbol{Z}^\mathsf{T}\boldsymbol{Z} + \kappa)^{-1}\widehat{\boldsymbol{\Sigma}}_{\boldsymbol{F}}^{1/2}\right), \tag{83}$$

$$\Phi_4(\boldsymbol{Z}; \boldsymbol{A}, \kappa) := \left(\boldsymbol{A}\widehat{\boldsymbol{\Sigma}}_{\boldsymbol{F}}^{1/2}(\boldsymbol{Z}^\mathsf{T}\boldsymbol{Z} + \kappa)^{-1}\frac{\boldsymbol{Z}^\mathsf{T}\boldsymbol{Z}}{n}(\boldsymbol{Z}^\mathsf{T}\boldsymbol{Z} + \kappa)^{-1}\widehat{\boldsymbol{\Sigma}}_{\boldsymbol{F}}^{1/2}\right),$$

where we recall that $\boldsymbol{Z}$ has i.i.d. rows with covariance $\widehat{\boldsymbol{\Sigma}}_{\boldsymbol{F}} = \boldsymbol{F}\boldsymbol{F}^\mathsf{T}/p$. We show that these functional are well approximated by functionals of $\boldsymbol{F}$ that can be written in terms of the following functionals:

$$\widetilde{\Phi}_2(\boldsymbol{F}; \kappa) := \mathrm{Tr}\left(\frac{\boldsymbol{F}^\mathsf{T}\boldsymbol{F}}{p}(\boldsymbol{F}^\mathsf{T}\boldsymbol{F} + \kappa)^{-1}\right),$$

$$\widetilde{\Phi}_5(\boldsymbol{F}; \boldsymbol{A}, \kappa) := \frac{1}{n} \cdot \frac{\widetilde{\Phi}_6(\boldsymbol{F}; \boldsymbol{A}, \kappa)}{n - \widetilde{\Phi}_6(\boldsymbol{F}; \boldsymbol{I}, \kappa)}, \tag{84}$$

$$\widetilde{\Phi}_6(\boldsymbol{F}; \boldsymbol{A}, \kappa) := \mathrm{Tr}\left(\boldsymbol{A}(\boldsymbol{F}\boldsymbol{F}^\mathsf{T})^2(\boldsymbol{F}\boldsymbol{F}^\mathsf{T} + \kappa)^{-2}\right).$$

The following proposition gather the approximation guarantees for $\Phi_2, \Phi_3, \Phi_4$ listed in Eq. (83).

**Proposition B.6** (Deterministic equivalents for $\Phi(\boldsymbol{Z})$ conditional on $\boldsymbol{F}$). *Under Assumption B.1 and assuming that $\boldsymbol{F} \in \mathcal{A}_{\mathcal{F}}$ defined in Eq. (79), for any $D, K > 0$, there exist constants $\eta_* \in (0, 1/4)$, $C_{D,K} > 0$, and $C_{*,D,K} > 0$ such that the followings holds. Let $\rho_\kappa(p)$ and $\widetilde{\rho}_\kappa(n, p)$ be defined as per Eqs. (26) and (25). For any $n \geq C_{D,K}$ and regularization parameter $\lambda > 0$, if it holds that*

$$\lambda \geq n^{-K}, \qquad \widetilde{\rho}_\lambda(n, p)^{5/2} \log^{3/2}(n) \leq K\sqrt{n}, \tag{85}$$
$$\widetilde{\rho}_\lambda(n, p)^2 \cdot \rho_{\gamma_+}(p)^{5/2} \log^3(p) \leq K\sqrt{p},$$

*then for any p.s.d. matrix $\boldsymbol{A} \in \mathbb{R}^{p \times p}$ (independent of $\boldsymbol{Z}|\boldsymbol{F}$), with probability at least $1 - n^{-D}$ on $\boldsymbol{Z}$ conditional on $\boldsymbol{F}$, we have*

$$\left| \Phi_2(\boldsymbol{Z}; \lambda) - \frac{p}{n} \widetilde{\Phi}_2(\boldsymbol{F}; p\nu_1) \right| \leq C_{*,D,K} \cdot \mathcal{E}_1(n, p) \cdot \frac{p}{n} \widetilde{\Phi}_2(\boldsymbol{F}; p\nu_1), \tag{86}$$

$$\left| \Phi_3(\boldsymbol{Z}; \boldsymbol{A}, \lambda) - \left(\frac{n\nu_1}{\lambda}\right)^2 \widetilde{\Phi}_5(\boldsymbol{F}; \boldsymbol{A}, p\nu_1) \right| \leq C_{*,D,K} \cdot \mathcal{E}_1(n, p) \cdot \left(\frac{n\nu_1}{\lambda}\right)^2 \widetilde{\Phi}_5(\boldsymbol{F}; \boldsymbol{A}, p\nu_1), \tag{87}$$

$$\left| \Phi_4(\boldsymbol{Z}; \boldsymbol{A}, \lambda) - \widetilde{\Phi}_5(\boldsymbol{F}; \boldsymbol{A}, p\nu_1) \right| \leq C_{*,D,K} \cdot \mathcal{E}_1(n, p) \cdot \widetilde{\Phi}_5(\boldsymbol{F}; \boldsymbol{A}, p\nu_1), \tag{88}$$

*where the approximation rate is given by*

$$\mathcal{E}_1(n, p) := \frac{\widetilde{\rho}_\lambda(n, p)^6 \log^{5/2}(n)}{\sqrt{n}} + \frac{\widetilde{\rho}_\lambda(n, p)^2 \cdot \rho_{\gamma_+}(p)^{5/2} \log^3(p)}{\sqrt{p}}. \tag{89}$$

Proposition B.6 is a consequence of [Misiakiewicz and Saeed, 2024, Theorem 4] (see Appendix A and Theorem A.2 for background) and Proposition B.4. We defer its proof to Section B.7.3. Note that the term $\widetilde{O}(p^{-1/2})$ in the approximation rate $\mathcal{E}_1(n, p)$ defined in Eq. (89) comes from comparing $p\nu_1$ with $p\tilde{\nu}_1$ and is equal to $\widetilde{\rho}_\lambda(n, p) \cdot \mathcal{E}_\nu(n, p)$ where $\mathcal{E}_\nu(n, p)$ is defined in Proposition B.4. If instead, we compared to functionals with regularization $p\tilde{\nu}_1$, then the approximation rate in Proposition B.6 would scale $\widetilde{O}(n^{-1/2})$ as expected.

In the analysis of the bias term, we will further need to show deterministic equivalents in the case where $\boldsymbol{A}$ is itself a random matrix uncorrelated (but not independent) to $\boldsymbol{Z}|\boldsymbol{F}$. The following proposition gather these approximation guarantees and is a consequence of [Misiakiewicz and Saeed, 2024, Lemma 10] and Proposition B.4.

**Proposition B.7** (Deterministic equivalents for $\Phi(\boldsymbol{Z})$, uncorrelated numerator). *Assume the same setting as Proposition B.6 and the same conditions (85). Consider a deterministic vector $\boldsymbol{v} \in \mathbb{R}^p$ and a random vector $\boldsymbol{u} = (u_i)_{i \in [n]}$ with i.i.d. entries and $\mathbb{E}[u_i] = 0$, $\mathbb{E}[u_i^2] = 1$, and $\mathbb{E}[\boldsymbol{z}_i u_i | \boldsymbol{F}] = \boldsymbol{0}$. Then with probability at least $1 - n^{-D}$ on $\boldsymbol{Z}$ conditional on $\boldsymbol{F}$, we have*

$$\left| \langle \boldsymbol{u}, \boldsymbol{Z}(\boldsymbol{Z}^\mathsf{T}\boldsymbol{Z} + \lambda)^{-1} \widehat{\boldsymbol{\Sigma}}_{\boldsymbol{F}} (\boldsymbol{Z}^\mathsf{T}\boldsymbol{Z} + \lambda)^{-1} \boldsymbol{Z}^\mathsf{T} \boldsymbol{u} \rangle - n \widetilde{\Phi}_5(\boldsymbol{F}; \mathbf{I}, p\nu_1) \right| \leq C_{*,D,K} \cdot \mathcal{E}_2(n, p), \tag{90}$$

$$\left| \langle \boldsymbol{u}, \boldsymbol{Z}(\boldsymbol{Z}^\mathsf{T}\boldsymbol{Z} + \lambda)^{-1} \widehat{\boldsymbol{\Sigma}}_{\boldsymbol{F}} (\boldsymbol{Z}^\mathsf{T}\boldsymbol{Z} + \lambda)^{-1} \boldsymbol{v} \rangle \right| \leq C_{*,D,K} \cdot \mathcal{E}_2(n, p) \cdot \frac{n\nu_1}{\lambda} \sqrt{\widetilde{\Phi}_5(\boldsymbol{F}; \overline{\boldsymbol{v}}\overline{\boldsymbol{v}}^\mathsf{T}, p\nu_1)}, \tag{91}$$

*where we denoted $\overline{\boldsymbol{v}} := \widehat{\boldsymbol{\Sigma}}_{\boldsymbol{F}}^{-1/2} \boldsymbol{v}$ and the approximation rate is given by*

$$\mathcal{E}_2(n, p) := \frac{\widetilde{\rho}_\lambda(n, p)^6 \log^{7/2}(n)}{\sqrt{n}} + \frac{\widetilde{\rho}_\lambda(n, p)^2 \cdot \rho_{\gamma_+}(p)^{5/2} \log^3(p)}{\sqrt{p}}. \tag{92}$$

We defer the proof of Proposition B.7 to Section B.7.3.

## B.4 Deterministic equivalents for functionals of F

After replacing the bias and variance terms by their deterministic equivalents over the randomness in $\boldsymbol{Z}|\boldsymbol{F}$, we obtain functionals in terms of $\widetilde{\Phi}_2(\boldsymbol{F}; p\nu_1)$ and $\widetilde{\Phi}_5(\boldsymbol{F}; \boldsymbol{A}, p\nu_1)$ listed in Eq. (84). As mentioned in Section B.2, we will analyze the low-degree and high degree part of the feature matrix separately. Using Lemma B.3, we can replace the high-degree part $\boldsymbol{F}_+ \boldsymbol{F}_+^\mathsf{T}$ by a deterministic matrix, which results in a regularization parameter $\gamma(p\nu_1) = p\nu_1 + \text{Tr}(\boldsymbol{\Sigma}_+)$.

The functionals of $\boldsymbol{F}_0$ can be written in terms of the following quantities: for any deterministic matrix $\boldsymbol{B} \in \mathbb{R}^{\mathsf{m}\times\mathsf{m}}$, define

$$\widetilde{\Phi}_1(\boldsymbol{F}_0; \boldsymbol{B}, \kappa) = \operatorname{Tr}\left(\boldsymbol{B}\boldsymbol{\Sigma}_0^{1/2}(\boldsymbol{F}_0^\mathsf{T}\boldsymbol{F}_0 + \gamma(\kappa))^{-1}\boldsymbol{\Sigma}_0^{1/2}\right),$$

$$\widetilde{\Phi}_2(\boldsymbol{F}_0; \kappa) = \operatorname{Tr}\left(\frac{\boldsymbol{F}_0^\mathsf{T}\boldsymbol{F}_0}{p}(\boldsymbol{F}_0^\mathsf{T}\boldsymbol{F}_0 + \gamma(\kappa))^{-1}\right),$$

$$\widetilde{\Phi}_3(\boldsymbol{F}_0; \boldsymbol{B}, \kappa) = \operatorname{Tr}\left(\boldsymbol{B}_0\boldsymbol{\Sigma}_0^{1/2}(\boldsymbol{F}_0^\mathsf{T}\boldsymbol{F}_0 + \gamma(\kappa))^{-1}\boldsymbol{\Sigma}_0(\boldsymbol{F}_0^\mathsf{T}\boldsymbol{F}_0 + \gamma(\kappa))^{-1}\boldsymbol{\Sigma}_0^{1/2}\right),$$

$$\widetilde{\Phi}_4(\boldsymbol{F}_0; \boldsymbol{B}, \kappa) = \left(\boldsymbol{B}\boldsymbol{\Sigma}_0^{1/2}(\boldsymbol{F}_0^\mathsf{T}\boldsymbol{F}_0 + \gamma(\kappa))^{-1}\frac{\boldsymbol{F}_0^\mathsf{T}\boldsymbol{F}_0}{p}(\boldsymbol{F}_0^\mathsf{T}\boldsymbol{F}_0 + \gamma(\kappa))^{-1}\boldsymbol{\Sigma}^{1/2}\right). \tag{93}$$

We show that these functionals can be well approximated by the deterministic functions that can be written in terms of

$$\Psi_1(\nu; \boldsymbol{B}) = \operatorname{Tr}\left(\boldsymbol{B}\boldsymbol{\Sigma}_0(\boldsymbol{\Sigma}_0 + \nu)^{-1}\right),$$

$$\Psi_2(\nu) = \frac{1}{p}\operatorname{Tr}\left(\boldsymbol{\Sigma}_0(\boldsymbol{\Sigma}_0 + \nu)^{-1}\right),$$

$$\Psi_3(\nu; \boldsymbol{B}) = \frac{1}{p}\cdot\frac{\operatorname{Tr}(\boldsymbol{B}\boldsymbol{\Sigma}_0^2(\boldsymbol{\Sigma}_0 + \nu)^{-2})}{p - \operatorname{Tr}(\boldsymbol{\Sigma}_0^2(\boldsymbol{\Sigma}_0 + \nu)^{-2})}. \tag{94}$$

Recall that for $\kappa = p\nu_1$, we denote $\gamma_+ := \gamma(p\nu_1)$ and $\nu_{2,0}$ the effective regularization associated to model $(p, \boldsymbol{\Sigma}_0, \gamma_+)$. The following proposition gather the approximation guarantees for $\widetilde{\Phi}_1, \widetilde{\Phi}_2, \widetilde{\Phi}_3, \widetilde{\Phi}_4$ listed in Eq. (93).

**Proposition B.8** (Deterministic equivalents for $\boldsymbol{F}_0$). *Under Assumption B.1, for any $D, K > 0$, there exist constants $\eta_* \in (0, 1/4)$, $C_{D,K} > 0$, and $C_{*,D,K} > 0$ such that the followings holds. Let $\rho_\kappa(p)$ be defined as per Eq. (26). For any $p \geq C_{D,K}$ and $\lambda > 0$, if it holds that*

$$\gamma_+ \geq p^{-K}, \qquad \rho_{\gamma_+}(p)^{5/2}\log^{3/2}(p) \leq K\sqrt{p}, \tag{95}$$

*then for any deterministic p.s.d. matrix $\boldsymbol{B} \in \mathbb{R}^{\mathsf{m}\times\mathsf{m}}$, with probability at least $1 - p^{-D}$, we have*

$$\left|\widetilde{\Phi}_1(\boldsymbol{F}_0; \boldsymbol{B}, p\nu_1) - \frac{\nu_{2,0}}{\gamma_+}\Psi_1(\nu_{2,0}; \boldsymbol{B})\right| \leq C_{*,D,K}\cdot\mathcal{E}_3(p)\cdot\frac{\nu_{2,0}}{\gamma_+}\Psi_1(\nu_{2,0}; \boldsymbol{B}), \tag{96}$$

$$\left|\widetilde{\Phi}_2(\boldsymbol{F}_0; p\nu_1) - \Psi_2(\nu_{2,0})\right| \leq C_{*,D,K}\cdot\mathcal{E}_3(p)\cdot\Psi_2(\nu_{2,0}), \tag{97}$$

$$\left|\widetilde{\Phi}_3(\boldsymbol{F}_0; \boldsymbol{B}, p\nu_1) - \left(\frac{p\nu_{2,0}}{\gamma_+}\right)^2\Psi_3(\nu_{2,0}; \boldsymbol{B})\right| \leq C_{*,D,K}\cdot\mathcal{E}_3(p)\cdot\left(\frac{p\nu_{2,0}}{\gamma_+}\right)^2\Psi_3(\nu_{2,0}; \boldsymbol{B}), \tag{98}$$

$$\left|\widetilde{\Phi}_4(\boldsymbol{F}_0; \boldsymbol{B}, p\nu_1) - \Psi_3(\nu_{2,0}; \boldsymbol{B})\right| \leq C_{*,D,K}\cdot\mathcal{E}_3(p)\cdot\Psi_3(\nu_{2,0}; \boldsymbol{B}), \tag{99}$$

*where the approximation rate is given by*

$$\mathcal{E}_3(p) := \frac{\rho_{\gamma_+}(p)^6\log^3(p)}{\sqrt{p}}. \tag{100}$$

This proposition is obtained by directly applying Theorem A.2 with no modifications.

Again, in the analysis of the bias term, because we separated the analysis of the low-degree and high-degree parts $\boldsymbol{F}_0$ and $\boldsymbol{F}_+$, we will further need deterministic equivalents when $\boldsymbol{B}$ is itself a random matrix uncorrelated but not independent to $\boldsymbol{F}$. We gather the associated deterministic equivalents in the following proposition.

**Proposition B.9** (Deterministic equivalents for $\boldsymbol{F}_0$, uncorrelated numerator). *Assume the same setting as Proposition B.8 and the same conditions (95). Consider a deterministic vector $\boldsymbol{v} \in \mathbb{R}^{\mathsf{m}}$ and a random vector $\boldsymbol{u} = (u_j)_{j\in[p]}$ with i.i.d. entries and $\mathbb{E}[u_j] = 0$, $\mathbb{E}[u_j^2] = 1$, and $\mathbb{E}[\boldsymbol{f}_{0,j}u_j] = \boldsymbol{0}$.*

*Then with probability at least $1 - p^{-D}$, we have*

$$\left| \frac{1}{p} \langle \boldsymbol{u}, (\boldsymbol{F}_0 \boldsymbol{F}_0^\mathsf{T} + \gamma_+)^{-2} \boldsymbol{u} \rangle - \frac{1}{(p\nu_{2,0})^2} \frac{1}{1 - \frac{1}{p}\mathrm{Tr}(\boldsymbol{\Sigma}_0^2 (\boldsymbol{\Sigma}_0 + \nu_{2,0})^{-2})} \right| \le C_{*,D,K} \cdot \mathcal{E}_4(p) \cdot \frac{1}{\gamma_+^2}, \tag{101}$$

$$\left| \frac{1}{p} \langle \boldsymbol{u}, (\boldsymbol{F}_0 \boldsymbol{F}_0^\mathsf{T} + \gamma_+)^{-2} \boldsymbol{F}_0 \boldsymbol{v} \rangle \right| \le C_{*,D,K} \cdot \mathcal{E}_4(p) \frac{p\nu_{2,0}}{\gamma_+^2} \sqrt{\Psi_3(\nu_{2,0}; \overline{\boldsymbol{v}\boldsymbol{v}^\mathsf{T}})}, \tag{102}$$

*where we denoted $\overline{\boldsymbol{v}} := \boldsymbol{\Sigma}_0^{-1/2} \boldsymbol{v}$ and the approximation rate is given by*

$$\mathcal{E}_4(p) := \frac{\rho_{\gamma_+}(p)^6 \log^{7/2}(p)}{\sqrt{p}}. \tag{103}$$

This proposition is a direct consequence of [Misiakiewicz and Saeed, 2024, Lemma 10].

Throughout the proofs, with a slight abuse of notations, we will denote functionals $\widetilde{\Phi}_i(\boldsymbol{F}_0; \boldsymbol{B}, \kappa)$, $i \in [4]$, the functionals listed in Eq. (93) applied to the truncated feature matrix $\boldsymbol{F}_0$, with covariance $\boldsymbol{\Sigma}_0$, regularization $\gamma(\kappa)$, and deterministic matrix $\boldsymbol{B} \in \mathbb{R}^{\mathsf{m} \times \mathsf{m}}$, and $\widetilde{\Phi}_i(\boldsymbol{F}; \boldsymbol{B}, \kappa)$, $i \in [4]$, the functionals applied to the full feature matrix $\boldsymbol{F} \in \mathbb{R}^{p \times \infty}$, where $\boldsymbol{\Sigma}_0$ is replaced by $\boldsymbol{\Sigma}$, the regularization parameter is $\kappa$, and the deterministic matrix $\boldsymbol{B} \in \mathbb{R}^{\infty \times \infty}$. Similarly, we will us the notation $\Psi_i(\nu_{2,0}; \boldsymbol{B})$, $i \in [3]$ for the truncated deterministic functionals (94), and the notation $\Psi_i(\nu_2; \boldsymbol{B})$, $i \in [3]$ to denote the full functionals with $\boldsymbol{\Sigma}_0$ replaced by $\boldsymbol{\Sigma}$ and $\boldsymbol{B} \in \mathbb{R}^{\infty \times \infty}$.

## B.5 Approximation guarantee for the variance term

Recall the expressions for the variance term

$$\mathcal{V}(\boldsymbol{G}, \boldsymbol{F}, \lambda) = \sigma_\varepsilon^2 \cdot \mathrm{Tr}\big(\widehat{\boldsymbol{\Sigma}}_{\boldsymbol{F}} \boldsymbol{Z}^\mathsf{T} \boldsymbol{Z} (\boldsymbol{Z}^\mathsf{T} \boldsymbol{Z} + \lambda)^{-2}\big),$$

and its associated deterministic equivalent

$$\mathsf{V}_{n,p}(\lambda) = \sigma_\varepsilon^2 \frac{\Upsilon(\nu_1, \nu_2)}{1 - \Upsilon(\nu_1, \nu_2)}, \tag{104}$$

$$\Upsilon(\nu_1, \nu_2) = \frac{p}{n} \left[ \left( 1 - \frac{\nu_1}{\nu_2} \right)^2 + \left( \frac{\nu_1}{\nu_2} \right)^2 \frac{\mathrm{Tr}(\boldsymbol{\Sigma}^2 (\boldsymbol{\Sigma} + \nu_2)^{-2})}{p - \mathrm{Tr}(\boldsymbol{\Sigma}^2 (\boldsymbol{\Sigma} + \nu_2)^{-2})} \right]. \tag{105}$$

We prove in this section an approximation guarantee between $\mathcal{V}(\boldsymbol{G}, \boldsymbol{F}, \lambda)$ and $\mathsf{V}_{n,p}(\lambda)$. For convenience, we state a separate theorem for this term.

**Theorem B.10** (Deterministic equivalent for the variance term). *Assume the features $(\boldsymbol{z}_i)_{i \in [n]}$ and $(\boldsymbol{f}_j)_{j \in [p]}$ satisfy Assumption B.1, and the covariance $\boldsymbol{\Sigma}$ and target coefficients $\boldsymbol{\beta}_*$ satisfy Assumption 3.2. Then, for any $D, K > 0$, there exist constants $\eta_* \in (0, 1/2)$ and $C_{*,D,K} > 0$ such that the following holds. Let $\rho_\kappa$ and $\widetilde{\rho}_\kappa$ be defined as per Eqs. (26) and (25). For any $n, p \ge C_{*,D,K}$ and $\lambda > 0$, if it holds that*

$$\lambda \ge n^{-K}, \qquad \gamma_\lambda \ge p^{-K}, \qquad \begin{aligned} &\widetilde{\rho}_\lambda(n,p)^{5/2} \cdot \log^{3/2}(n) \le K\sqrt{n}, \\ &\widetilde{\rho}_\lambda(n,p)^2 \cdot \rho_{\gamma_+}(p)^7 \cdot \log^4(p) \le K\sqrt{p}, \end{aligned} \tag{106}$$

*then with probability at least $1 - n^{-D} - p^{-D}$, we have*

$$|\mathcal{V}(\boldsymbol{G}, \boldsymbol{F}, \lambda) - \mathsf{V}_{n,p}(\lambda)| \le C_{*,D,K} \cdot \mathcal{E}_V(n,p) \cdot \mathsf{V}_{n,p}(\lambda),$$

*where the approximation rate is given by*

$$\mathcal{E}_V(n,p) := \frac{\widetilde{\rho}_\lambda(n,p)^6 \log^{5/2}(n)}{\sqrt{n}} + \frac{\widetilde{\rho}_\lambda(n,p)^2 \cdot \rho_{\gamma_+}(p)^7 \log^3(p)}{\sqrt{p}}. \tag{107}$$

*Proof of Theorem B.10.* First, note that $\mathcal{V}(\boldsymbol{G}, \boldsymbol{F}, \lambda)$ can be written in terms of the functional $\Phi_4$ defined in Eq. (83):

$$\mathcal{V}(\boldsymbol{G}, \boldsymbol{F}, \lambda) = \sigma_\varepsilon^2 \cdot n \Phi_4(\boldsymbol{Z}; \boldsymbol{I}, \lambda).$$

Recall that $\mathcal{A}_{\mathcal{F}}$ is the event defined in Eq. (79). Under the assumptions of Theorem B.10, we can apply Lemma B.2 and Proposition B.4 to obtain

$$\mathbb{P}(\mathcal{A}_{\mathcal{F}}) \geq 1 - p^{-D}.$$

Hence, applying Proposition B.6 for $\boldsymbol{F} \in \mathcal{A}_{\mathcal{F}}$ and via union bound, we obtain that with probability at least $1 - p^{-D} - n^{-D}$,

$$\left| n\Phi_4(\boldsymbol{Z}; \mathbf{I}, \lambda) - n\widetilde{\Phi}_5(\boldsymbol{F}; \mathbf{I}, p\nu_1) \right| \leq C_{*,D,K} \cdot \mathcal{E}_1(p, n) \cdot n\widetilde{\Phi}_5(\boldsymbol{F}; \mathbf{I}, p\nu_1), \qquad (108)$$

where $\mathcal{E}_1(n, p)$ is defined in Eq. (89) and we recall the expressions

$$n\widetilde{\Phi}_5(\boldsymbol{F}; \mathbf{I}, p\nu_1) = \frac{\widetilde{\Phi}_6(\boldsymbol{F}; \mathbf{I}, p\nu_1)}{n - \widetilde{\Phi}_6(\boldsymbol{F}; \mathbf{I}, p\nu_1)}, \qquad \widetilde{\Phi}_6(\boldsymbol{F}; \mathbf{I}, p\nu_1) = \mathrm{Tr}\big((\boldsymbol{F}\boldsymbol{F}^{\mathsf{T}})^2(\boldsymbol{F}\boldsymbol{F}^{\mathsf{T}} + p\nu_1)^{-2}\big).$$

Let us decompose $\widetilde{\Phi}_6(\boldsymbol{F}; \mathbf{I}, p\nu_1)$ into

$$\widetilde{\Phi}_6(\boldsymbol{F}; \mathbf{I}, p\nu_1) = \mathrm{Tr}(\boldsymbol{F}\boldsymbol{F}^{\mathsf{T}}(\boldsymbol{F}\boldsymbol{F}^{\mathsf{T}} + p\nu_1)^{-1}) - p\nu_1\mathrm{Tr}(\boldsymbol{F}\boldsymbol{F}^{\mathsf{T}}(\boldsymbol{F}\boldsymbol{F}^{\mathsf{T}} + p\nu_1)^{-2}).$$

From Lemma B.11 stated below, we have with probability at least $1 - p^{-D}$

$$\left| \frac{1}{p}\mathrm{Tr}(\boldsymbol{F}\boldsymbol{F}^{\mathsf{T}}(\boldsymbol{F}\boldsymbol{F}^{\mathsf{T}} + p\nu_1)^{-1}) - \Psi_2(\nu_2) \right| \leq C_{*,D,K} \cdot \mathcal{E}_3(p) \cdot \Psi_2(\nu_2),$$

$$\left| \frac{1}{p}\mathrm{Tr}(\boldsymbol{F}\boldsymbol{F}^{\mathsf{T}}(\boldsymbol{F}\boldsymbol{F}^{\mathsf{T}} + p\nu_1)^{-2}) - \Psi_3(\nu_2; \boldsymbol{\Sigma}^{-1}) \right| \leq C_{*,D,K} \cdot \rho_{\gamma_+}(p)\mathcal{E}_3(p) \cdot \Psi_3(\nu_2; \boldsymbol{\Sigma}^{-1}),$$

where $\mathcal{E}_3(p)$ is the approximation rate defined in Eq. (100). Note that

$$p\Psi_2(\nu_2) - p^2\nu_1\Psi_3(\nu_2; \boldsymbol{\Sigma}^{-1}) = \mathrm{Tr}(\boldsymbol{\Sigma}(\boldsymbol{\Sigma} + \nu_2)^{-1}) - \frac{\nu_1\mathrm{Tr}(\boldsymbol{\Sigma}(\boldsymbol{\Sigma} + \nu_2)^{-2})}{1 - \frac{1}{p}\mathrm{Tr}(\boldsymbol{\Sigma}^2(\boldsymbol{\Sigma} + \nu_2)^{-2})}$$

$$= p\left\{ 1 - \frac{\nu_1}{\nu_2} - \frac{\nu_1}{\nu_2}\frac{1 - \frac{\nu_1}{\nu_2} - \frac{1}{p}\mathrm{Tr}(\boldsymbol{\Sigma}^2(\boldsymbol{\Sigma} + \nu_2)^{-2})}{1 - \frac{1}{p}\mathrm{Tr}(\boldsymbol{\Sigma}^2(\boldsymbol{\Sigma} + \nu_2)^{-2})} \right\}$$

$$= p\left\{ \left(1 - \frac{\nu_1}{\nu_2}\right)^2 + \left(\frac{\nu_1}{\nu_2}\right)^2 \frac{\mathrm{Tr}(\boldsymbol{\Sigma}^2(\boldsymbol{\Sigma} + \nu_2)^{-2})}{p - \mathrm{Tr}(\boldsymbol{\Sigma}^2(\boldsymbol{\Sigma} + \nu_2)^{-2})} \right\}$$

$$= n\Upsilon(\nu_1, \nu_2).$$

Combining the above displays, we deduce that

$$\left| \widetilde{\Phi}_6(\boldsymbol{F}; \mathbf{I}, p\nu_1) - n\Upsilon(\nu_1, \nu_2) \right| \leq C_{*,D,K} \cdot \rho_{\gamma_+}(p)\mathcal{E}_3(p) \cdot \left[ p\Psi_2(\nu_2) + p^2\nu_1\Psi_3(\nu_2; \boldsymbol{\Sigma}^{-1}) \right]$$

$$\leq C_{*,D,K} \cdot \rho_{\gamma_+}(p)\mathcal{E}_3(p) \cdot \left[ n\Upsilon(\nu_1, \nu_2) + 2p^2\nu_1\Psi_3(\nu_2; \boldsymbol{\Sigma}^{-1}) \right]$$

Using Eq. (120) in Lemma B.14 stated in Section B.7, we conclude that with probability at least $1 - p^{-D}$,

$$\left| \widetilde{\Phi}_6(\boldsymbol{F}; \mathbf{I}, p\nu_1) - n\Upsilon(\nu_1, \nu_2) \right| \leq C_{*,D,K} \cdot \rho_{\gamma_+}(p)\mathcal{E}_3(p) \cdot n\Upsilon(\nu_1, \nu_2). \qquad (109)$$

Finally, by simple algebra, we have

$$\left| n\widetilde{\Phi}_5(\boldsymbol{F}; \mathbf{I}, p\nu_1) - \frac{\Upsilon(\nu_1, \nu_2)}{1 - \Upsilon(\nu_1, \nu_2)} \right|$$

$$\leq \left\{ n\widetilde{\Phi}_5(\boldsymbol{F}; \mathbf{I}, p\nu_1) + 1 \right\} \frac{|\widetilde{\Phi}_6(\boldsymbol{F}; \mathbf{I}, p\nu_1) - n\Upsilon(\nu_1, \nu_2)|}{n - \Upsilon(\nu_1, \nu_2)}$$

$$\leq \left\{ \left| n\widetilde{\Phi}_5(\boldsymbol{F}; \mathbf{I}, p\nu_1) - \frac{\Upsilon(\nu_1, \nu_2)}{1 - \Upsilon(\nu_1, \nu_2)} \right| + \frac{1}{1 - \Upsilon(\nu_1, \nu_2)} \right\} \cdot C_{*,D,K} \cdot \rho_{\gamma_+}(p)\mathcal{E}_3(p)\frac{\Upsilon(\nu_1, \nu_2)}{1 - \Upsilon(\nu_1, \nu_2)}. \qquad (110)$$

Note that by Eq. (119) in Lemma B.14, we have

$$\frac{\Upsilon(\nu_1, \nu_2)}{1 - \Upsilon(\nu_1, \nu_2)} \leq (1 - \Upsilon(\nu_1, \nu_2))^{-1} \leq C_* \cdot \widetilde{\rho}_\lambda(n, p).$$

Hence rearranging the terms in Eq. (110) and using from conditions (106) and that $p \geq C_{*,D,K}$, we obtain with probability at least $1 - p^{-D}$ that

$$\left| n\widetilde{\Phi}_5(\boldsymbol{F}; \mathbf{I}, p\nu_1) - \frac{\Upsilon(\nu_1, \nu_2)}{1 - \Upsilon(\nu_1, \nu_2)} \right| \leq C_{*,D,K} \cdot \widetilde{\rho}_\lambda(n,p)\rho_{\gamma_+}(p)\mathcal{E}_3(p)\frac{\Upsilon(\nu_1, \nu_2)}{1 - \Upsilon(\nu_1, \nu_2)}. \qquad (111)$$

Using an union bound and combining bounds Eqs. (108) and (111), we obtain with probability at least $1 - n^{-D} - p^{-D}$, that

$$|\mathcal{V}(\boldsymbol{G}, \boldsymbol{F}, \lambda) - \mathsf{V}_{n,p}(\lambda)|$$

$$\leq \left| \mathcal{V}(\boldsymbol{G}, \boldsymbol{F}, \lambda) - \sigma_\varepsilon^2 \cdot n\widetilde{\Phi}_5(\boldsymbol{F}; \mathbf{I}, p\nu_1) \right| + \sigma_\varepsilon^2 \left| n\widetilde{\Phi}_5(\boldsymbol{F}; \mathbf{I}, p\nu_1) - \frac{\Upsilon(\nu_1, \nu_2)}{1 - \Upsilon(\nu_1, \nu_2)} \right|$$

$$\leq C_{*,D,K} \cdot \left\{ \mathcal{E}_1(p,n) + \widetilde{\rho}_\lambda(n,p)\rho_{\gamma_+}(p)\mathcal{E}_3(p) \right\} \cdot \left[ \sigma_\varepsilon^2 \cdot n\widetilde{\Phi}_5(\boldsymbol{F}; \mathbf{I}, p\nu_1) + \mathsf{V}_{n,p}(\lambda) \right]$$

$$\leq C_{*,D,K} \cdot \left\{ \mathcal{E}_1(p,n) + \widetilde{\rho}_\lambda(n,p)\rho_{\gamma_+}(p)\mathcal{E}_3(p) \right\} \cdot \mathsf{V}_{n,p}(\lambda),$$

where we used Eq. (111) and conditions (106) in the last line. Replacing the rates $\mathcal{E}_j$ by their expressions conclude the proof of this theorem. $\qquad \square$

**Lemma B.11.** *Under the setting of Theorem B.10 and assuming the same conditions (106), we have with probability at least $1 - p^{-D}$,*

$$\left| \frac{1}{p}\mathrm{Tr}(\boldsymbol{F}\boldsymbol{F}^\mathsf{T}(\boldsymbol{F}\boldsymbol{F}^\mathsf{T} + p\nu_1)^{-1}) - \Psi_2(\nu_2) \right| \leq C_{*,D,K} \cdot \mathcal{E}_3(p) \cdot \Psi_2(\nu_2), \qquad (112)$$

$$\left| \frac{1}{p}\mathrm{Tr}(\boldsymbol{F}\boldsymbol{F}^\mathsf{T}(\boldsymbol{F}\boldsymbol{F}^\mathsf{T} + p\nu_1)^{-2}) - \Psi_3(\nu_2; \boldsymbol{\Sigma}^{-1}) \right| \leq C_{*,D,K} \cdot \rho_{\gamma_+}(p)\mathcal{E}_3(p) \cdot \Psi_3(\nu_2; \boldsymbol{\Sigma}^{-1}), \quad (113)$$

*where $\mathcal{E}_3(p)$ is the approximation rate defined in Eq. (100).*

The proof of this lemma can be found in Section B.7.4.

## B.6 Approximation guarantee for the bias term

Recall the expression for the bias term

$$\mathcal{B}(\boldsymbol{\beta}_*; \boldsymbol{G}, \boldsymbol{F}, \lambda) = \|\boldsymbol{\beta}_* - p^{-1/2}\boldsymbol{F}^\mathsf{T}(\boldsymbol{Z}^\mathsf{T}\boldsymbol{Z} + \lambda)^{-1}\boldsymbol{Z}^\mathsf{T}\boldsymbol{G}\boldsymbol{\beta}_*\|_2^2,$$

and its associated deterministic equivalent

$$\chi(\nu_2) = \frac{\mathrm{Tr}(\boldsymbol{\Sigma}(\boldsymbol{\Sigma} + \nu_2)^{-2})}{p - \mathrm{Tr}(\boldsymbol{\Sigma}^2(\boldsymbol{\Sigma} + \nu_2)^{-2})},$$

$$\mathsf{B}_{n,p}(\boldsymbol{\beta}_*, \lambda) = \frac{\nu_2^2}{1 - \Upsilon(\nu_1, \nu_2)}\left[ \langle \boldsymbol{\beta}_*, (\boldsymbol{\Sigma} + \nu_2)^{-2}\boldsymbol{\beta}_* \rangle + \chi(\nu_2)\langle \boldsymbol{\beta}_*, \boldsymbol{\Sigma}(\boldsymbol{\Sigma} + \nu_2)^{-2}\boldsymbol{\beta}_* \rangle \right].$$

We prove in this section an approximation guarantee between $\mathcal{B}(\boldsymbol{\beta}_*; \boldsymbol{G}, \boldsymbol{F}, \lambda)$ and $\mathsf{B}_{n,p}(\boldsymbol{\beta}_*, \lambda)$. For convenience, we state a separate theorem for this term.

**Theorem B.12** (Deterministic equivalent for the bias term)**.** *Assume the features $(\boldsymbol{z}_i)_{i\in[n]}$ and $(\boldsymbol{f}_j)_{j\in[p]}$ satisfy Assumption B.1, and that there exists $\mathsf{m} \in \mathbb{N}$ such that $p^2\xi_\mathsf{m}^2 \leq \gamma_\mathsf{m}(n\lambda/p)$. Then, for any $D, K > 0$, there exist constants $\eta_* \in (0, 1/2)$ and $C_{*,D,K} > 0$ such that the following holds. Let $\rho_\kappa$ and $\widetilde{\rho}_\kappa$ be defined as per Eqs. (26) and (25), and recall that $\gamma_\lambda := \gamma_\mathsf{m}(n\lambda/p)$ and $\gamma_+ := \gamma_\mathsf{m}(p\nu_1)$. For any $n, p \geq C_{*,D,K}$ and $\lambda > 0$, if it holds that*

$$\lambda \geq n^{-K}, \qquad \gamma_\lambda \geq p^{-K}, \qquad \widetilde{\rho}_\lambda(n,p)^{5/2} \cdot \log^{3/2}(n) \leq K\sqrt{n}, \qquad (114)$$
$$\widetilde{\rho}_\lambda(n,p)^2 \cdot \rho_{\gamma_+}(p)^8 \cdot \log^4(p) \leq K\sqrt{p},$$

*then with probability at least $1 - n^{-D} - p^{-D}$, we have*

$$|\mathcal{B}(\boldsymbol{\beta}_*; \boldsymbol{G}, \boldsymbol{F}, \lambda) - \mathsf{B}_{n,p}(\boldsymbol{\beta}_*, \lambda)| \leq C_{*,D,K} \cdot \mathcal{E}_B(n,p) \cdot \mathsf{B}_{n,p}(\boldsymbol{\beta}_*, \lambda),$$

*where the approximation rate is given by*

$$\mathcal{E}_B(n,p) := \frac{\widetilde{\rho}_\lambda(n,p)^6 \log^{7/2}(n)}{\sqrt{n}} + \frac{\widetilde{\rho}_\lambda(n,p)^2 \cdot \rho_{\gamma_+}(p)^8 \log^{7/2}(p)}{\sqrt{p}}. \qquad (115)$$

Before starting the proof, let us introduce some notations. First, define $\mathsf{P}_{\boldsymbol{F}}$ the projection onto the span of $\boldsymbol{F}$, and $\mathsf{P}_{\perp,\boldsymbol{F}}$ the projection orthogonal to the span of $\boldsymbol{F}$, i.e.,

$$\mathsf{P}_{\boldsymbol{F}} := (\boldsymbol{F}^{\mathsf{T}}\boldsymbol{F})^{\dagger}\boldsymbol{F}^{\mathsf{T}}\boldsymbol{F}, \qquad \mathsf{P}_{\perp,\boldsymbol{F}} = \boldsymbol{I} - \mathsf{P}_{\boldsymbol{F}}.$$

We can decompose the feature $\boldsymbol{g}$ with respect to the orthogonal sum of these two subspaces

$$\boldsymbol{g} = \mathsf{P}_{\boldsymbol{F}}\boldsymbol{g} + \mathsf{P}_{\perp,\boldsymbol{F}}\boldsymbol{g} = \sqrt{p}(\boldsymbol{F}^{\mathsf{T}}\boldsymbol{F})^{\dagger}\boldsymbol{F}^{\mathsf{T}}\boldsymbol{z} + \mathsf{P}_{\perp,\boldsymbol{F}}\boldsymbol{g}.$$

We define $\boldsymbol{r} := \mathsf{P}_{\perp,\boldsymbol{F}}\boldsymbol{g}$ and $\boldsymbol{R} = [\boldsymbol{r}_1,\ldots,\boldsymbol{r}_n]^{\mathsf{T}} \in \mathbb{R}^{n\times\infty}$. Similarly, we can decompose the target function

$$h_*(\boldsymbol{g}) = \langle\boldsymbol{\beta}_*,\boldsymbol{g}\rangle = \langle\boldsymbol{\beta}_{\boldsymbol{F}},\boldsymbol{z}\rangle + \langle\boldsymbol{\beta}_{\perp,\boldsymbol{F}},\boldsymbol{r}\rangle,$$

where we introduced

$$\boldsymbol{\beta}_{\boldsymbol{F}} := \sqrt{p}\boldsymbol{F}(\boldsymbol{F}^{\mathsf{T}}\boldsymbol{F})^{\dagger}\boldsymbol{\beta}_*, \qquad \boldsymbol{\beta}_{\perp,\boldsymbol{F}} = \mathsf{P}_{\perp,\boldsymbol{F}}\boldsymbol{\beta}_*.$$

Note that in particular $\mathbb{E}[\boldsymbol{z}\langle\boldsymbol{r},\boldsymbol{\beta}_{\perp,\boldsymbol{F}}\rangle] = \boldsymbol{0}$ by orthogonality.

*Proof of Theorem B.12.* **Step 0: Decomposing the bias term.**

Note that using the notations introduced above, we can decompose the bias term into

$$\begin{aligned}
\mathcal{B}(\boldsymbol{\beta}_*;\boldsymbol{G},\boldsymbol{F},\lambda) &= \|\boldsymbol{\beta}_* - p^{-1/2}\boldsymbol{F}^{\mathsf{T}}(\boldsymbol{Z}^{\mathsf{T}}\boldsymbol{Z}+\lambda)^{-1}\boldsymbol{Z}^{\mathsf{T}}\boldsymbol{G}\boldsymbol{\beta}_*\|_2^2 \\
&= \frac{1}{p}\left\|\boldsymbol{F}^{\mathsf{T}}\left(\boldsymbol{\beta}_{\boldsymbol{F}} - (\boldsymbol{Z}^{\mathsf{T}}\boldsymbol{Z}+\lambda)^{-1}\boldsymbol{Z}^{\mathsf{T}}(\boldsymbol{Z}\boldsymbol{\beta}_{\boldsymbol{F}}+\boldsymbol{R}\boldsymbol{\beta}_{\perp,\boldsymbol{F}})\right)\right\|_2^2 + \|\boldsymbol{\beta}_{\perp,\boldsymbol{F}}\|_2^2 \\
&= T_1 - 2T_2 + T_3 + \|\boldsymbol{\beta}_{\perp,\boldsymbol{F}}\|_2^2.
\end{aligned}$$

where we denoted

$$\begin{aligned}
T_1 &:= \lambda^2\langle\boldsymbol{\beta}_{\boldsymbol{F}},(\boldsymbol{Z}^{\mathsf{T}}\boldsymbol{Z}+\lambda)^{-1}\widehat{\boldsymbol{\Sigma}}_{\boldsymbol{F}}(\boldsymbol{Z}^{\mathsf{T}}\boldsymbol{Z}+\lambda)^{-1}\boldsymbol{\beta}_{\boldsymbol{F}}\rangle, \\
T_2 &:= \lambda\langle\boldsymbol{\beta}_{\boldsymbol{F}},(\boldsymbol{Z}^{\mathsf{T}}\boldsymbol{Z}+\lambda)^{-1}\widehat{\boldsymbol{\Sigma}}_{\boldsymbol{F}}(\boldsymbol{Z}^{\mathsf{T}}\boldsymbol{Z}+\lambda)^{-1}\boldsymbol{Z}^{\mathsf{T}}\boldsymbol{R}\boldsymbol{\beta}_{\perp,\boldsymbol{F}}\rangle, \\
T_3 &:= \langle\boldsymbol{\beta}_{\perp,\boldsymbol{F}},\boldsymbol{R}^{\mathsf{T}}\boldsymbol{Z}(\boldsymbol{Z}^{\mathsf{T}}\boldsymbol{Z}+\lambda)^{-1}\widehat{\boldsymbol{\Sigma}}_{\boldsymbol{F}}(\boldsymbol{Z}^{\mathsf{T}}\boldsymbol{Z}+\lambda)^{-1}\boldsymbol{Z}^{\mathsf{T}}\boldsymbol{R}\boldsymbol{\beta}_{\perp,\boldsymbol{F}}\rangle.
\end{aligned}$$

We proceed similarly to the proof for the variance term, by first considering the deterministic equivalent over $\boldsymbol{Z}$ conditional on $\boldsymbol{F}$, and then over $\boldsymbol{F}$. We omit some repetitive details for the sake of brevity.

**Step 1: Deterministic equivalent over $\boldsymbol{Z}$ conditional on $\boldsymbol{F}$.**

First note that, denoting $\tilde{\boldsymbol{A}}_* = \widehat{\boldsymbol{\Sigma}}_{\boldsymbol{F}}^{-1/2}\boldsymbol{\beta}_{\boldsymbol{F}}\boldsymbol{\beta}_{\boldsymbol{F}}^{\mathsf{T}}\widehat{\boldsymbol{\Sigma}}_{\boldsymbol{F}}^{-1/2}$, we have

$$T_1 = \lambda^2\Phi_3(\boldsymbol{Z};\tilde{\boldsymbol{A}}_*,\lambda).$$

Furthermore, $T_3$ and $T_2$ correspond respectively to the terms (90) and (91) in Proposition B.7 with $\boldsymbol{v} = \boldsymbol{\beta}_{\boldsymbol{F}}$ and $\boldsymbol{u} = \boldsymbol{R}\boldsymbol{\beta}_{\perp,\boldsymbol{F}}$ where

$$\mathbb{E}[\boldsymbol{z}_iu_i] = \mathbb{E}[\boldsymbol{z}_i\langle\boldsymbol{r}_i,\boldsymbol{\beta}_{\perp,\boldsymbol{F}}\rangle] = 0, \qquad \mathbb{E}[u_i^2] = \|\boldsymbol{\beta}_{\perp,\boldsymbol{F}}\|_2^2.$$

Thus, under the assumptions of Theorem B.12, we can apply Propositions B.8 and B.7 to obtain (via union bound) that with probability at least $1 - n^{-D} - p^{-D}$,

$$\left|T_1 - (n\nu_1)^2\widetilde{\Phi}_5(\boldsymbol{F};\tilde{\boldsymbol{A}}_*,p\nu_1)\right| \le C_{*,D,K}\cdot\mathcal{E}_1(p,n)\cdot(n\nu_1)^2\widetilde{\Phi}_5(\boldsymbol{F};\tilde{\boldsymbol{A}}_*,p\nu_1),$$

$$|T_2| \le C_{*,D,K}\cdot\mathcal{E}_2(p,n)\cdot\sqrt{\|\boldsymbol{\beta}_{\perp,\boldsymbol{F}}\|_2^2\cdot(n\nu_1)^2\widetilde{\Phi}_5(\boldsymbol{F};\tilde{\boldsymbol{A}}_*,p\nu_1)},$$

$$\left|T_3 - \|\boldsymbol{\beta}_{\perp,\boldsymbol{F}}\|_2^2\cdot n\widetilde{\Phi}_5(\boldsymbol{F};\boldsymbol{I},p\nu_1)\right| \le C_{*,D,K}\cdot\mathcal{E}_2(p,n)\cdot\|\boldsymbol{\beta}_{\perp,\boldsymbol{F}}\|_2^2.$$

Hence we deduce that

$$\left|\mathcal{B}(\boldsymbol{\beta}_*;\boldsymbol{G},\boldsymbol{F},\lambda) - (n\nu_1)^2\widetilde{\Phi}_5(\boldsymbol{F};\tilde{\boldsymbol{A}}_*,p\nu_1) - \|\boldsymbol{\beta}_{\perp,\boldsymbol{F}}\|_2^2\cdot n\widetilde{\Phi}_5(\boldsymbol{F};\boldsymbol{I},p\nu_1) - \|\boldsymbol{\beta}_{\perp,\boldsymbol{F}}\|_2^2\right|$$

$$\le C_{*,D,K}\cdot\{\mathcal{E}_1(n,p)+\mathcal{E}_2(n,p)\}\cdot\left[(n\nu_1)^2\widetilde{\Phi}_5(\boldsymbol{F};\tilde{\boldsymbol{A}}_*,p\nu_1)+\|\boldsymbol{\beta}_{\perp,\boldsymbol{F}}\|_2^2\right].$$

Let us simplify these terms. Recall that

$$n\widetilde{\Phi}_5(\boldsymbol{F}; \tilde{\boldsymbol{A}}_*, p\nu_1) = \frac{\widetilde{\Phi}_6(\boldsymbol{F}; \tilde{\boldsymbol{A}}_*, p\nu_1)}{n - \widetilde{\Phi}_6(\boldsymbol{F}; \mathbf{I}, p\nu_1)}, \qquad n\widetilde{\Phi}_5(\boldsymbol{F}; \mathbf{I}, p\nu_1)\frac{\widetilde{\Phi}_6(\boldsymbol{F}; \mathbf{I}, p\nu_1)}{n - \widetilde{\Phi}_6(\boldsymbol{F}; \mathbf{I}, p\nu_1)}.$$

For the term involving $\tilde{\boldsymbol{A}}_*$, we can rewrite it as

$$\begin{aligned}
\nu_1^2\widetilde{\Phi}_6(\boldsymbol{F}; \tilde{\boldsymbol{A}}_*, p\nu_1) &= \nu_1^2\langle\boldsymbol{\beta_F}, \widehat{\boldsymbol{\Sigma}}_{\boldsymbol{F}}^{-1}(\boldsymbol{F}^\top\boldsymbol{F})^2(\boldsymbol{F}^\top\boldsymbol{F} + p\nu_1)^{-2}\boldsymbol{\beta_F}\rangle \\
&= (p\nu_1)^2\langle\boldsymbol{\beta}_*, (\boldsymbol{F}^\top\boldsymbol{F})^\dagger(\boldsymbol{F}^\top\boldsymbol{F})(\boldsymbol{F}^\top\boldsymbol{F} + p\nu_1)^{-2}(\boldsymbol{F}^\top\boldsymbol{F})(\boldsymbol{F}^\top\boldsymbol{F})^\dagger\boldsymbol{\beta}_*\rangle \\
&= (p\nu_1)^2\langle\boldsymbol{\beta}_*, (\boldsymbol{F}^\top\boldsymbol{F} + p\nu_1)^{-2}\boldsymbol{\beta}_*\rangle - \|\boldsymbol{\beta}_{\perp,\boldsymbol{F}}\|_2^2.
\end{aligned}$$

Hence the terms involving $\|\boldsymbol{\beta}_{\perp,\boldsymbol{F}}\|_2^2$ cancel out and we obtain

$$(n\nu_1)^2\widetilde{\Phi}_5(\boldsymbol{F}; \tilde{\boldsymbol{A}}_*, p\nu_1) + \|\boldsymbol{\beta}_{\perp,\boldsymbol{F}}\|_2^2 \cdot n\widetilde{\Phi}_5(\boldsymbol{F}; \mathbf{I}, p\nu_1) + \|\boldsymbol{\beta}_{\perp,\boldsymbol{F}}\|_2^2 = (p\nu_1)^2\frac{\langle\boldsymbol{\beta}_*, (\boldsymbol{F}^\top\boldsymbol{F} + p\nu_1)^{-2}\boldsymbol{\beta}_*\rangle}{1 - \frac{1}{n}\widetilde{\Phi}_6(\boldsymbol{F}; \mathbf{I}, p\nu_1)}.$$

Combining the above displays, we deduce that with probability at least $1 - n^{-D} - p^{-D}$,

$$\begin{aligned}
&\left|\mathcal{B}(\boldsymbol{\beta}_*; \boldsymbol{G}, \boldsymbol{F}, \lambda) - (p\nu_1)^2\frac{\langle\boldsymbol{\beta}_*, (\boldsymbol{F}^\top\boldsymbol{F} + p\nu_1)^{-2}\boldsymbol{\beta}_*\rangle}{1 - \frac{1}{n}\widetilde{\Phi}_6(\boldsymbol{F}; \mathbf{I}, p\nu_1)}\right| \\
&\leq C_{*,D,K} \cdot \{\mathcal{E}_1(n,p) + \mathcal{E}_2(n,p)\} \cdot (p\nu_1)^2\frac{\langle\boldsymbol{\beta}_*, (\boldsymbol{F}^\top\boldsymbol{F} + p\nu_1)^{-2}\boldsymbol{\beta}_*\rangle}{1 - \frac{1}{n}\widetilde{\Phi}_6(\boldsymbol{F}; \mathbf{I}, p\nu_1)}.
\end{aligned} \tag{116}$$

**Step 2: Deterministic equivalents over $\boldsymbol{F}$.**

Following the same steps as Eq. (110) in the proof of Theorem B.10, we have with probability at least $1 - p^{-D}$,

$$\left|(1 - n^{-1}\widetilde{\Phi}_6(\boldsymbol{F}; \mathbf{I}, p\nu_1))^{-1} - (1 - \Upsilon(\nu_1, \nu_2))^{-1}\right| \leq C_{*,D,K} \cdot \widetilde{\rho}_\lambda(n,p)\rho_{\gamma_+}(p)\mathcal{E}_3(p) \cdot (1 - \Upsilon(\nu_1, \nu_2))^{-1}.$$

Furthermore, from Lemma B.13 stated below, with probability at least $1 - p^{-D}$,

$$\left|(p\nu_1)^2\langle\boldsymbol{\beta}_*, (\boldsymbol{F}^\top\boldsymbol{F} + p\nu_1)^{-2}\boldsymbol{\beta}_*\rangle - \widetilde{\mathsf{B}}_{n,p}(\boldsymbol{\beta}_*, \lambda)\right| \leq C_{*,D,K} \cdot \rho_{\gamma_+}(p)^2\mathcal{E}_4(p) \cdot \widetilde{\mathsf{B}}_{n,p}(\boldsymbol{\beta}_*, \lambda),$$

where $\widetilde{\mathsf{B}}_{n,p}(\boldsymbol{\beta}_*, \lambda)$ is defined in Eq. (118). Combining these two bounds and recalling conditions (114), we obtain

$$\begin{aligned}
&\left|(p\nu_1)^2\frac{\langle\boldsymbol{\beta}_*, (\boldsymbol{F}^\top\boldsymbol{F} + p\nu_1)^{-2}\boldsymbol{\beta}_*\rangle}{1 - \frac{1}{n}\widetilde{\Phi}_6(\boldsymbol{F}; \mathbf{I}, p\nu_1)} - \frac{\widetilde{\mathsf{B}}_{n,p}(\boldsymbol{\beta}_*, \lambda)}{1 - \Upsilon(\nu_1, \nu_2)}\right| \\
&\leq C_{*,D,K}\left\{\widetilde{\rho}_\lambda(n,p)\rho_{\gamma_+}(p)\mathcal{E}_3(p) + \rho_{\gamma_+}(p)^2\mathcal{E}_4(p)\right\} \cdot \frac{\widetilde{\mathsf{B}}_{n,p}(\boldsymbol{\beta}_*, \lambda)}{1 - \Upsilon(\nu_1, \nu_2)}.
\end{aligned}$$

Noting that $\mathsf{B}_{n,p}(\boldsymbol{\beta}_*, \lambda) = \widetilde{\mathsf{B}}_{n,p}(\boldsymbol{\beta}_*, \lambda)/(1 - \Upsilon(\nu_1, \nu_2))$, we can combine this bound with Eq. (116) to obtain via union bound that with probability at least $1 - n^{-D} - p^{-D}$,

$$\begin{aligned}
&|\mathcal{B}(\boldsymbol{\beta}_*; \boldsymbol{G}, \boldsymbol{F}, \lambda) - \mathsf{B}_{n,p}(\boldsymbol{\beta}_*, \lambda)| \\
&\leq C_{*,D,K}\left\{\mathcal{E}_1(n,p) + \mathcal{E}_2(n,p) + \widetilde{\rho}_\lambda(n,p)\rho_{\gamma_+}(p)\mathcal{E}_3(p) + \rho_{\gamma_+}(p)^2\mathcal{E}_4(p)\right\} \cdot \mathsf{B}_{n,p}(\boldsymbol{\beta}_*, \lambda).
\end{aligned}$$

Replacing the rates $\mathcal{E}_j$ by their expressions conclude the proof of this theorem. $\qquad\square$

**Lemma B.13.** *Under the setting of Theorem B.12 and assuming the same conditions* (114)*, we have with probability at least $1 - p^{-D}$,*

$$\left|(p\nu_1)^2\langle\boldsymbol{\beta}_*, (\boldsymbol{F}^\top\boldsymbol{F} + p\nu_1)^{-2}\boldsymbol{\beta}_*\rangle - \widetilde{\mathsf{B}}_{n,p}(\boldsymbol{\beta}_*, \lambda)\right| \leq C_{*,D,K} \cdot \rho_{\gamma_+}(p)^2\mathcal{E}_4(p) \cdot \widetilde{\mathsf{B}}_{n,p}(\boldsymbol{\beta}_*, \lambda), \tag{117}$$

*where $\mathcal{E}_3(p)$ and $\mathcal{E}_4(p)$ are the approximation rates defined in Eqs.* (100) *and* (103)*, and we denoted*

$$\widetilde{\mathsf{B}}_{n,p}(\boldsymbol{\beta}_*, \lambda) := \nu_2^2\Big[\langle\boldsymbol{\beta}_*, (\boldsymbol{\Sigma} + \nu_2)^{-2}\boldsymbol{\beta}_*\rangle + \chi(\nu_2)\langle\boldsymbol{\beta}_*, \boldsymbol{\Sigma}(\boldsymbol{\Sigma} + \nu_2)^{-2}\boldsymbol{\beta}_*\rangle\Big]. \tag{118}$$

The proof of Lemma B.13 can be found in Section B.7.5.

## B.7  Technical results

In this section, we prove the technical results that were deferred from the previous sections. We start with a lemma that gathers useful bounds on the deterministic functionals.

**Lemma B.14.** *There exists a constant $C_* > 0$ such that*

$$(1 - \Upsilon(\nu_1, \nu_2))^{-1} \leq C_* \cdot \widetilde{\rho}_\lambda(n, p), \tag{119}$$

*where we recall that $\Upsilon(\nu_1, \nu_2)$ is defined as per Eq. (105). Furthermore, under Assumption 3.2, we have*

$$p\nu_1 \frac{\mathrm{Tr}(\boldsymbol{\Sigma}(\boldsymbol{\Sigma} + \nu_2)^{-2})}{p - \mathrm{Tr}(\boldsymbol{\Sigma}^2(\boldsymbol{\Sigma} + \nu_2)^{-2})} \leq C_* \cdot n\Upsilon(\nu_1, \nu_2), \tag{120}$$

$$p\nu_1 \frac{\langle \boldsymbol{\beta}_*, \boldsymbol{\Sigma}(\boldsymbol{\Sigma} + \nu_2)^{-2}\boldsymbol{\beta}_* \rangle}{p - \mathrm{Tr}(\boldsymbol{\Sigma}^2(\boldsymbol{\Sigma} + \nu_2)^{-2})} \leq C_* \cdot \widetilde{\mathsf{B}}_{n,p}(\boldsymbol{\beta}_*, \lambda), \tag{121}$$

*where $\widetilde{\mathsf{B}}_{n,p}(\boldsymbol{\beta}_*, \lambda)$ is defined as per Eq. (118) in Lemma B.13.*

*Proof of Lemma B.14.* **Step 1: Equation (119).**

Note that we have

$$\Upsilon(\nu_1, \nu_2) \leq \frac{1}{n}\mathrm{Tr}(\boldsymbol{\Sigma}(\boldsymbol{\Sigma} + \nu_2)^{-1}) \leq \frac{p}{n}.$$

In particular, if $n \geq p/\eta_*$, then we can simply write

$$(1 - \Upsilon(\nu_1, \nu_2))^{-1} \leq \frac{1}{1 - \eta_*} = C_*.$$

For $n \leq p/\eta_*$, note that using the first identity in Eq. (76), we have

$$(1 - \Upsilon(\nu_1, \nu_2))^{-1} \leq \left(1 - \frac{1}{n}\mathrm{Tr}(\boldsymbol{\Sigma}(\boldsymbol{\Sigma} + \nu_2)^{-1})\right)^{-1} \leq \frac{n\nu_1}{\lambda} \leq C_*\rho_\lambda(n).$$

Combining the previous two displays, we obtain Eq. (119).

**Step 2: Equation (120).**

We rewrite the left-hand side as

$$p\nu_1 \frac{\mathrm{Tr}(\boldsymbol{\Sigma}(\boldsymbol{\Sigma} + \nu_2)^{-2})}{p - \mathrm{Tr}(\boldsymbol{\Sigma}^2(\boldsymbol{\Sigma} + \nu_2)^{-2})} \leq \frac{p\nu_1}{p - \mathrm{Tr}(\boldsymbol{\Sigma}(\boldsymbol{\Sigma} + \nu_2)^{-1})}\mathrm{Tr}(\boldsymbol{\Sigma}(\boldsymbol{\Sigma} + \nu_2)^{-2})$$

$$= \mathrm{Tr}(\boldsymbol{\Sigma}(\boldsymbol{\Sigma} + \nu_2)^{-1}) - \mathrm{Tr}(\boldsymbol{\Sigma}^2(\boldsymbol{\Sigma} + \nu_2)^{-2})$$

$$\leq \left(1 - \frac{1}{C_*}\right)\mathrm{Tr}(\boldsymbol{\Sigma}(\boldsymbol{\Sigma} + \nu_2)^{-1}),$$

where we uses the second of the identities (76) in the second line, and Assumption 3.2 in the third line. Hence,

$$p\nu_1 \frac{\mathrm{Tr}(\boldsymbol{\Sigma}(\boldsymbol{\Sigma} + \nu_2)^{-2})}{p - \mathrm{Tr}(\boldsymbol{\Sigma}^2(\boldsymbol{\Sigma} + \nu_2)^{-2})} \leq (C_* - 1) \cdot \left\{\mathrm{Tr}(\boldsymbol{\Sigma}(\boldsymbol{\Sigma} + \nu_2)^{-1}) - p\nu_1\frac{\mathrm{Tr}(\boldsymbol{\Sigma}(\boldsymbol{\Sigma} + \nu_2)^{-2})}{p - \mathrm{Tr}(\boldsymbol{\Sigma}^2(\boldsymbol{\Sigma} + \nu_2)^{-2})}\right\}$$

$$= (C_* - 1) \cdot n\Upsilon(\nu_1, \nu_2).$$

**Step 3: Equation (121).**

Similarly, we get again using Eq. (76) and Assumption 3.2 that

$$p\nu_1 \frac{\langle \boldsymbol{\beta}_*, \boldsymbol{\Sigma}(\boldsymbol{\Sigma} + \nu_2)^{-2}\boldsymbol{\beta}_* \rangle}{p - \mathrm{Tr}(\boldsymbol{\Sigma}^2(\boldsymbol{\Sigma} + \nu_2)^{-2})} \leq \nu_2 \left\{\langle \boldsymbol{\beta}_*, (\boldsymbol{\Sigma} + \nu_2)^{-1}\boldsymbol{\beta}_* \rangle - \nu_2\langle \boldsymbol{\beta}_*, \boldsymbol{\Sigma}^2(\boldsymbol{\Sigma} + \nu_2)^{-2}\boldsymbol{\beta}_* \rangle\right\}$$

$$\leq \left(1 - \frac{1}{C_*}\right) \cdot \nu_2\langle \boldsymbol{\beta}_*, (\boldsymbol{\Sigma} + \nu_2)^{-1}\boldsymbol{\beta}_* \rangle,$$

and therefore

$$p\nu_1 \frac{\langle \boldsymbol{\beta}_*, \boldsymbol{\Sigma}(\boldsymbol{\Sigma} + \nu_2)^{-2}\boldsymbol{\beta}_* \rangle}{p - \mathrm{Tr}(\boldsymbol{\Sigma}^2(\boldsymbol{\Sigma} + \nu_2)^{-2})} \leq (C_* - 1) \left\{\nu_2\langle \boldsymbol{\beta}_*, (\boldsymbol{\Sigma} + \nu_2)^{-1}\boldsymbol{\beta}_* \rangle - p\nu_1\frac{\langle \boldsymbol{\beta}_*, \boldsymbol{\Sigma}(\boldsymbol{\Sigma} + \nu_2)^{-2}\boldsymbol{\beta}_* \rangle}{p - \mathrm{Tr}(\boldsymbol{\Sigma}^2(\boldsymbol{\Sigma} + \nu_2)^{-2})}\right\}$$

$$= \widetilde{\mathsf{B}}_{n,p}(\boldsymbol{\beta}_*, \lambda),$$

which concludes the proof of this lemma. $\qquad\square$

### B.7.1 Feature covariance matrix

*Proof of Lemma B.2.* Recall that we consider $\widehat{\boldsymbol{\Sigma}}_{\boldsymbol{F}} = p^{-1}\boldsymbol{F}\boldsymbol{F}^{\mathsf{T}}$, with $\boldsymbol{F} = [\boldsymbol{f}_1, \ldots, \boldsymbol{f}_p]^{\mathsf{T}} \in \mathbb{R}^{p \times \infty}$ and the $\boldsymbol{f}_i$ are i.i.d. random vectors satisfying Assumption B.1. For any integers $k_2 \geq k_1 \geq 1$, we split the weight feature matrix into $\boldsymbol{F} = [\boldsymbol{F}_1, \boldsymbol{F}_2, \boldsymbol{F}_3]$, where $\boldsymbol{F}_1 = [\boldsymbol{f}_{1,1}, \ldots, \boldsymbol{f}_{1,p}]^{\mathsf{T}} \in \mathbb{R}^{p \times k_1}$ with $\boldsymbol{f}_{1,j}$ the first $k_1$ coordinates of the feature vector $\boldsymbol{f}_j$, $\boldsymbol{F}_2 = [\boldsymbol{f}_{2,1}, \ldots, \boldsymbol{f}_{2,p}]^{\mathsf{T}} \in \mathbb{R}^{p \times (k_2-k_1)}$ with $\boldsymbol{f}_{2,j}$ the next $k_2 - k_1$ coordinates of $\boldsymbol{f}_j$, and $\boldsymbol{F}_3 = [\boldsymbol{f}_{3,1}, \ldots, \boldsymbol{f}_{3,p}]^{\mathsf{T}} \in \mathbb{R}^{p \times \infty}$ contains the rest of the coordinates. In other words, we split the feature vector into $\boldsymbol{f}_j = [\boldsymbol{f}_{1,j}, \boldsymbol{f}_{2,j}, \boldsymbol{f}_{3,j}]$. Denote

$$\boldsymbol{\Sigma}_1 = \mathbb{E}[\boldsymbol{f}_{1,j}\boldsymbol{f}_{1,j}^{\mathsf{T}}] = \mathrm{diag}(\xi_1^2, \ldots, \xi_{k_1}^2) \in \mathbb{R}^{k_1 \times k_1},$$
$$\boldsymbol{\Sigma}_2 = \mathbb{E}[\boldsymbol{f}_{2,j}\boldsymbol{f}_{2,j}^{\mathsf{T}}] = \mathrm{diag}(\xi_{k_1+1}^2, \ldots, \xi_{k_2}^2) \in \mathbb{R}^{(k_2-k_1) \times (k_2-k_1)},$$
$$\boldsymbol{\Sigma}_3 = \mathbb{E}[\boldsymbol{f}_{3,j}\boldsymbol{f}_{3,j}^{\mathsf{T}}] = \mathrm{diag}(\xi_{k_2+1}^2, \xi_{k_2+2}^2, \ldots) \in \mathbb{R}^{\infty \times \infty}.$$

We decompose the feature covariance matrix into

$$\frac{\boldsymbol{F}\boldsymbol{F}^{\mathsf{T}}}{p} = \frac{\boldsymbol{F}_1\boldsymbol{F}_1^{\mathsf{T}}}{p} + \frac{\boldsymbol{F}_2\boldsymbol{F}_2^{\mathsf{T}}}{p} + \frac{\boldsymbol{F}_3\boldsymbol{F}_3^{\mathsf{T}}}{p}. \tag{122}$$

**Step 1: Bounding the eigenvalues of $\boldsymbol{F}_1$ and $\boldsymbol{F}_2$.**

Introduce the whitened matrices

$$\overline{\boldsymbol{F}}_1 = \boldsymbol{F}_1\boldsymbol{\Sigma}_1^{-1/2} = [\overline{\boldsymbol{f}}_{1,1}, \ldots, \overline{\boldsymbol{f}}_{1,p}]^{\mathsf{T}}, \qquad \overline{\boldsymbol{F}}_2 = \boldsymbol{F}_2\boldsymbol{\Sigma}_2^{-1/2} = [\overline{\boldsymbol{f}}_{2,1}, \ldots, \overline{\boldsymbol{f}}_{2,p}]^{\mathsf{T}},$$

so that the feature vectors $\overline{\boldsymbol{f}}_{1,j}$ and $\overline{\boldsymbol{f}}_{2,j}$ have covariance $\mathbf{I}_{k_1}$ and $\mathbf{I}_{k_2-k_1}$ respectively. We have

$$\|\overline{\boldsymbol{F}}_1^{\mathsf{T}}\overline{\boldsymbol{F}}_1/p - \mathbf{I}_{k_1}\|_{\mathrm{op}} = \sup_{\boldsymbol{v} \in \mathbb{R}^{k_1}, \|\boldsymbol{v}\|_2=1} \left| \frac{1}{p}\sum_{j \in [p]} \langle \boldsymbol{v}, \overline{\boldsymbol{f}}_{1,j} \rangle^2 - 1 \right|$$

Denote $Z_j := \langle \boldsymbol{v}, \overline{\boldsymbol{f}}_{1,j} \rangle^2 - 1$. Then we have from Equation (60) applied to $\boldsymbol{B} = \boldsymbol{\Sigma}_1^{-1/2}\boldsymbol{v}\boldsymbol{v}^{\mathsf{T}}\boldsymbol{\Sigma}_1^{-1/2}$, for any integer $q \geq 1$,

$$\mathbb{E}[|Z_j|^q] \leq q\mathsf{C}_* \int_0^\infty t^{q-1}e^{-\mathsf{c}_x t}\mathrm{d}t = \frac{\mathsf{C}_* q!}{\mathsf{c}_*^q}.$$

Hence we can apply Bernstein's inequality for centered sub-exponential random variable and obtain

$$\mathbb{P}\left( \left| \frac{1}{p}\sum_{j \in [p]} Z_j \right| \geq \varepsilon/2 \right) \leq 2\exp\left\{ -c_* p \cdot \min(\varepsilon^2, \varepsilon) \right\}. \tag{123}$$

Following the proof of [Vershynin, 2010, Theorem 5.39], we deduce that there exist constants $C_*, C_{*,D} > 0$ such that with probability at least $1 - p^{-D}$,

$$\|\overline{\boldsymbol{F}}_1^{\mathsf{T}}\overline{\boldsymbol{F}}_1/p - \mathbf{I}_{k_1}\|_{\mathrm{op}} \leq C_*\sqrt{\frac{k_1}{p}} + C_{*,D}\sqrt{\frac{\log(p)}{p}}.$$

In particular, there exists $\eta_* \in (0, 1/4)$ and $C_{*,D} > 0$ such that for $p \geq C_{*,D}$ and via union bound, we have with probability at least $1 - p^{-D}$ (reparametrizing $D$), that for any $k_1 \leq \lfloor \eta_* \cdot p \rfloor$,

$$\|\overline{\boldsymbol{F}}_1^{\mathsf{T}}\overline{\boldsymbol{F}}_1/p - \mathbf{I}_{k_1}\|_{\mathrm{op}} \leq 1/2.$$

From Eq. (122), we deduce that the $k_1$-th eigenvalue of $\widehat{\boldsymbol{\Sigma}}_{\boldsymbol{F}}$ for $k_1 < \lfloor \eta_* \cdot p \rfloor$ is lower bounded by

$$\hat{\xi}_{k_1}^2 \geq \lambda_{\min}\left( \frac{\boldsymbol{F}_1\boldsymbol{F}_1^{\mathsf{T}}}{p} \right) \geq \xi_{k_1}^2 \cdot \lambda_{\min}\left( \frac{\overline{\boldsymbol{F}}_1\overline{\boldsymbol{F}}_1^{\mathsf{T}}}{p} \right) \geq \frac{\xi_{k_1}^2}{2}, \tag{124}$$

and

$$\lambda_{\max}\left( \frac{\boldsymbol{F}_1\boldsymbol{F}_1^{\mathsf{T}}}{p} \right) \leq \frac{3}{2}\xi_1^2 = \frac{3}{2}.$$

Similarly for $\overline{\boldsymbol{F}}_2$, we have with probability at least $1 - p^{-D}$ that for any $k_1 < k_2 \le \lfloor \eta_* \cdot p \rfloor$,

$$\|\overline{\boldsymbol{F}}_2^\mathsf{T} \overline{\boldsymbol{F}}_2 / p - \mathbf{I}_{k_2 - k_1}\|_{\mathrm{op}} \le 1/2,$$

and therefore

$$\lambda_{\max}\left(\frac{\boldsymbol{F}_2 \boldsymbol{F}_2^\mathsf{T}}{p}\right) \le \frac{3}{2}\xi_{k_1+1}^2. \tag{125}$$

**Step 2: Bounding the eigenvalues of $\boldsymbol{F}$.**

Equation (124) provides a lower bound on the eigenvalues $\hat{\xi}_k^2$ of $\widehat{\boldsymbol{\Sigma}}_{\boldsymbol{F}}$ up to $k < \lfloor \eta_* \cdot p \rfloor$. Let's upper bound the $p$ eigenvalues of $\widehat{\boldsymbol{\Sigma}}_{\boldsymbol{F}}$. From now on, set $k_2 = p_* - 1$ where $p_* := \lfloor \eta_* \cdot p \rfloor$.

For the contribution of $\|\boldsymbol{F}_+\|$, we use the matrix Bernstein's inequality as in [Misiakiewicz and Saeed, 2024, Lemma 1] (see proof of Theorem A.2 in Appendix A) and obtain that for $p \ge C_K$, with probability at least $1 - p^{-D}$,

$$\frac{1}{p}\|\boldsymbol{F}_3 \boldsymbol{F}_3^\mathsf{T}\|_{\mathrm{op}} \le C_{*,D,K} \cdot \xi_{p_*}^2 \left\{1 + \frac{r_{\boldsymbol{\Sigma}}(p_*) \vee p}{p} \log\left(r_{\boldsymbol{\Sigma}}(p_*) \vee p\right)\right\}.$$

By the min-max theorem, we have with probability at least $1 - p^{-D}$ for all $k_1 < p_* - 1$,

$$\hat{\xi}_{k_1+1}^2 \le \lambda_{\max}\left(\frac{\boldsymbol{F}_2 \boldsymbol{F}_2^\mathsf{T}}{p} + \frac{\boldsymbol{F}_3 \boldsymbol{F}_3^\mathsf{T}}{p}\right) \le \frac{3}{2}\xi_{k_1+1}^2 + C_{*,D,K} \cdot \xi_{p_*}^2 \left\{1 + \frac{r_{\boldsymbol{\Sigma}}(p_*) \vee p}{p} \log\left(r_{\boldsymbol{\Sigma}}(p_*) \vee p\right)\right\},$$

where we used Eq. (125). For $k \ge p_*$, we simply use that

$$\hat{\xi}_k^2 \le \frac{1}{p}\|\boldsymbol{F}_3 \boldsymbol{F}_3^\mathsf{T}\|_{\mathrm{op}} \le \frac{3}{2}\xi_k^2 + C_{*,D,K} \cdot \xi_{p_*}^2 \left\{1 + \frac{r_{\boldsymbol{\Sigma}}(p_*) \vee p}{p} \log\left(r_{\boldsymbol{\Sigma}}(p_*) \vee p\right)\right\}.$$

We deduce from the above two displays that with probability at least $1 - p^{-D}$, we have for any $k \le p$

$$\hat{\xi}_k^2 \le C_{*,K,D} \cdot \left\{\xi_k^2 + \xi_{p_*}^2 \cdot M_{\boldsymbol{\Sigma}}(p)\right\}, \tag{126}$$

where we recall that we defined

$$M_{\boldsymbol{\Sigma}}(k) := 1 + \frac{r_{\boldsymbol{\Sigma}}(\lfloor \eta_* \cdot k \rfloor) \vee k}{k} \log\left(r_{\boldsymbol{\Sigma}}(\lfloor \eta_* \cdot k \rfloor) \vee k\right).$$

**Step 3: Bounding $r_{\widehat{\boldsymbol{\Sigma}}_{\boldsymbol{F}}}(k)$ for $k \le p$.**

Recall that the intrinsic dimension $r_{\widehat{\boldsymbol{\Sigma}}_{\boldsymbol{F}}}(k)$ at level $k \le p$ is given by

$$r_{\widehat{\boldsymbol{\Sigma}}_{\boldsymbol{F}}}(k) = \frac{\sum_{j=k}^p \hat{\xi}_j^2}{\hat{\xi}_k^2}.$$

First note that for $p \ge k \ge \lfloor \eta_* \cdot p \rfloor$, we simply use that and the eigenvalues are nonincreasing to get

$$\frac{\sum_{j=k}^p \hat{\xi}_j^2}{\hat{\xi}_k^2} \le (p + 1 - k) \le C(1 - \eta_*)p \le C\frac{1 - \eta_*}{\eta_*}k.$$

For $k \le p_* - 1$, we use that from Eq. (124) with probability at least $1 - p^{-D}$,

$$\frac{\sum_{j=k}^p \hat{\xi}_j^2}{\hat{\xi}_k^2} \le \frac{2}{\xi_k^2}\left(\frac{3}{2}\sum_{j=k}^{\lfloor \eta_* \cdot p \rfloor - 1} \xi_k^2 + \sum_{j=\lfloor \eta_* \cdot p \rfloor}^p \hat{\xi}_j^2\right).$$

Let's bound the second term on the right-hand side. Notice that

$$\sum_{j=\lfloor \eta_* \cdot p \rfloor + 1}^p \hat{\xi}_j^2 = \min_{\boldsymbol{V}} \mathrm{Tr}(\widehat{\boldsymbol{\Sigma}}_{\boldsymbol{F}}(\mathbf{I} - \boldsymbol{V}\boldsymbol{V}^\mathsf{T})) \le \frac{1}{p}\mathrm{Tr}(\boldsymbol{F}_3 \boldsymbol{F}_3^\mathsf{T}),$$

where the minimization is over $\boldsymbol{V} \in \mathbb{R}^{p \times (\lfloor \eta_* \cdot p \rfloor - 1)}$ with $\boldsymbol{V}^\mathsf{T} \boldsymbol{V} = \boldsymbol{I}_{\lfloor \eta_* \cdot p \rfloor - 1}$ and the second inequality is obtained by taking $\boldsymbol{V}$ orthogonal to the matrix $[\boldsymbol{F}_1, \boldsymbol{F}_2]$. We can rewrite

$$\frac{1}{p} \mathrm{Tr}(\boldsymbol{F}_3 \boldsymbol{F}_3^\mathsf{T}) - \mathrm{Tr}(\boldsymbol{\Sigma}_3) = \frac{1}{p} \sum_{j \in [p]} \|\boldsymbol{f}_{3,j}\|_2^2 - \mathrm{Tr}(\boldsymbol{\Sigma}_3).$$

Introduce $Z_j := \|\boldsymbol{f}_{3,j}\|_2^2 - \mathrm{Tr}(\boldsymbol{\Sigma}_3)$. By Assumption B.1 with $\boldsymbol{B} = \mathrm{diag}(\boldsymbol{0}, \boldsymbol{I})$ (identity on the subspace $\boldsymbol{\Sigma}_3$ and $0$ otherwise), we have

$$\mathbb{E}[|Z_j|^q] \leq q\mathsf{C}_x \|\boldsymbol{\Sigma}_3\|_F^q \int_0^\infty t^{q-1} e^{-\mathsf{c}_x t} \mathrm{d}t \leq \mathsf{C}_x \cdot \mathrm{Tr}(\boldsymbol{\Sigma}_3)^q \frac{q!}{\mathsf{c}_x^q}.$$

We therefore we can apply Bernstein's inequality (123) again and we get

$$\mathbb{P}\left(\left|\frac{1}{p} \sum_{j \in [p]} Z_j\right| \geq t \cdot \mathrm{Tr}(\boldsymbol{\Sigma}_3)\right) \leq 2\exp(-c_* \min(pt^2, pt)).$$

Hence, for any $p \geq C_{*,D}$ with probability at least $1 - p^{-D}$,

$$\frac{1}{p} \mathrm{Tr}(\boldsymbol{F}_3 \boldsymbol{F}_3^\mathsf{T}) \leq 2\mathrm{Tr}(\boldsymbol{\Sigma}_3).$$

Combining the above display with the previous inequalities yields with probability at least $1 - p^{-D}$ that for any $k \leq \lfloor \eta_* \cdot p \rfloor$,

$$r_{\widehat{\boldsymbol{\Sigma}}_{\boldsymbol{F}}}(k) = \frac{\sum_{j=k}^p \hat{\xi}_j^2}{\hat{\xi}_k^2} \leq C \frac{\sum_{j=k}^\infty \xi_j^2}{\xi_k^2}.$$

We deduce that there exist a constant $C_* > 0$ such that with probability at least $1 - p^{-D}$, we have for any $n_* := \lfloor \eta_* \cdot n \rfloor \leq p$

$$r_{\widehat{\boldsymbol{\Sigma}}_{\boldsymbol{F}}}(n_*) \vee n \leq C_* \cdot r_{\boldsymbol{\Sigma}}(n_*) \vee n, \tag{127}$$

where $r_{\boldsymbol{\Sigma}}(n)$ is the effective rank associated to $\boldsymbol{\Sigma}$.

### Step 4: Concluding the proof.

For any $n_* = \lfloor \eta_* \cdot n \rfloor > p$, we have $\widehat{\rho}_\lambda(n) = \widetilde{\rho}_\lambda(n, p) = 1$. Hence, we only need to consider the case $n_* \leq p$. Using Eqs. (126) and (127), we get

$$\widehat{\rho}_\lambda(n) = 1 + \frac{n\hat{\xi}_{n_*}^2}{\lambda} \left\{ 1 + \frac{r_{\widehat{\boldsymbol{\Sigma}}_{\boldsymbol{F}}}(n_*) \vee n}{n} \log\left(r_{\widehat{\boldsymbol{\Sigma}}_{\boldsymbol{F}}}(n_*) \vee n\right) \right\}$$

$$\leq 1 + C_{*,D,K} \cdot \left\{ \frac{n\xi_{n_*}^2}{\lambda} + \frac{n}{p} \cdot \frac{p \cdot \xi_{p_*}}{\lambda} M_{\boldsymbol{\Sigma}}(p) \right\} M_{\boldsymbol{\Sigma}}(n)$$

$$\leq 1 + C_{*,D,K} \cdot \left\{ \frac{n\xi_{n_*}^2}{\lambda} + \frac{n}{p} \cdot \rho_\lambda(p) \right\} M_{\boldsymbol{\Sigma}}(n),$$

which concludes the proof. $\qquad\square$

### B.7.2 Concentration of the fixed points

*Proof of Proposition B.4.* Recall that $(\tilde{\nu}_1, \tilde{\nu}_2) \in \mathbb{R}_{>0}^2$ are the unique solution to the random fixed point equations (77). From the first equation, we have the following bounds on $\tilde{\nu}_1$:

$$\frac{p\lambda}{n} \leq p\tilde{\nu}_1 \leq \frac{p\|\widehat{\boldsymbol{\Sigma}}_{\boldsymbol{F}}\|_{\mathrm{op}} + p\lambda}{n}.$$

From Lemma B.2, we have with probability at least $1 - p^{-D}$ that $\|\widehat{\boldsymbol{\Sigma}}_{\boldsymbol{F}}\|_{\mathrm{op}} \leq C_{*,D,K} \leq p$. Hence, by the uniform concentration over $\kappa \in [p\lambda/n, (p^3 + p\lambda)/n]$ in Lemma B.15 stated below and an union bound, we deduce that with probability at least $1 - p^{-D}$,

$$\left|\mathrm{Tr}\left(\boldsymbol{F}\boldsymbol{F}^\mathsf{T}(\boldsymbol{F}\boldsymbol{F}^\mathsf{T} + p\tilde{\nu}_1)^{-1}\right) - \mathrm{Tr}\left(\boldsymbol{\Sigma}(\boldsymbol{\Sigma} + \tilde{\nu}_2)^{-1}\right)\right| \leq C_{*,K,D} \frac{\rho_{\gamma_+}(p)^{5/2} \log^3(p)}{\sqrt{p}} \mathrm{Tr}\left(\boldsymbol{\Sigma}(\boldsymbol{\Sigma} + \tilde{\nu}_2)^{-1}\right)$$

$$=: \widetilde{\mathcal{E}}_2 \cdot \mathrm{Tr}\left(\boldsymbol{\Sigma}(\boldsymbol{\Sigma} + \tilde{\nu}_2)^{-1}\right).$$

Therefore, we can rewrite the fixed equations (77) as

$$n - \frac{\lambda}{\tilde{\nu}_1} = \text{Tr}\big(\mathbf{\Sigma}(\mathbf{\Sigma} + \tilde{\nu}_2)^{-1}\big) \cdot (1 + \delta(\mathbf{F})),$$

$$p - \frac{p\tilde{\nu}_1}{\tilde{\nu}_2} = \text{Tr}\left(\mathbf{\Sigma}(\mathbf{\Sigma} + \tilde{\nu}_2)^{-1}\right).$$

where with probability at least $1 - p^{-D}$, we have $|\delta(\mathbf{F})| \leq \widetilde{\mathcal{E}}(p)$. Therefore, by condition (78) and $p \geq C_{*,D,K}$

$$C_* \widetilde{\mathcal{E}}(p) \cdot \widetilde{\rho}_\lambda(n, p) \leq \frac{C_{*,D,K}}{\log(p)} \leq \frac{1}{2},$$

and we can directly use Lemma B.17 stated below to obtain with probability at least $1 - p^{-D}$,

$$\frac{|\tilde{\nu}_1 - \nu_1|}{\nu_1} \leq C_* \cdot \mathcal{E}_2(p) \cdot \widetilde{\rho}_\lambda(n, p), \qquad \frac{|\tilde{\nu}_2 - \nu_2|}{\nu_2} \leq C_* \cdot \mathcal{E}_2(p) \cdot \widetilde{\rho}_\lambda(n, p).$$

This concludes the proof of this proposition. $\qquad\square$

For any $\kappa \geq 0$, denote $\nu_2(\kappa) \in \mathbb{R}_{>0}$ the unique positive solution to

$$p - \frac{\kappa}{\nu_2(\kappa)} = \text{Tr}\big(\mathbf{\Sigma}(\mathbf{\Sigma} + \nu_2(\kappa))^{-1}\big). \tag{128}$$

We will further define analogously to Eq. (80) the truncated fixed point

$$p - \frac{\gamma(\kappa)}{\nu_{2,0}(\kappa)} = \text{Tr}\big(\mathbf{\Sigma}_0(\mathbf{\Sigma}_0 + \nu_{2,0}(\kappa))^{-1}\big), \tag{129}$$

where we recall that we denoted $\gamma(\kappa) = \kappa + \text{Tr}(\mathbf{\Sigma}_+)$. It will be convenient to recall the notations

$$\widetilde{\Phi}_2(\mathbf{F}; \kappa) = \frac{1}{p}\text{Tr}\big(\mathbf{F}\mathbf{F}^\mathsf{T}(\mathbf{F}\mathbf{F}^\mathsf{T} + \kappa)^{-1}\big),$$

$$\widetilde{\Phi}_2(\mathbf{F}_0; \gamma(\kappa)) = \frac{1}{p}\text{Tr}\big(\mathbf{F}_0\mathbf{F}_0^\mathsf{T}(\mathbf{F}_0\mathbf{F}_0^\mathsf{T} + \gamma(\kappa))^{-1}\big),$$

and the deterministic equivalents

$$\Psi_2(\nu_2(\kappa)) = \frac{1}{p}\text{Tr}\big(\mathbf{\Sigma}(\mathbf{\Sigma} + \nu_2(\kappa))^{-1}\big),$$

$$\Psi_2(\nu_{2,0}(\kappa)) = \frac{1}{p}\text{Tr}\big(\mathbf{\Sigma}_0(\mathbf{\Sigma}_0 + \nu_{2,0}(\kappa))^{-1}\big).$$

The next lemma show that $\widetilde{\Phi}_2(\mathbf{F}, \kappa)$ concentrates on $\Psi_2(\nu_2(\kappa))$ uniformly on an interval of $\kappa$.

**Lemma B.15.** *Under the setting of Proposition B.4 and for any $D, K \geq 0$, there exist constants $\eta_* \in (0, 1/4)$ and $C_{*,D,K} > 0$ such that the following holds. For any $p \geq C_{*,D,K}$ and $\lambda > 0$, if it holds that*

$$\gamma_\lambda = \gamma(p\lambda/n) \geq p^{-K}, \qquad \rho_{\gamma_\lambda}(p)^{5/2}\log^{3/2}(p) \leq K\sqrt{p}, \tag{130}$$

*then with probability at least $1 - p^{-D}$, we have for any $\kappa \in [p\lambda/n, p(p^2 + \lambda)/n]$,*

$$\left|\widetilde{\Phi}_2(\mathbf{F}; \kappa) - \Psi_2(\nu_2(\kappa))\right| \leq C_{*,D,K}\frac{\rho_{\gamma(\kappa)}(p)^{5/2}\log^3(p)}{\sqrt{p}}\Psi_2(\nu_2(\kappa)). \tag{131}$$

*Proof of Lemma B.15.* Throughout the proof we assume that we are working on the event

$$\|\mathbf{F}_+\mathbf{F}_+^\mathsf{T} - \gamma(0)\mathbf{I}_p\|_{\text{op}} \leq C_{*,D}\frac{\log^3(p)}{\sqrt{p}}\gamma(p\lambda/n),$$

which happens with probability at least $1 - p^{-D}$ by Lemma B.3 via union bound. Note that $\gamma(\kappa) \geq \gamma(p\lambda/n)$ for all the $\kappa \in [p\lambda/n, p(p^2 + \lambda)/n]$. Furthermore, we assume that $p \geq C_{*,D}$ chosen large enough so that $\|\mathbf{F}_+\mathbf{F}_+^\mathsf{T} - \gamma(0)\mathbf{I}_p\|_{\text{op}} \leq 1/2 \cdot \gamma(p\lambda/n)$ so that

$$\mathbf{F}\mathbf{F}^\mathsf{T} + \kappa \succeq \frac{1}{2}\gamma(\kappa).$$

The proof will proceed via a standard union bound argument over $\kappa$ in the interval $[p\lambda/n, p(p^2+\lambda)/n]$. Let us first prove Eq. (131) for a fixed $\kappa$. We first simplify the functional by rewriting it as

$$\widetilde{\Phi}_2(\boldsymbol{F}; \kappa) = 1 - \frac{\kappa}{p}\mathrm{Tr}\big((\boldsymbol{F}\boldsymbol{F}^{\mathsf{T}} + \kappa)^{-1}\big)$$

$$= 1 - \frac{\kappa}{p}\mathrm{Tr}\big((\boldsymbol{F}_0\boldsymbol{F}_0^{\mathsf{T}} + \gamma(\kappa))^{-1}\big) + \frac{\kappa}{p}\Delta,$$

where we denoted

$$|\Delta| = \left|\mathrm{Tr}\big((\boldsymbol{F}\boldsymbol{F}^{\mathsf{T}} + \kappa)^{-1}\big) - \mathrm{Tr}\big((\boldsymbol{F}_0\boldsymbol{F}_0^{\mathsf{T}} + \gamma(\kappa))^{-1}\big)\right|$$

$$= \left|\mathrm{Tr}\big((\boldsymbol{F}\boldsymbol{F}^{\mathsf{T}} + \kappa)^{-1}(\boldsymbol{F}_1\boldsymbol{F}_1^{\mathsf{T}} - \gamma(0)\mathbf{I}_p)(\boldsymbol{F}_0\boldsymbol{F}_0^{\mathsf{T}} + \gamma(\kappa))^{-1}\big)\right|$$

$$\leq C_{*,D}\frac{\log^3(p)}{\sqrt{p}}\left|\mathrm{Tr}\big((\boldsymbol{F}_0\boldsymbol{F}_0^{\mathsf{T}} + \gamma(\kappa))^{-1}\big)\right|.$$

Using again the above identity, we rewrite

$$\frac{1}{p}\mathrm{Tr}\big((\boldsymbol{F}_0\boldsymbol{F}_0^{\mathsf{T}} + \gamma(\kappa))^{-1}\big) = \frac{1}{\gamma(\kappa)} - \frac{1}{\gamma(\kappa)}\widetilde{\Phi}_2(\boldsymbol{F}_0; \gamma(\kappa)).$$

We can now apply Eq. (55) in Theorem A.2: under the assumption of Proposition B.4 and recalling the conditions (130), we have with probability at least $1 - p^{-D}$,

$$\left|\widetilde{\Phi}_2(\boldsymbol{F}_0; \gamma(\kappa)) - \Psi_2(\nu_{2,0}(\kappa))\right| \leq C_{*,D,K}\frac{\rho_{\gamma(\kappa)}(p)^{5/2}\log^{3/2}(p)}{\sqrt{p}}\Psi_2(\nu_{2,0}(\kappa)).$$

Combining the above displays, we obtain that with probability at least $1 - p^{-D}$

$$\left|\widetilde{\Phi}_2(\boldsymbol{F}; \kappa) - \Psi_2(\nu_{2,0}(\kappa)) - \frac{\mathrm{Tr}(\boldsymbol{\Sigma}_+)}{p\nu_{2,0}}\right|$$

$$\leq \frac{\kappa}{\gamma(\kappa)}\left|\widetilde{\Phi}_2(\boldsymbol{F}_0; \gamma(\kappa)) - \Psi_2(\nu_{2,0}(\kappa))\right| + \kappa|\Delta|$$

$$\leq C_{*,D,K}\frac{\rho_{\gamma(\kappa)}(p)^{5/2}\log^{3/2}(p)}{\sqrt{p}}\Psi_2(\nu_{2,0}(\kappa)) + C_{*,D}\frac{\log^3(p)}{\sqrt{p}}\left|\frac{\kappa}{p}\mathrm{Tr}\big((\boldsymbol{F}_0\boldsymbol{F}_0^{\mathsf{T}} + \gamma(\kappa))^{-1}\big)\right|,$$

where we used in the first inequality identity (129) to get

$$\left(1 - \frac{\kappa}{\gamma(\kappa)}\right)\left(1 - \frac{1}{p}\mathrm{Tr}(\boldsymbol{\Sigma}_0(\boldsymbol{\Sigma}_0 + \nu_{2,0})^{-1})\right) = \frac{\mathrm{Tr}(\boldsymbol{\Sigma}_+)}{p\nu_{2,0}}.$$

Further note that using condition (130), we can simplify the right-hand side

$$\left|\kappa\mathrm{Tr}\big((\boldsymbol{F}_0\boldsymbol{F}_0^{\mathsf{T}} + \gamma(\kappa))^{-1}\big)\right| \leq \frac{\kappa}{\gamma(\kappa)}\left|1 - \Psi_2(\nu_{2,0}(\kappa))\right| + C_{*,D,K}\cdot K\cdot\Psi_2(\nu_{2,0}(\kappa))$$

$$\leq C_{*,D,K}\left\{\Psi_2(\nu_{2,0}(\kappa)) + \frac{\mathrm{Tr}(\boldsymbol{\Sigma}_+)}{p\nu_{2,0}}\right\}.$$

We can now use Eq. (82) in Lemma B.5 applied to $\nu_2(\kappa)$ and $\nu_{2,0}(\kappa)$ to concluded that with probability at least $1 - p^{-D}$,

$$\left|\widetilde{\Phi}_2(\boldsymbol{F}; \kappa) - \Psi_2(\nu_2(\kappa))\right| \leq \widetilde{\mathcal{E}}(p)\cdot\Psi_2(\nu_2(\kappa)), \qquad \widetilde{\mathcal{E}}(p) := C_{*,D,K}\frac{\rho_{\gamma(\kappa)}(p)^{5/2}\log^3(p)}{\sqrt{p}}. \tag{132}$$

We now consider a $p^{-P}$-grid $\mathcal{P}_n$ of the interval $\kappa \in [p\lambda/n, p(p^2 + \lambda)/n]$, which contains at most $p^{P+3}$ points. We can use a union bound over $\kappa \in \mathcal{P}_n$ and reparametrizing $D' = D + P + 3$, so that with probability at least $1 - p^{-D}$, Equation (132) holds for any $\kappa \in \mathcal{P}_n$. Then for any point $\kappa_1 \in [p\lambda/n, p(Kp^2 + \lambda)/n]$, denote $\kappa_0 \in \mathcal{P}_n$ its closest point. Then by Lemma B.16 stated below, we have

$$\left|\widetilde{\Phi}_2(\boldsymbol{F}; \kappa_1) - \Psi_2(\nu_2(\kappa_1))\right|$$

$$\leq \left|\widetilde{\Phi}_2(\boldsymbol{F}; \kappa_1) - \widetilde{\Phi}_2(\boldsymbol{F}; \kappa_0)\right| + \left|\widetilde{\Phi}_2(\boldsymbol{F}; \kappa_0) - \Psi_2(\nu_2(\kappa_0))\right| + |\Psi_2(\nu_2(\kappa_0)) - \Psi_2(\nu_2(\kappa_1))|$$

$$\leq p^{CK}|\kappa_1 - \kappa_0|\widetilde{\Phi}_2(\boldsymbol{F}; \kappa_0) + \widetilde{\mathcal{E}}(p)\cdot\Psi_2(\nu_2(\kappa_0)) + p^{CK}|\kappa_1 - \kappa_0|\Psi_2(\nu_2(\kappa_1))$$

$$\leq \left(p^{CK-P} + \widetilde{\mathcal{E}}(p)\right)\left(1 + p^{CK-P} + \widetilde{\mathcal{E}}(p)\right)\cdot\Psi_2(\nu_2(\kappa_1)).$$

where we used that $|\kappa_1 - \kappa_0| \leq p^{-P}$. Taking $P = CK + 1$ concludes the proof. $\qquad\square$

**Lemma B.16.** *Under the setting of Proposition B.4 and for any $D, K \geq 0$, there exist constants $\eta_* \in (0, 1/4)$, $C > 0$ and $C_{*,D} > 0$ such that the following holds. For any $p \geq 1$ and $\lambda > 0$, it holds that*

$$\gamma(p\lambda/n) = \frac{p\lambda}{n} + \mathrm{Tr}(\boldsymbol{\Sigma}_+) \geq p^{-K}, \tag{133}$$

*then for any $\kappa_0, \kappa_1 \geq p\lambda/n$, we have*

$$\left| \frac{\Psi_2(\nu_2(\kappa_1))}{\Psi_2(\nu_2(\kappa_0))} - 1 \right| \leq p^{CK} |\kappa_1 - \kappa_0|. \tag{134}$$

*Furthermore, if $p \geq C_{*,D}$, then we have with probability $1 - p^{-D}$ that for any $\kappa_1, \kappa_2 \geq p\lambda/n$,*

$$\left| \frac{\widetilde{\Phi}_2(\boldsymbol{F}; \kappa_1)}{\widetilde{\Phi}_2(\boldsymbol{F}; \kappa_0)} - 1 \right| \leq p^{CK} |\kappa_1 - \kappa_0|. \tag{135}$$

*Proof of Lemma B.16.* Using the identity (128), we can decompose the first difference into

$$
\left| \frac{\Psi_2(\nu_2(\kappa_1))}{\Psi_2(\nu_2(\kappa_0))} - 1 \right| = \frac{\left| \frac{\kappa_0}{\nu_2(\kappa_0)} - \frac{\kappa_1}{\nu_2(\kappa_1)} \right|}{p \Psi_2(\nu_2(\kappa_0))}
$$

$$
\leq \frac{1}{\nu_2(\kappa_0) \cdot p \Psi_2(\nu_2(\kappa_0))} \left\{ |\kappa_1 - \kappa_0| + \left| \frac{\nu_2(\kappa_0)}{\nu_2(\kappa_1)} - 1 \right| \right\}.
$$

From condition (133) and the proof of [Misiakiewicz and Saeed, 2024, Lemma 11], we have

$$
\frac{1}{\nu_2(\kappa_0) \cdot p \Psi_2(\nu_2(\kappa_0))} \leq p^{3+2K},
$$

$$
\left| \frac{\nu_2(\kappa_0)}{\nu_2(\kappa_1)} - 1 \right| \leq p^{2+2K} |\kappa_1 - \kappa_0|.
$$

Combining the above inequalities, we deduce that there exists a constant $C > 0$ such that

$$
\left| \frac{\Psi_2(\nu_2(\kappa_1))}{\Psi_2(\nu_2(\kappa_0))} - 1 \right| \leq p^{CK} |\kappa_1 - \kappa_0|.
$$

For the second inequality (135), we rewrite the difference as

$$
\left| \frac{\widetilde{\Phi}_2(\boldsymbol{F}; \kappa_1)}{\widetilde{\Phi}_2(\boldsymbol{F}; \kappa_0)} - 1 \right| = \frac{\mathrm{Tr}(\boldsymbol{F}\boldsymbol{F}^{\mathsf{T}}(\boldsymbol{F}\boldsymbol{F}^{\mathsf{T}} + \kappa_1)^{-1}(\boldsymbol{F}\boldsymbol{F}^{\mathsf{T}} + \kappa_0)^{-1})}{\mathrm{Tr}(\boldsymbol{F}\boldsymbol{F}^{\mathsf{T}}(\boldsymbol{F}\boldsymbol{F}^{\mathsf{T}} + \kappa_0)^{-1})} |\kappa_1 - \kappa_0|
$$

$$
\leq \|(\boldsymbol{F}\boldsymbol{F}^{\mathsf{T}} + \kappa_1)^{-1}\|_{\mathrm{op}} |\kappa_1 - \kappa_0|.
$$

Hence, recalling Lemma B.3 and condition (133), for any $p \geq C_{*,D}$, we get with probability at least $1 - p^{-D}$ and for all $\kappa_1, \kappa_2 \geq p\lambda/n$,

$$
\left| \frac{\widetilde{\Phi}_2(\boldsymbol{F}; \kappa_1)}{\widetilde{\Phi}_2(\boldsymbol{F}; \kappa_0)} - 1 \right| \leq \frac{2}{\gamma(\kappa_1)} |\kappa_1 - \kappa_0| \leq n^K |\kappa_1 - \kappa_0|,
$$

which concludes the proof of this lemma. $\qquad\square$

We consider $(\nu_1^\varepsilon, \nu_2^\varepsilon) \in \mathbb{R}_{\geq 0}$ the unique positive solutions of the perturbed equations:

$$
n - \frac{\lambda}{\nu_1^\varepsilon} = \mathrm{Tr}\left( \boldsymbol{\Sigma}(\boldsymbol{\Sigma} + \nu_2^\varepsilon)^{-1} \right)(1 + \varepsilon), \tag{136}
$$

$$
p - \frac{p\nu_1^\varepsilon}{\nu_2^\varepsilon} = \mathrm{Tr}\left( \boldsymbol{\Sigma}(\boldsymbol{\Sigma} + \nu_2^\varepsilon)^{-1} \right). \tag{137}
$$

**Lemma B.17.** *Let $\eta_* \in (0, 1/4)$ be chosen as in Proposition B.4. Then there exists $C_*, C'_* > 0$ such that for any $\varepsilon \in \mathbb{R}$ with*

$$|\varepsilon| \cdot C_* \cdot \widetilde{\rho}_\lambda(n, p) \leq \frac{1}{2},$$

*then*

$$\left| \frac{\nu_1^\varepsilon - \nu_1^0}{\nu_1^0} \right| \leq C'_* \cdot \widetilde{\rho}_\lambda(n, p) \cdot |\varepsilon|, \qquad\qquad \left| \frac{\nu_2^\varepsilon - \nu_2^0}{\nu_2^0} \right| \leq C'_* \cdot \widetilde{\rho}_\lambda(n, p) \cdot |\varepsilon|.$$

*Proof of Lemma B.17.* For convenience, we introduce the notations

$$\delta_1 := \frac{\nu_1^\varepsilon - \nu_1^0}{\nu_1^\varepsilon}, \qquad\qquad \delta_2 := \frac{\nu_2^\varepsilon - \nu_2^0}{\nu_2^\varepsilon}.$$

We first consider the second equation (137) and subtract the identities for $(\nu_1^\varepsilon, \nu_2^\varepsilon)$ and $(\nu_1^0, \nu_2^0)$ to obtain

$$p \left( \frac{\nu_1^0}{\nu_2^0} - \frac{\nu_1^\varepsilon}{\nu_2^\varepsilon} \right) + \mathrm{Tr}\left( \mathbf{\Sigma}(\mathbf{\Sigma} + \nu_2^0)^{-1} \right) - \mathrm{Tr}\left( \mathbf{\Sigma}(\mathbf{\Sigma} + \nu_2^\varepsilon)^{-1} \right)$$

$$= p \left[ \frac{\nu_1^0}{\nu_2^0} \delta_2 - \frac{\nu_1^\varepsilon}{\nu_2^\varepsilon} \delta_1 \right] + \delta_2 \cdot \nu_2^\varepsilon \mathrm{Tr}(\mathbf{\Sigma}(\mathbf{\Sigma} + \nu_2^0)^{-1}(\mathbf{\Sigma} + \nu_2^\varepsilon)^{-1}) = 0.$$

Hence, we obtain the first identity

$$\delta_2 = \delta_1 \frac{(\nu_1^\varepsilon/\nu_2^\varepsilon)}{(\nu_1^0/\nu_2^0) + \frac{\nu_2^\varepsilon}{p}\mathrm{Tr}(\mathbf{\Sigma}(\mathbf{\Sigma} + \nu_2^0)^{-1}(\mathbf{\Sigma} + \nu_2^\varepsilon)^{-1})}. \tag{138}$$

We now turn to the first equation (136): we obtain similarly

$$\lambda \left( \frac{1}{\nu_1^0} - \frac{1}{\nu_1^\varepsilon} \right) = \mathrm{Tr}\left( \mathbf{\Sigma}(\mathbf{\Sigma} + \nu_2^\varepsilon)^{-1} \right) (1 + \varepsilon) - \mathrm{Tr}\left( \mathbf{\Sigma}(\mathbf{\Sigma} + \nu_2^0)^{-1} \right)$$

$$\implies \frac{\lambda}{\nu_1^0} \delta_1 = \delta_2 \cdot \nu_2^\varepsilon \mathrm{Tr}(\mathbf{\Sigma}(\mathbf{\Sigma} + \nu_2^0)^{-1}(\mathbf{\Sigma} + \nu_2^\varepsilon)^{-1})(1 + \varepsilon) + \varepsilon \mathrm{Tr}\left( \mathbf{\Sigma}(\mathbf{\Sigma} + \nu_2^0)^{-1} \right).$$

Hence, rearranging the term and recalling the first identity (136), we get the second identity

$$\delta_1 \left[ \frac{\lambda}{\nu_1^0} + (1 + \varepsilon) \frac{\nu_2^\varepsilon \mathrm{Tr}(\mathbf{\Sigma}(\mathbf{\Sigma} + \nu_2^0)^{-1}(\mathbf{\Sigma} + \nu_2^\varepsilon)^{-1}) \cdot (\nu_1^\varepsilon/\nu_2^\varepsilon)}{(\nu_1^0/\nu_2^0) + \frac{\nu_2^\varepsilon}{p}\mathrm{Tr}(\mathbf{\Sigma}(\mathbf{\Sigma} + \nu_2^0)^{-1}(\mathbf{\Sigma} + \nu_2^\varepsilon)^{-1})} \right] = \varepsilon \mathrm{Tr}\left( \mathbf{\Sigma}(\mathbf{\Sigma} + \nu_2^0)^{-1} \right). \tag{139}$$

From this identity, we directly have

$$|\delta_1| \leq |\varepsilon| \cdot \frac{\nu_1^0}{\lambda} \mathrm{Tr}(\mathbf{\Sigma}(\mathbf{\Sigma} + \nu_2^0)^{-1}).$$

Note that for $n \geq p/\eta_*$, we simply have by Eq. (136) that

$$\frac{\nu_1^0}{\lambda} \mathrm{Tr}(\mathbf{\Sigma}(\mathbf{\Sigma} + \nu_2^0)^{-1}) = \frac{\mathrm{Tr}(\mathbf{\Sigma}(\mathbf{\Sigma} + \nu_2^0)^{-1})}{n - \mathrm{Tr}(\mathbf{\Sigma}(\mathbf{\Sigma} + \nu_2^0)^{-1})} \leq \frac{p}{n - p} \leq \frac{\eta_*}{1 - \eta_*},$$

where we use that $\mathrm{Tr}(\mathbf{\Sigma}(\mathbf{\Sigma} + \nu_2^0)^{-1}) \leq p$ by Eq. (137). For $n \leq p/\eta_*$, let's denote $\mu_1^0 = \lambda/\nu_1^0$. Rewriting Eq. (136), we get that

$$\mu_* = \frac{n}{1 + \mathrm{Tr}(\mathbf{\Sigma}(\mu_* \mathbf{\Sigma} + \mu_* \nu_2^0)^{-1})}.$$

Hence

$$\frac{\nu_1^0}{\lambda} \mathrm{Tr}(\mathbf{\Sigma}(\mathbf{\Sigma} + \nu_2^0)^{-1}) = \mathrm{Tr}(\mathbf{\Sigma}_{<\lfloor \eta_* \cdot n \rfloor}(\mu_1^0 \mathbf{\Sigma}_{<\lfloor \eta_* \cdot n \rfloor} + \mu_1^0 \nu_2^0)^{-1}) + \mathrm{Tr}(\mathbf{\Sigma}_{\geq \lfloor \eta_* \cdot n \rfloor}(\mu_1^0 \mathbf{\Sigma}_{\geq \lfloor \eta_* \cdot n \rfloor} + \mu_1^0 \nu_2^0)^{-1})$$

$$\leq \frac{\eta_* \cdot n}{\mu_1^0} + \frac{\mathrm{Tr}(\mathbf{\Sigma}_{\geq \lfloor \eta_* \cdot n \rfloor})}{\mu_1^0 \nu_2^0}$$

$$\leq \eta_* \left\{ 1 + \mathrm{Tr}(\mathbf{\Sigma}(\mu_* \mathbf{\Sigma} + \mu_* \nu_2^0)^{-1}) \right\} + \frac{\mathrm{Tr}(\mathbf{\Sigma}_{\geq \lfloor \eta_* \cdot n \rfloor})}{\lambda},$$

where we use in the last inequality that $\nu_1^0/\nu_2^0 \leq 1$ and the definition of the effective rank. Rearranging the terms we obtain

$$\frac{\nu_1^0}{\lambda} \mathrm{Tr}(\mathbf{\Sigma}(\mathbf{\Sigma} + \nu_2^0)^{-1}) \leq \frac{1}{1 - \eta_*} \left\{ 1 + \frac{\mathrm{Tr}(\mathbf{\Sigma}_{\geq \lfloor \eta_* \cdot n \rfloor})}{\lambda} \right\}.$$

Combining the above bounds we deduce that

$$|\delta_1| \leq C_* |\varepsilon| \cdot \left\{ 1 + \mathbb{1}_{n \leq p/\eta_*} \frac{\mathrm{Tr}(\mathbf{\Sigma}_{\geq \lfloor \eta_* \cdot n \rfloor})}{\lambda} \right\} \leq C_* \cdot \widetilde{\rho}_\lambda(n, p) \cdot |\varepsilon|.$$

By assumption $C_* \cdot \widetilde{\rho}_\lambda(n, p) \cdot |\varepsilon| \leq 1/2$ and therefore $\nu_1^\varepsilon \leq 2\nu_1^0$. We conclude the first inequality

$$\left| \frac{\nu_1^\varepsilon - \nu_1^0}{\nu_1^0} \right| \leq \frac{\nu_1^\varepsilon}{\nu_1^0} |\delta_1| \leq C_* \cdot \widetilde{\rho}_\lambda(n, p) \cdot |\varepsilon|.$$

Recalling Eq. (138), we have directly

$$|\delta_2| \leq |\delta_1| \frac{\nu_1^\varepsilon \nu_2^0}{\nu_1^0 \nu_2^\varepsilon} \quad \Longrightarrow \quad \left| \frac{\nu_2^\varepsilon - \nu_2^0}{\nu_2^0} \right| \leq \left| \frac{\nu_1^\varepsilon - \nu_1^0}{\nu_1^0} \right|,$$

which concludes the proof. $\qquad\square$

### B.7.3 Proof of deterministic equivalents for functionals on Z

*Proof of Proposition B.6.* Recall that $\widehat{\rho}_\lambda(n)$ is defined in Eq. (72) and that for $\boldsymbol{F} \in \mathcal{A}_\mathcal{F}$, we have $\widehat{\rho}_\lambda(n) \leq C_{*,D,K} \widetilde{\rho}_\lambda(n, p)$ for all $n \in \mathbb{N}$. Under the assumptions of Proposition B.6, we can apply Theorem A.2 to $\boldsymbol{Z}$ conditional on $\boldsymbol{F}$ to obtain that with probability at least $1 - n^{-D}$,

$$\left| \Phi_2(\boldsymbol{Z}; \lambda) - \frac{p}{n} \widetilde{\Phi}_2(\boldsymbol{F}; p\tilde{\nu}_1) \right| \leq C_{*,D,K} \frac{\widetilde{\rho}_\lambda(n, p)^{5/2} \log^{3/2}(n)}{\sqrt{n}} \cdot \frac{p}{n} \widetilde{\Phi}_2(\boldsymbol{F}; p\tilde{\nu}_1),$$

$$\left| \Phi_3(\boldsymbol{Z}; \boldsymbol{A}, \lambda) - \left( \frac{n\tilde{\nu}_1}{\lambda} \right)^2 \widetilde{\Phi}_5(\boldsymbol{F}; \boldsymbol{A}, p\tilde{\nu}_1) \right| \leq C_{*,D,K} \frac{\widetilde{\rho}_\lambda(n, p)^6 \log^{5/2}(n)}{\sqrt{n}} \cdot \left( \frac{n\tilde{\nu}_1}{\lambda} \right)^2 \widetilde{\Phi}_5(\boldsymbol{F}; \boldsymbol{A}, p\tilde{\nu}_1),$$

$$\left| \Phi_4(\boldsymbol{Z}; \boldsymbol{A}, \lambda) - \widetilde{\Phi}_5(\boldsymbol{F}; \boldsymbol{A}, p\tilde{\nu}_1) \right| \leq C_{*,D,K} \frac{\widetilde{\rho}_\lambda(n, p)^6 \log^{3/2}(n)}{\sqrt{n}} \cdot \widetilde{\Phi}_5(\boldsymbol{F}; \boldsymbol{A}, p\tilde{\nu}_1),$$

where $\tilde{\nu}_1$ is the solution of the fixed point equation (77). We conclude the proof using Lemma B.18 stated below and that by condition (85), we have $C_{*,D,K} \cdot \widetilde{\rho}_\lambda(n, p) \mathcal{E}_\nu(p) \leq C_{*,D,K} K$. $\qquad\square$

*Proof of Proposition B.7.* Again, under the assumptions of the proposition and for $\boldsymbol{F} \in \mathcal{A}_\mathcal{F}$, we can apply [Misiakiewicz and Saeed, 2024, Lemma 10] to get

$$\left| \langle \boldsymbol{u}, \boldsymbol{Z}(\boldsymbol{Z}^\mathsf{T}\boldsymbol{Z} + \lambda)^{-1} \widehat{\boldsymbol{\Sigma}}_{\boldsymbol{F}} (\boldsymbol{Z}^\mathsf{T}\boldsymbol{Z} + \lambda)^{-1} \boldsymbol{Z}^\mathsf{T} \boldsymbol{u} \rangle - n\widetilde{\Phi}_5(\boldsymbol{F}; \boldsymbol{I}, p\tilde{\nu}_1) \right| \leq C_{*,D,K} \frac{\widetilde{\rho}_\lambda(n, p)^6 \log^{7/2}(n)}{\sqrt{n}},$$

$$\left| \langle \boldsymbol{u}, \boldsymbol{Z}(\boldsymbol{Z}^\mathsf{T}\boldsymbol{Z} + \lambda)^{-1} \widehat{\boldsymbol{\Sigma}}_{\boldsymbol{F}} (\boldsymbol{Z}^\mathsf{T}\boldsymbol{Z} + \lambda)^{-1} \boldsymbol{Z}^\mathsf{T} \boldsymbol{v} \rangle \right| \leq C_{*,D,K} \frac{\widetilde{\rho}_\lambda(n, p)^6 \log^{7/2}(n)}{\sqrt{n}} \cdot \frac{n\nu_1}{\lambda} \sqrt{\widetilde{\Phi}_5(\boldsymbol{F}; \overline{\boldsymbol{v}\boldsymbol{v}}^\mathsf{T}, p\tilde{\nu}_1)}.$$

We conclude by combining these inequalities with Lemma B.18. $\qquad\square$

**Lemma B.18.** *For $\boldsymbol{F} \in \mathcal{A}_\mathcal{F}$, we have*

$$\left| \widetilde{\Phi}_2(\boldsymbol{F}; p\tilde{\nu}_1) - \widetilde{\Phi}_2(\boldsymbol{F}; p\nu_1) \right| \leq C_{*,D,K} \cdot \mathcal{E}_\nu(p) \cdot \widetilde{\Phi}_2(\boldsymbol{F}; p\nu_1), \tag{140}$$

$$\left| \widetilde{\Phi}_5(\boldsymbol{F}; \boldsymbol{A}, p\tilde{\nu}_1) - \widetilde{\Phi}_5(\boldsymbol{F}; \boldsymbol{A}, p\nu_1) \right| \leq C_{*,D,K} \cdot \widetilde{\rho}_\lambda(n, p) \mathcal{E}_\nu(p) \cdot \widetilde{\Phi}_5(\boldsymbol{F}; \boldsymbol{A}, p\nu_1). \tag{141}$$

*and*

$$\left| \left( \frac{n\tilde{\nu}_1}{\lambda} \right)^2 \widetilde{\Phi}_5(\boldsymbol{F}; \boldsymbol{A}, p\tilde{\nu}_1) - \left( \frac{n\nu_1}{\lambda} \right)^2 \widetilde{\Phi}_5(\boldsymbol{F}; \boldsymbol{A}, p\nu_1) \right|$$

$$\leq C_{*,D,K} \cdot \widetilde{\rho}_\lambda(n, p) \mathcal{E}_\nu(p) \cdot \left( \frac{n\nu_1}{\lambda} \right)^2 \widetilde{\Phi}_5(\boldsymbol{F}; \boldsymbol{A}, p\nu_1). \tag{142}$$

*Proof of Lemma B.18.* For convenience, we denote $\kappa := \nu_1$, $\kappa' := \tilde{\nu}_1$, and $\Gamma := \widehat{\Sigma}_{\boldsymbol{F}}$. For Eq. (140), we simply use that

$$
\begin{aligned}
\left| p\widetilde{\Phi}_2(\boldsymbol{F}; p\tilde{\nu}_1) - p\widetilde{\Phi}_2(\boldsymbol{F}; p\nu_1) \right| &= \left| \mathrm{Tr}\left( \Gamma(\Gamma + \kappa')^{-1} \right) - \mathrm{Tr}\left( \Gamma(\Gamma + \kappa)^{-1} \right) \right| \\
&= |\kappa - \kappa'| \mathrm{Tr}\left( \Gamma(\Gamma + \kappa')^{-1}(\Gamma + \kappa)^{-1} \right) \\
&\leq \frac{|\kappa' - \kappa|}{\kappa'} \mathrm{Tr}\left( \Gamma(\Gamma + \kappa)^{-1} \right) \\
&\leq C_{*,D,K} \cdot \mathcal{E}_\nu(p) \cdot p\widetilde{\Phi}_2(\boldsymbol{F}; p\nu_1).
\end{aligned}
$$

Similarly, by simple algebra, we have

$$
\begin{aligned}
&\left| \widetilde{\Phi}_6(\boldsymbol{F}; \boldsymbol{A}, p\tilde{\nu}_1) - \widetilde{\Phi}_6(\boldsymbol{F}; \boldsymbol{A}, p\nu_1) \right| \\
&= \left| \mathrm{Tr}\left( \boldsymbol{A}\Gamma^2(\Gamma + \kappa')^{-2} \right) - \mathrm{Tr}\left( \boldsymbol{A}\Gamma^2(\Gamma + \kappa)^{-2} \right) \right| \\
&\leq 2|\kappa - \kappa'| \mathrm{Tr}\left( \boldsymbol{A}\Gamma^3(\Gamma + \kappa')^{-2}(\Gamma + \kappa)^{-2} \right) + |\kappa^2 - (\kappa')^2| \mathrm{Tr}\left( \boldsymbol{A}\Gamma^3(\Gamma + \kappa')^{-2}(\Gamma + \kappa)^{-2} \right) \\
&\leq \frac{|\kappa - \kappa'|}{\kappa'} \left\{ 2 + \frac{|\kappa + \kappa'|}{\kappa'} \right\} \mathrm{Tr}\left( \boldsymbol{A}\Gamma^2(\Gamma + \kappa)^{-2} \right) \\
&\leq C_{*,D,K} \cdot \mathcal{E}_\nu(p) \cdot \widetilde{\Phi}_6(\boldsymbol{F}; \boldsymbol{A}, p\nu_1).
\end{aligned}
\tag{143}
$$

Furthermore, note that by the identity (77),

$$
\widetilde{\Phi}_6(\boldsymbol{F}; \boldsymbol{I}, p\tilde{\nu}_1) \left( n - \widetilde{\Phi}_6(\boldsymbol{F}; \boldsymbol{I}, p\tilde{\nu}_1) \right)^{-1} \leq \widetilde{\Phi}_2(\boldsymbol{F}; p\tilde{\nu}_1) \left( n - \widetilde{\Phi}_2(\boldsymbol{F}; p\tilde{\nu}_1) \right)^{-1} = \frac{\tilde{\nu}_1}{\lambda} \widetilde{\Phi}_2(\boldsymbol{F}; p\tilde{\nu}_1).
$$

Furthermore, we have from the previous computation and simple algebra that

$$
\frac{\tilde{\nu}_1}{\lambda} \widetilde{\Phi}_2(\boldsymbol{F}; p\tilde{\nu}_1) \leq (1 + C_{*,D,K}\mathcal{E}_\nu(p)) \cdot \frac{\nu_1}{\lambda} \mathrm{Tr}(\Sigma(\Sigma + \nu_2)^{-1}) \leq C_{*,D,K} \cdot \tilde{\rho}_\lambda(n, p),
$$

where we used the same argument as in the proof of Lemma B.17 in the last inequality as well as condition (85). The proof of Eqs. (142) and (141) from simple algebra from the above displays and (143). $\qquad \square$

### B.7.4  Proof of Lemma B.11

*Proof of Lemma B.11.* For convenience, we introduce the notations

$$
\boldsymbol{G}_{\boldsymbol{F}} := (\boldsymbol{F}\boldsymbol{F}^{\mathsf{T}} + p\nu_1)^{-1}, \qquad \boldsymbol{G}_0 := (\boldsymbol{F}_0\boldsymbol{F}_0^{\mathsf{T}} + \gamma_+)^{-1}, \qquad \boldsymbol{R}_0 := (\boldsymbol{F}_0^{\mathsf{T}}\boldsymbol{F}_0 + \gamma_+)^{-1}, \tag{144}
$$

where we recall that we denoted $\gamma_+ = \gamma(p\nu_1) = p\nu_1 + \mathrm{Tr}(\Sigma_+)$. Recall that for the first approximation guarantee, we denote

$$
\widetilde{\Phi}_2(\boldsymbol{F}; p\nu_1) = \frac{1}{p}\mathrm{Tr}(\boldsymbol{F}\boldsymbol{F}^{\mathsf{T}}\boldsymbol{G}_{\boldsymbol{F}}), \qquad \widetilde{\Phi}_2(\boldsymbol{F}_0; p\nu_1) = \frac{1}{p}\mathrm{Tr}(\boldsymbol{F}_0\boldsymbol{F}_0^{\mathsf{T}}\boldsymbol{G}_0),
$$

and the deterministic equivalents

$$
\Psi_2(\nu_2) = \frac{1}{p}\mathrm{Tr}(\Sigma(\Sigma + \nu_2)^{-1}), \qquad \Psi_2(\nu_{2,0}) = \frac{1}{p}\mathrm{Tr}(\Sigma_0(\Sigma_0 + \nu_{2,0})^{-1}).
$$

For the second approximation guarantee, recall that we denote

$$
\widetilde{\Phi}_4(\boldsymbol{F}; \Sigma^{-1}, p\nu_1) = \frac{1}{p}\mathrm{Tr}(\boldsymbol{F}\boldsymbol{F}^{\mathsf{T}}\boldsymbol{G}_{\boldsymbol{F}}^2), \qquad \widetilde{\Phi}_4(\boldsymbol{F}; \Sigma_0^{-1}, p\nu_1) = \frac{1}{p}\mathrm{Tr}(\boldsymbol{F}_0\boldsymbol{F}_0^{\mathsf{T}}\boldsymbol{G}_0^2),
$$

and their associated deterministic equivalents

$$
\Psi_3(\nu_2; \Sigma^{-1}) = \frac{1}{p} \cdot \frac{\mathrm{Tr}(\Sigma(\Sigma + \nu_2)^{-2})}{p - \mathrm{Tr}(\Sigma^2(\Sigma^2 + \nu_2)^{-2})}, \qquad \Psi_3(\nu_{2,0}; \Sigma_0^{-1}) = \frac{1}{p} \cdot \frac{\mathrm{Tr}(\Sigma_0(\Sigma_0 + \nu_{2,0})^{-2})}{p - \mathrm{Tr}(\Sigma_0^2(\Sigma_0^2 + \nu_{2,0})^{-2})}.
$$

We separate the analysis of the low-degree part $\boldsymbol{F}_0$ from the high-degree part $\boldsymbol{F}_+$. To remove the dependency on the high-degree part $\boldsymbol{F}_+$, we recall that by Lemma B.3, we have with probability at least $1 - p^{-D}$

$$
\|\boldsymbol{\Delta}\|_{\mathrm{op}} \leq C_{*,D} \frac{\log^3(p)}{\sqrt{p}}\gamma_+, \tag{145}
$$

where we denoted $\boldsymbol{\Delta} := \boldsymbol{F}_+\boldsymbol{F}_+ - \mathrm{Tr}(\boldsymbol{\Sigma}_+) \cdot \mathbf{I}_p$ and we recall that $\gamma_+ = \gamma(p\nu_1) = p\nu_1 + \mathrm{Tr}(\boldsymbol{\Sigma}_+)$. In particular, taking $p \geq C_{*,D}$, we have $\|\boldsymbol{\Delta}\|_{\mathrm{op}} \leq \frac{1}{2}\gamma_+$ and therefore

$$\|\boldsymbol{G_F}\|_{\mathrm{op}} \leq 2\|\boldsymbol{G}_0\|_{\mathrm{op}} \leq \frac{2}{\gamma_+}. \tag{146}$$

**Step 1: Bound on $|\widetilde{\Phi}_2(\boldsymbol{F}; p\nu_1) - \Psi_2(\nu_2)|$.**

First note that we have the identity

$$\widetilde{\Phi}_2(\boldsymbol{F}; p\nu_1) = 1 - \nu_1 \mathrm{Tr}(\boldsymbol{G_F}) = 1 - \nu_1 \mathrm{Tr}(\boldsymbol{G}_0) + \nu_1\Theta,$$

where we denoted $\Theta = \mathrm{Tr}(\boldsymbol{G}_0) - \mathrm{Tr}(\boldsymbol{G_F})$. By Eqs. (145) and (146), we have

$$|\Theta| = |\mathrm{Tr}(\boldsymbol{G_F}\boldsymbol{\Delta}\boldsymbol{G}_0)| \leq C_{*,D}\frac{\log^3(p)}{\sqrt{p}} \cdot \mathrm{Tr}(\boldsymbol{G}_0). \tag{147}$$

Using again the above identity, we have

$$\frac{1}{p}\mathrm{Tr}(\boldsymbol{G}_0) = \frac{1}{\gamma_+} - \frac{1}{\gamma_+}\widetilde{\Phi}_2(\boldsymbol{F}_0; p\nu_1). \tag{148}$$

Under the assumption of the lemma, we can apply Proposition B.8 and obtain with probability at least $1 - p^{-D}$ that

$$\left|\widetilde{\Phi}_2(\boldsymbol{F}_0; p\nu_1) - \Psi_2(\nu_{2,0})\right| \leq C_{*,D,K} \cdot \mathcal{E}_3(p) \cdot \Psi_2(\nu_{2,0}), \tag{149}$$

where $\mathcal{E}_3(p)$ is defined in Eq. (100). Furthermore note that using identity (80), we have

$$\frac{\mathrm{Tr}(\boldsymbol{\Sigma}_+)}{\gamma_+} + \frac{p\nu_1}{\gamma_+}\Psi_2(\nu_{2,0}) = \Psi_2(\nu_{2,0}) + \frac{\mathrm{Tr}(\boldsymbol{\Sigma}_+)}{p\nu_{2,0}},$$

and that by Eq. (82) in Lemma B.5,

$$\left|\Psi_2(\nu_{2,0}) + \frac{\mathrm{Tr}(\boldsymbol{\Sigma}_+)}{p\nu_{2,0}} - \Psi_2(\nu_2)\right| \leq \frac{C}{p}\Psi_2(\nu_2). \tag{150}$$

Thus combining Eqs. (147), (149) and (150), we obtain

$$|\widetilde{\Phi}_2(\boldsymbol{F}; p\nu_1) - \Psi_2(\nu_2)| \leq \nu_1|\Theta| + \frac{p\nu_1}{\gamma_+}\left|\widetilde{\Phi}_2(\boldsymbol{F}_0; p\nu_1) - \Psi_2(\nu_{2,0})\right| + \left|\Psi_2(\nu_{2,0}) + \frac{\mathrm{Tr}(\boldsymbol{\Sigma}_+)}{p\nu_{2,0}} - \Psi_2(\nu_2)\right|$$

$$\leq C_{*,D,K}\left\{\frac{\log^3(p)}{\sqrt{p}} + \mathcal{E}_3(p) + \frac{1}{p}\right\}\left[\nu_1\mathrm{Tr}(\boldsymbol{G}_0) + \frac{p\nu_1}{\gamma_+}\Psi_2(\nu_{2,0}) + \Psi_2(\nu_2)\right],$$

with probability at least $1 - p^{-D}$ via union bound. Using condition (106), we can simplify the right-hand side using identity (148) and bounds (149) and (150) to get

$$\kappa_1\mathrm{Tr}(\boldsymbol{G}_0) \leq \frac{p\nu_1}{\gamma_+}|1 - \Psi_2(\nu_{2,0})| + C_{*,D,K}K\Psi_2(\nu_{2,0})$$

$$\leq C_{*,D,K}\left\{\Psi_2(\nu_{2,0}) + \frac{\mathrm{Tr}(\boldsymbol{\Sigma}_+)}{p\nu_{2,0}}\right\} \leq C_{*,D,K} \cdot \Psi_2(\nu_2).$$

This concludes the proof of the first part of this lemma.

**Step 2: Bound on $|\widetilde{\Phi}_4(\boldsymbol{F}; \boldsymbol{\Sigma}^{-1}, p\nu_1) - \Psi_3(\nu_2; \boldsymbol{\Sigma}^{-1})|$.**

We proceed similarly as in the first part and omit some repetitive details. First note that we can rewrite

$$\begin{aligned}\widetilde{\Phi}_4(\boldsymbol{F}; \boldsymbol{\Sigma}^{-1}, p\nu_1) &= \frac{1}{p}\mathrm{Tr}(\boldsymbol{G_F}) - \nu_1\mathrm{Tr}(\boldsymbol{G_F^2})\\&= \frac{1}{p}\mathrm{Tr}(\boldsymbol{G}_0) - \nu_1\mathrm{Tr}(\boldsymbol{G}_0^2) + \Theta\\&= \frac{\mathrm{Tr}(\boldsymbol{\Sigma}_+)}{\gamma_+^2}\left(1 - \widetilde{\Phi}_2(\boldsymbol{F}_0; p\nu_1)\right) + \frac{p\nu_1}{\gamma_+}\widetilde{\Phi}_4(\boldsymbol{F}_0; \boldsymbol{\Sigma}_0^{-1}, p\nu_1) + \Theta,\end{aligned}$$

where

$$|\Theta| = \frac{1}{p} \left| \text{Tr}(\boldsymbol{G_F}) - \text{Tr}(\boldsymbol{G}_0) + p\nu_1 \text{Tr}(\boldsymbol{G}_0^2) - p\nu_1 \text{Tr}(\boldsymbol{G_F}^2) \right|$$

$$\leq C_{*,D} \frac{\log^3(p)}{\sqrt{p}} \cdot \left[ \frac{1}{p} \text{Tr}(\boldsymbol{G}_0) + \nu_1 \text{Tr}(\boldsymbol{G}_0^2) \right]. \tag{151}$$

Again, by Proposition B.8, we get that with probability at least $1 - p^{-D}$ that

$$\left| \widetilde{\Phi}_2(\boldsymbol{F}_0; p\nu_1) - \Psi_2(\nu_{2,0}) \right| \leq C_{*,D,K} \cdot \mathcal{E}_3(p) \cdot \Psi_2(\nu_{2,0}),$$

$$\left| \widetilde{\Phi}_4(\boldsymbol{F}_0; \boldsymbol{\Sigma}_0^{-1}, p\nu_1) - \Psi_3(\nu_{2,0}; \boldsymbol{\Sigma}_0^{-1}) \right| \leq C_{*,D,K} \cdot \mathcal{E}_3(p) \cdot \Psi_3(\nu_{2,0}; \boldsymbol{\Sigma}_0^{-1}). \tag{152}$$

Furthermore, note that by Lemma B.19 stated below, we have

$$\left| \frac{\text{Tr}(\boldsymbol{\Sigma}_+)}{\gamma_+^2} (1 - \Psi_2(\nu_{2,0})) + \frac{p\nu_1}{\gamma_+} \Psi_3(\nu_{2,0}; \boldsymbol{\Sigma}_0^{-1}) - \Psi_3(\nu_2; \boldsymbol{\Sigma}^{-1}) \right| \leq C \frac{\rho_{\gamma_+}(p)}{p} \Psi_3(\nu_2; \boldsymbol{\Sigma}^{-1}). \tag{153}$$

Combining Eqs. (151), (152) and (153), we deduce via union bound that with probability at least $1 - p^{-D}$

$$|\widetilde{\Phi}_4(\boldsymbol{F}; \boldsymbol{\Sigma}^{-1}, p\nu_1) - \Psi_3(\nu_2; \boldsymbol{\Sigma}^{-1})|$$

$$\leq C_{*,D,K} \cdot \mathcal{E}_3(p) \left[ \frac{1}{p} \text{Tr}(\boldsymbol{G}_0) + \nu_1 \text{Tr}(\boldsymbol{G}_0^2) + \frac{\text{Tr}(\boldsymbol{\Sigma}_+)^2}{\gamma_+^2} \Psi_2(\nu_{2,0}) + \frac{p\nu_1}{\gamma_+} \Psi_3(\nu_{2,0}; \boldsymbol{\Sigma}_0^{-1}) + \Psi_3(\nu_2; \boldsymbol{\Sigma}^{-1}) \right].$$

Let us simplify the right hand side. First, from the proof of Lemma B.19, we have

$$\frac{\text{Tr}(\boldsymbol{\Sigma}_+)^2}{\gamma_+^2} \Psi_2(\nu_{2,0}) + \frac{p\nu_1}{\gamma_+} \Psi_3(\nu_{2,0}; \boldsymbol{\Sigma}_0^{-1}) \leq C\rho_{\gamma_+}(p) \cdot \Psi_3(\nu_2; \boldsymbol{\Sigma}^{-1}).$$

Combining the above two displays with $\mathcal{E}_3(p) \leq K$ from conditions (106) yields

$$\frac{1}{p} \text{Tr}(\boldsymbol{G}_0) - \nu_1 \text{Tr}(\boldsymbol{G}_0^2) \leq C_{*,D,K} \cdot \rho_{\gamma_+}(p) \cdot \Psi_3(\nu_2; \boldsymbol{\Sigma}^{-1}).$$

Finally, note that using Eq. (152) and again $\mathcal{E}_3(p) \leq K$ that

$$\nu_1 \text{Tr}(\boldsymbol{G}_0^2) \leq \frac{p\nu_1}{\gamma_+} \widetilde{\Phi}_4(\boldsymbol{F}; \boldsymbol{\Sigma}_0^{-1/2}, p\nu_1) \leq C_{*,D,K} \Psi_3(\nu_2; \boldsymbol{\Sigma}^{-1}),$$

which concludes the proof of this lemma. $\qquad \square$

**Lemma B.19.** *Assuming that $p^2 \xi_{\mathsf{m}}^2 \leq \gamma_+$, we have*

$$\left| \frac{\text{Tr}(\boldsymbol{\Sigma}_+)}{\gamma_+^2} (1 - \Psi_2(\nu_{2,0})) + \frac{p\nu_1}{\gamma_+} \Psi_3(\nu_{2,0}; \boldsymbol{\Sigma}_0^{-1}) - \Psi_3(\nu_2; \boldsymbol{\Sigma}^{-1}) \right| \leq C \frac{\rho_{\gamma_+}(p)}{p} \Psi_3(\nu_2; \boldsymbol{\Sigma}^{-1}).$$

*Proof of Lemma B.19.* For convenience, we introduce

$$\Upsilon(\nu_{2,0}) := \frac{1}{p} \text{Tr}(\boldsymbol{\Sigma}_0^2 (\boldsymbol{\Sigma}_0 + \nu_{2,0})^{-2}), \qquad \Upsilon(\nu_2) := \frac{1}{p} \text{Tr}(\boldsymbol{\Sigma}^2 (\boldsymbol{\Sigma} + \nu_2)^{-2}).$$

Using that

$$\frac{1}{p} \text{Tr}(\boldsymbol{\Sigma}_0 (\boldsymbol{\Sigma}_0 + \nu_{2,0})^{-2}) = \frac{1}{\nu_{2,0}} \left\{ \Psi_2(\nu_{2,0}) - \Upsilon(\nu_{2,0}) \right\},$$

we obtain by simple algebra and using identity (80) that

$$\frac{\text{Tr}(\boldsymbol{\Sigma}_+)}{\gamma_+^2} (1 - \Psi_2(\nu_{2,0})) + \frac{p\nu_1}{\gamma_+} \Psi_3(\nu_{2,0}; \boldsymbol{\Sigma}_0^{-1}) = \frac{1}{\nu_{2,0}} \frac{\Psi_2(\nu_{2,0}) + \frac{\text{Tr}(\boldsymbol{\Sigma}_+)}{p\nu_{2,0}} - \Upsilon(\nu_{2,0})}{p(1 - \Upsilon(\nu_{2,0}))}.$$

Following the proof of [Misiakiewicz and Saeed, 2024, Lemma 7], we get that

$$|\Upsilon(\nu_{2,0}) - \Upsilon(\nu_2)| \leq 10 \frac{p\xi_{\mathsf{m}}^2}{\gamma_+} \leq \frac{C}{p},$$

$$(1 - \Upsilon(\nu_{2,0}))^{-1} \leq (1 - \Psi_2(\nu_{2,0}))^{-1} = \frac{p\nu_{2,0}}{\gamma_+} \leq C\rho_{\gamma_+}(p).$$

Recalling Eq. (82) in Lemma B.5, we can conclude the proof using simple algebraic manipulations similarly to [Misiakiewicz and Saeed, 2024, Lemma 7]. $\qquad \square$

### B.7.5 Proof of Lemma B.13

*Proof of Lemma B.13.* We will follow similar steps as in the proof of Lemma B.11. For the sake of brevity, we omit some repetitive details. Recall that we introduced the notations (144). We decompose the coefficient vector $\boldsymbol{\beta}_* = [\boldsymbol{\beta}_0, \boldsymbol{\beta}_+]$ with $\boldsymbol{\beta}_0 \in \mathbb{R}^m$ the first m coordinates of $\boldsymbol{\beta}_*$ (aligned with the top m eigenspaces), while $\boldsymbol{\beta}_+ \in \mathbb{R}^\infty$ corresponds to the rest of the coordinates. Further introduce the matrices

$$\boldsymbol{A}_* := \boldsymbol{\Sigma}^{-1/2}\boldsymbol{\beta}_*\boldsymbol{\beta}_*^\mathsf{T}\boldsymbol{\Sigma}^{-1/2}, \qquad \boldsymbol{A}_0 := \boldsymbol{\Sigma}_0^{-1/2}\boldsymbol{\beta}_0\boldsymbol{\beta}_0^\mathsf{T}\boldsymbol{\Sigma}_0^{-1/2},$$

and recall the expressions of the functionals

$$\widetilde{\Phi}_1(\boldsymbol{F}; \boldsymbol{A}_*, p\nu_1) = \mathrm{Tr}\big(\boldsymbol{A}_*\boldsymbol{\Sigma}^{1/2}\boldsymbol{R}\boldsymbol{\Sigma}^{1/2}\big), \qquad \widetilde{\Phi}_1(\boldsymbol{F}_0; \boldsymbol{A}_0, p\nu_1) = \mathrm{Tr}\big(\boldsymbol{A}_0\boldsymbol{\Sigma}_0^{1/2}\boldsymbol{R}_0\boldsymbol{\Sigma}_0^{1/2}\big),$$

as well as

$$\widetilde{\Phi}_4(\boldsymbol{F}; \boldsymbol{A}_*, p\nu_1) = \frac{1}{p}\mathrm{Tr}\big(\boldsymbol{A}_*\boldsymbol{\Sigma}^{1/2}\boldsymbol{R}\boldsymbol{F}^\mathsf{T}\boldsymbol{F}\boldsymbol{R}\boldsymbol{\Sigma}^{1/2}\big),$$

$$\widetilde{\Phi}_4(\boldsymbol{F}_0; \boldsymbol{A}_0, p\nu_1) = \frac{1}{p}\mathrm{Tr}\big(\boldsymbol{A}_0\boldsymbol{\Sigma}_0^{1/2}\boldsymbol{R}_0\boldsymbol{F}_0^\mathsf{T}\boldsymbol{F}_0\boldsymbol{R}_0\boldsymbol{\Sigma}_0^{1/2}\big).$$

We further recall the expressions of the deterministic equivalents

$$\Psi_1(\nu_2; \boldsymbol{A}_*) = \mathrm{Tr}\big(\boldsymbol{A}_*\boldsymbol{\Sigma}(\boldsymbol{\Sigma}+\nu_2)^{-1}\big), \qquad \Psi_1(\nu_{2,0}; \boldsymbol{A}_0) = \mathrm{Tr}\big(\boldsymbol{A}_0\boldsymbol{\Sigma}_0(\boldsymbol{\Sigma}_0+\nu_{2,0})^{-1}\big),$$

and

$$\psi_3(\nu_2; \boldsymbol{A}_*) = \frac{1}{p} \cdot \frac{\mathrm{Tr}(\boldsymbol{A}_*\boldsymbol{\Sigma}^2(\boldsymbol{\Sigma}+\nu_2)^{-2})}{p - \mathrm{Tr}(\boldsymbol{\Sigma}^2(\boldsymbol{\Sigma}+\nu_2)^{-2})},$$

$$\psi_3(\nu_{2,0}; \boldsymbol{A}_0) = \frac{1}{p} \cdot \frac{\mathrm{Tr}(\boldsymbol{A}_0\boldsymbol{\Sigma}_0^2(\boldsymbol{\Sigma}_0+\nu_{2,0})^{-2})}{p - \mathrm{Tr}(\boldsymbol{\Sigma}_0^2(\boldsymbol{\Sigma}_0+\nu_{2,0})^{-2})}.$$

We first rewrite our functionals into

$$(p\nu_1)^2\langle\boldsymbol{\beta}_*, \boldsymbol{R}^2\boldsymbol{\beta}_*\rangle = (p\nu_1)\langle\boldsymbol{\beta}_*, \boldsymbol{R}\boldsymbol{\beta}_*\rangle - (p\nu_1)\langle\boldsymbol{\beta}_*, \boldsymbol{R}\boldsymbol{F}^\mathsf{T}\boldsymbol{F}\boldsymbol{R}\rangle$$
$$= (p\nu_1)\left[\widetilde{\Phi}_1(\boldsymbol{F}; \boldsymbol{A}_*, p\nu_1) - p\widetilde{\Phi}_4(\boldsymbol{F}; \boldsymbol{A}_*, p\nu_1)\right],$$

and study each term separately.

**Step 1: Bounding term $\widetilde{\Phi}_1(\boldsymbol{F}; \boldsymbol{A}_*, p\nu_1)$.**

Let us start by removing the dependency on the high-degree part $\boldsymbol{F}_+$ in the denominator. Note that we can rewrite the matrix $\boldsymbol{R}$ in block matrix form

$$(p\nu_1)\boldsymbol{R} = (p\nu_1)\begin{pmatrix} \boldsymbol{F}_0^\mathsf{T}\boldsymbol{F}_0 + p\nu_1 & \boldsymbol{F}_0^\mathsf{T}\boldsymbol{F}_+ \\ \boldsymbol{F}_+^\mathsf{T}\boldsymbol{F}_0 & \boldsymbol{F}_+^\mathsf{T}\boldsymbol{F}_+ + p\nu_1 \end{pmatrix}^{-1} =: \begin{pmatrix} \widetilde{\boldsymbol{R}}_{00} & \widetilde{\boldsymbol{R}}_{0+} \\ \widetilde{\boldsymbol{R}}_{0+}^\mathsf{T} & \widetilde{\boldsymbol{R}}_{++} \end{pmatrix},$$

so that

$$(p\nu_1)\widetilde{\Phi}_1(\boldsymbol{F}; \boldsymbol{A}_*, p\nu_1) = \boldsymbol{\beta}_0^\mathsf{T}\widetilde{\boldsymbol{R}}_{00}\boldsymbol{\beta}_0 + 2\boldsymbol{\beta}_0\widetilde{\boldsymbol{R}}_{0+}\boldsymbol{\beta}_+ + \boldsymbol{\beta}_+^\mathsf{T}\widetilde{\boldsymbol{R}}_{++}\boldsymbol{\beta}_+.$$

Let us study each of these terms separately. Denote $\widetilde{\boldsymbol{G}}_+ := \boldsymbol{F}_+\boldsymbol{F}_+^\mathsf{T} + p\nu_1$. By simple algebra, we have

$$\boldsymbol{\beta}_0^\mathsf{T}\widetilde{\boldsymbol{R}}_{00}\boldsymbol{\beta}_0 = \boldsymbol{\beta}_0^\mathsf{T}\left(\boldsymbol{F}_0^\mathsf{T}\widetilde{\boldsymbol{G}}_+\boldsymbol{F}_0 + 1\right)^{-1}\boldsymbol{\beta}_0 = \gamma_+\widetilde{\Phi}_1(\boldsymbol{F}_0; \boldsymbol{A}_0, p\nu_1) + \Theta_{00},$$

where

$$|\Theta_{00}| = \left|\gamma_+\boldsymbol{\beta}_0^\mathsf{T}\boldsymbol{R}_0\boldsymbol{\beta}_0 - \boldsymbol{\beta}_0^\mathsf{T}\left(\boldsymbol{F}_0^\mathsf{T}\widetilde{\boldsymbol{G}}_+\boldsymbol{F}_0 + 1\right)^{-1}\boldsymbol{\beta}_0\right|$$

$$\leq C\left\|\boldsymbol{R}_0^{1/2}\boldsymbol{F}_0^\mathsf{T}(\boldsymbol{I} - \gamma_+\widetilde{\boldsymbol{G}}_+)\boldsymbol{F}_0\boldsymbol{R}_0^{1/2}\right\|_{\mathrm{op}} \cdot \gamma_+\boldsymbol{\beta}_0^\mathsf{T}\boldsymbol{R}_0\boldsymbol{\beta}_0$$

$$\leq C_{*,D}\frac{\log^3(p)}{\sqrt{p}}\gamma_+\widetilde{\Phi}_1(\boldsymbol{F}_0; \boldsymbol{A}_0, p\nu_1).$$

Furthermore, from Proposition B.8, we have with probability at least $1 - p^{-D}$ that

$$\left|\gamma_+\widetilde{\Phi}_1(\boldsymbol{F}_0; \boldsymbol{A}_0, p\nu_1) - \nu_{2,0}\Psi_1(\nu_{2,0}; \boldsymbol{A}_0)\right| \leq C_{*,D,K} \cdot \mathcal{E}_3(p) \cdot \nu_{2,0}\Psi_1(\nu_{2,0}; \boldsymbol{A}_0).$$

Hence, combining the previous two displays and using that $\mathcal{E}_3(p) \leq K$ by conditions (114), we have

$$\left| \boldsymbol{\beta}_0^\mathsf{T} \widetilde{\boldsymbol{R}}_{00} \boldsymbol{\beta}_0 - \nu_{2,0} \Psi_1(\nu_{2,0}; \boldsymbol{A}_0) \right| \leq C_{*,D,K} \cdot \left\{ \frac{\log^3(p)}{\sqrt{p}} + \mathcal{E}_3(p) \right\} \nu_{2,0} \Psi_1(\nu_{2,0}; \boldsymbol{A}_0). \qquad (154)$$

Similarly, denoting $\widetilde{\boldsymbol{G}}_0 = (\boldsymbol{F}_0 \boldsymbol{F}_0^\mathsf{T} + p\nu_1)^{-1}$, we can rewrite the third term as

$$\boldsymbol{\beta}_+^\mathsf{T} \widetilde{\boldsymbol{R}}_{++} \boldsymbol{\beta}_+ = \boldsymbol{\beta}_+^\mathsf{T} \left( \boldsymbol{F}_+^\mathsf{T} \widetilde{\boldsymbol{G}}_0 \boldsymbol{F}_+ + 1 \right)^{-1} \boldsymbol{\beta}_+ = \|\boldsymbol{\beta}_+\|_2^2 - \Theta_{++},$$

where with probability at least $1 - p^{-D}$,

$$\begin{aligned}
\Theta_{++} &= \boldsymbol{\beta}_+^\mathsf{T} \boldsymbol{F}_+^\mathsf{T} \widetilde{\boldsymbol{G}}_0^{1/2} \left( \widetilde{\boldsymbol{G}}_0^{1/2} \boldsymbol{F}_+ \boldsymbol{F}_+^\mathsf{T} \widetilde{\boldsymbol{G}}_0^{1/2} + 1 \right)^{-1} \widetilde{\boldsymbol{G}}_0^{1/2} \boldsymbol{F}_+ \boldsymbol{\beta}_+ \\
&\leq \|(\boldsymbol{F}_+ \boldsymbol{F}_+^\mathsf{T} + \widetilde{\boldsymbol{G}}_0^{-1})^{-1}\|_{\mathrm{op}} \cdot \|\boldsymbol{F}_+ \boldsymbol{\beta}_+\|_2^2 \\
&\leq \frac{C}{\gamma_+} \cdot p \|\boldsymbol{\Sigma}_+^{1/2} \boldsymbol{\beta}_+\|_2^2 \leq \frac{C}{p} \|\boldsymbol{\beta}_+\|_2^2,
\end{aligned} \qquad (155)$$

where we used the same concentration argument as in Step 3 of the proof of Lemma B.2.

Finally, we rewrite

$$\boldsymbol{\beta}_0^\mathsf{T} \widetilde{\boldsymbol{R}}_{0+} \boldsymbol{\beta}_+ = -\boldsymbol{\beta}_0^\mathsf{T} \left( \boldsymbol{F}_0^\mathsf{T} \widetilde{\boldsymbol{G}}_+ \boldsymbol{F}_0 + 1 \right)^{-1} \boldsymbol{F}_0^\mathsf{T} (\boldsymbol{F}_+ \boldsymbol{F}_+^\mathsf{T} + p\nu_1)^{-1} \boldsymbol{F}_+ \boldsymbol{\beta}_+.$$

Hence, using Eqs. (154) and (155) as well as conditions (114), we obtain

$$|\boldsymbol{\beta}_0^\mathsf{T} \widetilde{\boldsymbol{R}}_{0+} \boldsymbol{\beta}_+| \leq C_{*,D,K} \cdot \frac{1}{\sqrt{p}} \left\{ \nu_{2,0} \Psi_1(\nu_{2,0}; \boldsymbol{A}_0) + \|\boldsymbol{\beta}_+\|_2^2 \right\}. \qquad (156)$$

Finally, note that by Lemma B.20 stated below, we have

$$\left| \nu_{2,0} \langle \boldsymbol{\beta}_0, (\boldsymbol{\Sigma}_0 + \nu_{2,0})^{-1} \boldsymbol{\beta}_0 \rangle + \|\boldsymbol{\beta}_+\|_2^2 - \nu_2 \Psi_1(\nu_2; \boldsymbol{A}_*) \right| \leq \frac{C}{p} \nu_2 \Psi_1(\nu_2; \boldsymbol{A}_*). \qquad (157)$$

Combining Eqs. (154), (155), (156), and (157), we deduce that with probability at least $1 - p^{-D}$,

$$\left| (p\nu_1) \widetilde{\Phi}_1(\boldsymbol{F}; \boldsymbol{A}_*, p\nu_1) - \nu_2 \Psi_1(\nu_2; \boldsymbol{A}_*) \right| \leq C_{*,D,K} \cdot \left\{ \frac{\log^3(p)}{\sqrt{p}} + \mathcal{E}_3(p) \right\} \cdot \nu_2 \Psi_1(\nu_2; \boldsymbol{A}_*). \qquad (158)$$

**Step 2: Bounding term $\widetilde{\Phi}_4(\boldsymbol{F}; \boldsymbol{A}_*, p\nu_1)$.**

We rewrite this term as

$$\langle \boldsymbol{\beta}_*, \boldsymbol{R} \boldsymbol{F}^\mathsf{T} \boldsymbol{F} \boldsymbol{R} \boldsymbol{\beta}_* \rangle = \langle \boldsymbol{\beta}_*, \boldsymbol{F}^\mathsf{T} \boldsymbol{G}_0^2 \boldsymbol{F} \boldsymbol{\beta}_* \rangle + \Theta,$$

where

$$\begin{aligned}
|\Theta| &= \left| \langle \boldsymbol{\beta}_*, \boldsymbol{F}^\mathsf{T} (\boldsymbol{G}_{\boldsymbol{F}}^2 - \boldsymbol{G}_0^2) \boldsymbol{F} \boldsymbol{\beta}_* \rangle \right| \\
&= \left| \langle \boldsymbol{\beta}_*, \boldsymbol{F}^\mathsf{T} \boldsymbol{G}_0 \left( -2\boldsymbol{\Delta} \boldsymbol{G}_{\boldsymbol{F}} + \boldsymbol{\Delta} \boldsymbol{G}_{\boldsymbol{F}}^2 \boldsymbol{\Delta} \right) \boldsymbol{G}_0 \boldsymbol{F} \boldsymbol{\beta}_* \rangle \right| \leq C_{*,D,K} \frac{\log^3(d)}{p} \langle \boldsymbol{\beta}_*, \boldsymbol{F}^\mathsf{T} \boldsymbol{G}_0^2 \boldsymbol{F} \boldsymbol{\beta}_* \rangle.
\end{aligned} \qquad (159)$$

Let us decompose

$$\langle \boldsymbol{\beta}_*, \boldsymbol{F}^\mathsf{T} \boldsymbol{G}_0^2 \boldsymbol{F} \boldsymbol{\beta}_* \rangle = \langle \boldsymbol{\beta}_0, \boldsymbol{F}_0^\mathsf{T} \boldsymbol{G}_0^2 \boldsymbol{F}_0 \boldsymbol{\beta}_0 \rangle + 2\langle \boldsymbol{\beta}_+, \boldsymbol{F}_+^\mathsf{T} \boldsymbol{G}_0^2 \boldsymbol{F}_0 \boldsymbol{\beta}_0 \rangle + \langle \boldsymbol{\beta}_+, \boldsymbol{F}_+^\mathsf{T} \boldsymbol{G}_0^2 \boldsymbol{F}_+ \boldsymbol{\beta}_+ \rangle.$$

For the first term, we have directly from Proposition B.8 that

$$\left| \langle \boldsymbol{\beta}_0, \boldsymbol{F}_0^\mathsf{T} \boldsymbol{G}_0^2 \boldsymbol{F}_0 \boldsymbol{\beta}_0 \rangle - p \Psi_3(\nu_{2,0}; \boldsymbol{A}_0) \right| \leq C_{*,D,K} \cdot \mathcal{E}_3(p) \cdot p \Psi_3(\nu_{2,0}; \boldsymbol{A}_0). \qquad (160)$$

For the two other terms, notice that they correspond to the terms (101) and (102) in Proposition B.9 with $\boldsymbol{v} = \boldsymbol{\beta}_0$ and $\boldsymbol{u} = \boldsymbol{F}_+ \boldsymbol{\beta}_+$, where

$$\mathbb{E}[\boldsymbol{f}_{0,j} \langle \boldsymbol{f}_{+,j}, \boldsymbol{\beta}_+ \rangle] = 0, \qquad \mathbb{E}[u_i^2] = \|\boldsymbol{\Sigma}_+^{1/2} \boldsymbol{\beta}_*\|_2^2.$$

Hence, by Proposition B.9, we have with probability at least $1 - p^{-D}$ that

$$\left| \langle \boldsymbol{\beta}_+, \boldsymbol{F}_+^{\mathsf{T}} \boldsymbol{G}_0^2 \boldsymbol{F}_+ \boldsymbol{\beta}_+ \rangle - \frac{1}{\nu_{2,0}^2} \frac{\|\boldsymbol{\Sigma}_+^{1/2} \boldsymbol{\beta}_+\|_2^2}{p - \mathrm{Tr}(\boldsymbol{\Sigma}_0^2 (\boldsymbol{\Sigma}_0 + \nu_{2,0})^{-2})} \right| \le C_{*,D,K} \cdot \mathcal{E}_4(p) \cdot \frac{p \|\boldsymbol{\Sigma}_+^{1/2} \boldsymbol{\beta}_+\|_2^2}{\gamma_+^2},$$

$$\left| \langle \boldsymbol{\beta}_+, \boldsymbol{F}_+^{\mathsf{T}} \boldsymbol{G}_0^2 \boldsymbol{F}_0 \boldsymbol{\beta}_0 \rangle \right| \le C_{*,D,K} \cdot \mathcal{E}_4(p) \cdot \|\boldsymbol{\Sigma}_+^{1/2} \boldsymbol{\beta}_+\|_2 \frac{p^2 \nu_{2,0}}{\gamma_+^2} \sqrt{\Psi_3(\nu_{2,0}; \boldsymbol{A}_0)}.$$

(161)

Furthermore, by Lemma B.20 stated below

$$\left| \frac{\langle \boldsymbol{\beta}_0, \boldsymbol{\Sigma}_0 (\boldsymbol{\Sigma}_0 + \nu_{2,0})^{-2} \boldsymbol{\beta}_0 \rangle + \langle \boldsymbol{\beta}_+, \boldsymbol{\Sigma}_+ \boldsymbol{\beta}_+ \rangle / \nu_{2,0}^2}{p - \mathrm{Tr}(\boldsymbol{\Sigma}_0^2 (\boldsymbol{\Sigma}_0 + \nu_{2,0})^{-2})} - p \Psi_3(\nu_2; \boldsymbol{A}_*) \right| \le C \frac{\rho_{\gamma_+}(p)}{p} \cdot p \Psi_3(\nu_2; \boldsymbol{A}_*).$$

(162)

Combining Eqs. (154), (155), (156), and (157), we deduce that with probability at least $1 - p^{-D}$,

$$\left| p \widetilde{\Phi}_4(\boldsymbol{F}; \boldsymbol{A}_*, p\nu_1) - p \Psi_3(\nu_2; \boldsymbol{A}_*) \right| \le C_{*,D,K} \cdot \rho_{\gamma_+}(p)^2 \mathcal{E}_4(p) \cdot p \Psi_3(\nu_2; \boldsymbol{A}_*). \qquad (163)$$

where we used that

$$\frac{p \nu_{2,0}}{\gamma_+} \sqrt{\frac{p \|\boldsymbol{\Sigma}_+^{1/2} \boldsymbol{\beta}_*\|_2^2}{\gamma_+^2} p \Psi_3(\nu_{2,0}; \boldsymbol{A}_0)} \le C \rho_{\gamma_+}(p) \left\{ \frac{\|\boldsymbol{\Sigma}_+^{1/2} \boldsymbol{\beta}_*\|_2^2 / \nu_{2,0}^2}{p - \mathrm{Tr}(\boldsymbol{\Sigma}_0^2 (\boldsymbol{\Sigma}_0 + \nu_{2,0})^{-2})} + p \Psi_3(\nu_{2,0}; \boldsymbol{A}_0) \right\},$$

$$\frac{p \|\boldsymbol{\Sigma}_+^{1/2} \boldsymbol{\beta}_+\|_2^2}{\gamma_+^2} \le C \rho_{\gamma_+}(p)^2 \frac{\|\boldsymbol{\Sigma}_+^{1/2} \boldsymbol{\beta}_*\|_2^2 / \nu_{2,0}^2}{p - \mathrm{Tr}(\boldsymbol{\Sigma}_0^2 (\boldsymbol{\Sigma}_0 + \nu_{2,0})^{-2})}.$$

### Step 3: Combining the terms.

From Eqs. (158) and (163), we have with probability at least $1 - p^{-D}$ that

$$\left| (p\nu_1)^2 \langle \boldsymbol{\beta}_*, \boldsymbol{R}^2 \boldsymbol{\beta}_* \rangle - \nu_2 \Psi_1(\nu_2; \boldsymbol{A}_*) + (p\nu_1) p \Psi_3(\nu_2; \boldsymbol{A}_*) \right|$$
$$\le C_{*,D,K} \cdot \rho_{\gamma_+}(p)^2 \mathcal{E}_4(p) \cdot [\nu_2 \Psi_1(\nu_2; \boldsymbol{A}_*) + (p\nu_1) p \Psi_3(\nu_2; \boldsymbol{A}_*)].$$

First, it is straightforward to verify that indeed

$$\nu_2 \Psi_1(\nu_2; \boldsymbol{A}_*) - (p\nu_1) p \Psi_3(\nu_2; \boldsymbol{A}_*) = \widetilde{\mathsf{B}}_{n,p}(\boldsymbol{\beta}_*, \lambda).$$

Then by Eq. (121) in Lemma B.14, we conclude that with probability at least $1 - p^{-D}$,

$$\left| (p\nu_1)^2 \langle \boldsymbol{\beta}_*, \boldsymbol{R}^2 \boldsymbol{\beta}_* \rangle - \widetilde{\mathsf{B}}_{n,p}(\boldsymbol{\beta}_*, \lambda) \right| \le C_{*,D,K} \cdot \rho_{\gamma_+}(p)^2 \mathcal{E}_4(p) \cdot \widetilde{\mathsf{B}}_{n,p}(\boldsymbol{\beta}_*, \lambda),$$

which concludes the proof of this lemma. $\qquad \square$

**Lemma B.20.** *Assuming that $p^2 \xi_{\mathsf{m}}^2 \le \gamma_+$, we have*

$$\left| \nu_{2,0} \langle \boldsymbol{\beta}_0, (\boldsymbol{\Sigma}_0 + \nu_{2,0})^{-1} \boldsymbol{\beta}_0 \rangle + \|\boldsymbol{\beta}_+\|_2^2 - \nu_2 \Psi_1(\nu_2; \boldsymbol{A}_*) \right| \le \frac{C}{p} \nu_2 \Psi_1(\nu_2; \boldsymbol{A}_*),$$

$$\left| \frac{\langle \boldsymbol{\beta}_0, \boldsymbol{\Sigma}_0 (\boldsymbol{\Sigma}_0 + \nu_{2,0})^{-2} \boldsymbol{\beta}_0 \rangle + \langle \boldsymbol{\beta}_+, \boldsymbol{\Sigma}_+ \boldsymbol{\beta}_+ \rangle / \nu_{2,0}^2}{p - \mathrm{Tr}(\boldsymbol{\Sigma}_0^2 (\boldsymbol{\Sigma}_0 + \nu_{2,0})^{-2})} - p \Psi_3(\nu_2; \boldsymbol{A}_*) \right| \le C \frac{\rho_{\gamma_+}(p)}{p} \cdot p \Psi_3(\nu_2; \boldsymbol{A}_*).$$

*Proof of Lemma B.20.* This lemma follows from the same arguments as in the proofs of Lemma B.5 and Lemma B.19. $\qquad \square$

## C   Details of the numerical illustration

In this section we provide further examples of comparison between the excess risk computed using the deterministic equivalent in Theorem 3.3 and numerical simulations, together with details about their realization. Results from numerical experiments are obtained averaging over 20-50 seeds. The data dimension is $d = 100$, with the exception of experiments involving real data (see Appendix C.4) and the ones realized considering the Gaussian design described in Appendix C.2.

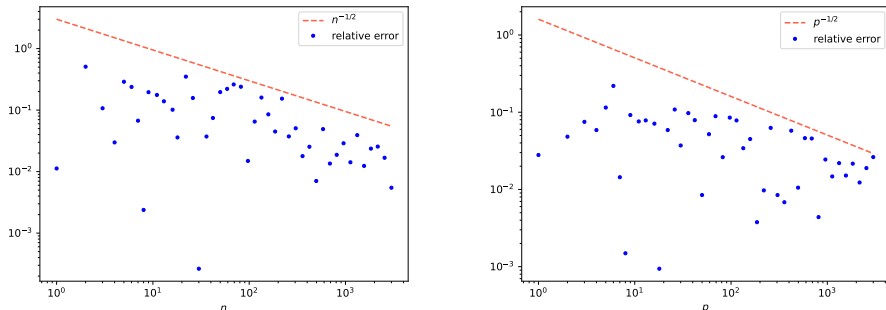

Figure 4: Relative difference between the excess risk (eq. (6)) of random features ridge regression from numerical simulation and its deterministic equivalent (Theorem 3.3), with regularization strength $\lambda = 0.1$, and noise variance $\sigma_\varepsilon^2 = 0.1$. The relative error is $O((n \wedge p)^{-1/2})$, in agreement with eq. (32). The simulations are made following the procedure described in appendix C.2, with $\xi_k = k^{-1.2}$ and $\beta_{*,k} = k^{-1.46}$; (**left**) $p = 3000$ fixed (**right**) $n = 3000$ fixed.

## C.1 Self-consistent equations

To solve Equations 18 and 19 numerically, the following approach has been employed. From equation 19,

$$\sqrt{\left(1 - \frac{n}{p}\right)^2 + 4\frac{\lambda}{p\nu_2}} = 2\frac{\nu_1}{\nu_2} - 1 + \frac{n}{p}. \tag{164}$$

Substituting this expression in equation 18, we obtain

$$2\left(1 - \frac{\nu_1}{\nu 2}\right) = \frac{2}{p}\mathrm{Tr}(\mathbf{\Sigma}(\mathbf{\Sigma} + \nu_2)^{-1}) \tag{165}$$

$$\overset{\nu_2 > 0}{\Longrightarrow} \nu_2 = \nu_1 + \frac{\nu_2}{p}\mathrm{Tr}(\mathbf{\Sigma}(\mathbf{\Sigma} + \nu_2)^{-1}). \tag{166}$$

Therefore, the parameters $\nu_1$ and $\nu_2$ have been computed by iterating

$$\nu_1^{t+1} = \frac{\nu_2^t}{2}\left[1 - \frac{n}{p} + \sqrt{\left(1 - \frac{n}{p}\right)^2 + 4\frac{\lambda}{p\nu_2^t}}\right], \tag{167}$$

$$\nu_2^{t+1} = \nu_1^{t+1} + \frac{\nu_2^t}{p}\mathrm{Tr}(\mathbf{\Sigma}(\mathbf{\Sigma} + \nu_2^t)^{-1}), \tag{168}$$

until a chosen tolerance $\epsilon$ was reached.

## C.2 Gaussian design

Figures 3, 4 and 9 have been realized considering Gaussian design for the vectors in 'feature space' defined in Appendix B.1. In particular, fixed $\mathbf{\Sigma}$ and $\boldsymbol{\beta}_*$, we have drawn

$$\{\boldsymbol{g}_i\}_{i \in [n]} \sim_{\text{i.i.d.}} \mathcal{N}(\mathbf{0}, \boldsymbol{I}), \quad \{\boldsymbol{f}_j\}_{j \in [p]} \sim_{\text{i.i.d.}} \mathcal{N}(\mathbf{0}, \mathbf{\Sigma}),$$

and consequently $\{y_i = \boldsymbol{\beta}_*^\top \boldsymbol{g}_i\}_{i \in [n]}$. Then the random feature estimator can be computed according to eqs. (61) and (62). In the figures produced with this setting, the elements of $\boldsymbol{\beta}_*$ and $\mathrm{diag}(\mathbf{\Sigma})$ follow power-laws truncated at the component $10^4$.

## C.3 Empirical diagonalization

Whenever the data probability distribution $\mu_x$ or the weights distribution $\mu_w$ are unknown (e.g. in all cases involving real data), we estimated the matrix $\mathbf{\Sigma}$ and the vector $\boldsymbol{\beta}_*$ following the procedure described in this section and summarized in Algorithm 1. Consider a data set of $N$ covariates

$\{\boldsymbol{x}_i\}_{i \in [N]}$ drawn from $\mu_x$ and $N$ noiseless labels $\{y_i = f_*(\boldsymbol{x}_i)\}_{i \in [N]}$, and a set of $P$ weights $\{\boldsymbol{w}_j\}_{j \in [P]}$. In Figures 1, 5, 6, 7, 8, for which this procedure was used, we take both $N, P = 10^4$ (with the exception of Fig. 7 (right), where $P = 8000$) and approximate $\mu_x$ and $\mu_w$ respectively with the empirical distributions $\tilde{\mu}_x = \sum_{i=1}^N N^{-1}\delta(\boldsymbol{x} - \boldsymbol{x}_i)$ and $\tilde{\mu}_w = \sum_{j=1}^P P^{-1}\delta(\boldsymbol{w} - \boldsymbol{w}_j)$. Then eq. (2) becomes

$$K(\boldsymbol{x}, \boldsymbol{x}') = \mathbb{E}_{\boldsymbol{w} \sim \mu_w}\left[\varphi(\boldsymbol{x}, \boldsymbol{w})\varphi(\boldsymbol{x}', \boldsymbol{w})\right] = \sum_{j=1}^P P^{-1}\varphi(\boldsymbol{x}, \boldsymbol{w}_j)\varphi(\boldsymbol{x}', \boldsymbol{w}_j), \tag{169}$$

and since eq. (11) implies $\mathbb{E}_{\boldsymbol{x}}K(\boldsymbol{x}, \boldsymbol{x}')\psi_k(\boldsymbol{x}) = \xi_k^2 \psi_k(\boldsymbol{x}')$, defining the Gram matrix $\boldsymbol{K}^{\text{emp}} \in \mathbb{R}^{N \times N}$ with elements $\tilde{K}_{ii'} = K(\boldsymbol{x}_i, \boldsymbol{x}_{i'})N^{-1}$ and the vectors $\tilde{\boldsymbol{\psi}}^k = (\psi_k(\boldsymbol{x}_1), \ldots, \psi_k(\boldsymbol{x}_N))^\top$, we can write the following eigenvalue problems, for $k \in [N]$:

$$\tilde{\boldsymbol{K}}\tilde{\boldsymbol{\psi}}_k = \tilde{\xi}_k^2 \tilde{\boldsymbol{\psi}}_k. \tag{170}$$

We then constructed the matrix $\tilde{\boldsymbol{\Sigma}} = \text{diag}(\tilde{\xi}_1^2, \ldots, \tilde{\xi}_N^2)$ and used it as an approximation of $\boldsymbol{\Sigma}$. One should note that in this situation $\mathbb{E}_{\boldsymbol{x}}\psi_k(\boldsymbol{x})\psi_{k'}(\boldsymbol{x}) = \delta_{kk'}$ corresponds to $\tilde{\boldsymbol{\psi}}_k\tilde{\boldsymbol{\psi}}_{k'} = N\delta_{kk'}$. Similarly, eq. (13) implies $\beta_{*,k} = \mathbb{E}_{\boldsymbol{x}}\left[f_*(\boldsymbol{x})\psi_k(\boldsymbol{x})\right]$, which can be approximated by

$$\tilde{\beta}_k = N^{-1}\boldsymbol{y}^\top\tilde{\boldsymbol{\psi}}_k. \tag{171}$$

---

**Algorithm 1** Empirical diagonalization

---

**Require:** $\{\boldsymbol{x}_i\}_{i \in [N]} \sim_{\text{i.i.d.}} \mu_x, \{y_i = f_*(\boldsymbol{x}_i)\}_{i \in [N]}, \{\boldsymbol{w}_j\}_{j \in [P]} \sim_{\text{i.i.d.}} \mu_w$
**Ensure:** $\boldsymbol{\Sigma}, \boldsymbol{\beta}_*, \mathsf{R}_{n,p}(\boldsymbol{\beta}_*, \lambda)$
   **for** $i, i' \in \{1, \ldots, N\}$ **do**
      $\tilde{K}_{ii'} \leftarrow (NP)^{-1}\sum_{j=1}^P \varphi(\boldsymbol{x}_i, \boldsymbol{w}_j)\varphi(\boldsymbol{x}_{i'}, \boldsymbol{w}_j)$
   **end for**
   $\{(\tilde{\xi}_k, \tilde{\boldsymbol{\psi}}_k)_{k \in [N]}\} \leftarrow \text{eig}(\tilde{\boldsymbol{K}})$                           $\triangleright \ \tilde{\boldsymbol{\psi}}_k\tilde{\boldsymbol{\psi}}_{k'} \overset{!}{=} N\delta_{kk'}$
   **for** $k \in \{1, \ldots, N\}$ **do**
      $\tilde{\beta}_k \leftarrow N^{-1}\boldsymbol{y}^\top\tilde{\boldsymbol{\psi}}_k$
   **end for**
   $\boldsymbol{\Sigma} \leftarrow \text{diag}(\tilde{\xi}_1, \ldots, \tilde{\xi}_N)$
   $\boldsymbol{\beta}_* \leftarrow (\beta_1, \ldots, \beta_N)^\top$
   Iterate eqs. (167-168) up to tolerance $\epsilon$
   Compute the deterministic equivalent for the excess risk (22-24)

---

### C.4 Real data

We performed numerical simulations sampling the training data from the MNIST data set Lecun et al. [1998] and the FashionMNIST data set Xiao et al. [2017], standardizing both covariates and labels, reshaping the images into vectors with $d = 748$. Results are shown in Figures 1 (right) and 7 (left).

### C.5 Trained network

In Figure 7 (right), we apply the procedure described in Appendix C.3 to the trained weights of a two layer neural network with hidden layer of size $p$

$$\hat{f}(\boldsymbol{x}; \boldsymbol{W}, \boldsymbol{a}) = \frac{1}{\sqrt{p}}\sum_{j=1}^p a_j \varphi(\langle \boldsymbol{x}, \boldsymbol{w}_j\rangle).$$

At initialization, the weights are randomly drawn; then, after sampling a training dataset $\boldsymbol{X}_{\text{tr}} \in \mathbb{R}^{n_{\text{tr}} \times d}$, $\boldsymbol{y}_{\text{tr}} \in \mathbb{R}^{n_{\text{tr}}}$, the weights of the first layer are trained using gradient descent, iterating for $t = 1, \ldots, T$ the following

$$\boldsymbol{W}_{t+1} = \boldsymbol{W}_t + \frac{\eta}{n_{\text{tr}}}\boldsymbol{X}_{\text{tr}}^\top\left[\left((\boldsymbol{y}_{\text{tr}} - \hat{f}(\langle \boldsymbol{X}_{\text{tr}}; \boldsymbol{W}_t, \boldsymbol{a}))^\top\boldsymbol{a}\right) \odot \varphi'(\boldsymbol{X}_{\text{tr}}\boldsymbol{W}_t^\top)\right], \tag{172}$$

where $\eta$ is the learning rate, $\odot$ is the Hadamard product, $\hat{f}$ and $\varphi$ are applied component-wise; finally, the weights of the second layer are minimized using ridge regression, as in eq. (5).

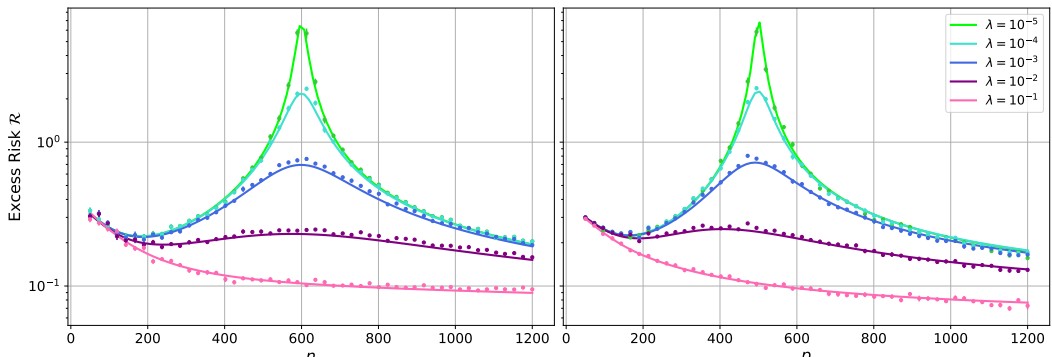

Figure 5: Excess risk eq. (6) of random features ridge regression. Solid lines are obtained from the deterministic equivalent in Theorem 3.3, and points are numerical simulations, with the different curves denoting different regularization strengths $\lambda \geq 0$. Training data $(\boldsymbol{x}_i, y_i)_{i \in [n]}$, sampled from a teacher-student model $y_i = \tanh(\langle \boldsymbol{\beta}, \boldsymbol{x}_i \rangle) + \varepsilon_i$, $\sigma_\varepsilon^2 = 0.1$, with random feature map $\varphi(\boldsymbol{x}, \boldsymbol{w}) = \mathrm{ReLU}(\langle \boldsymbol{w}, \boldsymbol{x} \rangle)$. Both covariates $\{\boldsymbol{x}_i\}$ and weights $\{\boldsymbol{w}_i\}$ are uniformly sampled from the $d$-dimensional spheres respectively with radius $\sqrt{d}$ and 1. (**Left**) Excess risk as a function of $n$, with $p = 600$ fixed. (**Right**) Excess risk as a function of $p$, with $n = 500$ fixed.

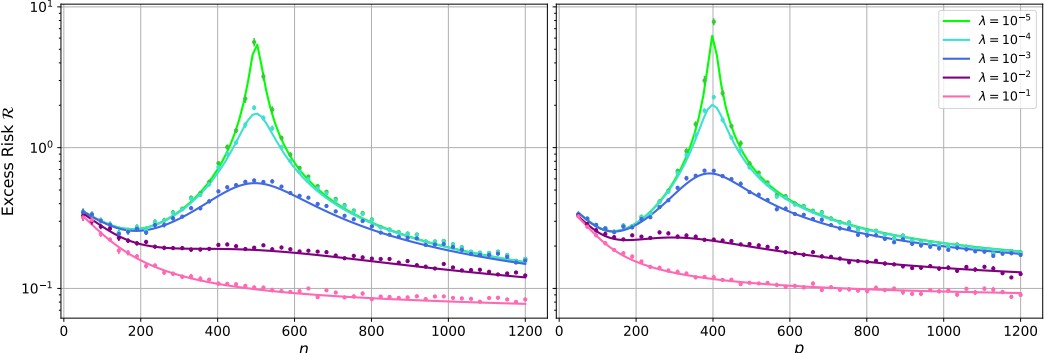

Figure 6: Excess risk eq. (6) of random features ridge regression. Solid lines are obtained from the deterministic equivalent in Theorem 3.3, and points are numerical simulations, with the different curves denoting different regularization strengths $\lambda \geq 0$. Training data $(\boldsymbol{x}_i, y_i)_{i \in [n]}$, sampled from a teacher-student model $y_i = \tanh(\langle \boldsymbol{\beta}, \boldsymbol{x}_i \rangle) + \varepsilon_i$, $\sigma_\varepsilon^2 = 0.1$, $\boldsymbol{x}_i \sim_{\text{i.i.d.}} \mathcal{N}(0, \boldsymbol{I}_d)$, with a spiked random feature map $\varphi(\boldsymbol{x}, \boldsymbol{w}) = \mathrm{erf}(\langle \boldsymbol{w} + u\boldsymbol{v}, \boldsymbol{x} \rangle)$ where $\boldsymbol{w} \sim \mathcal{N}(0, d^{-1}\boldsymbol{I}_d)$, $\boldsymbol{v} \in \mathbb{R}^d \sim \mathcal{N}(0, d^{-1}\boldsymbol{I}_d)$, and $u \sim \mathcal{N}(0, 1)$. (**Left**) Excess risk as a function of $n$, with $p = 500$ fixed. (**Right**) Excess risk as a function of $p$, with $n = 300$ fixed.

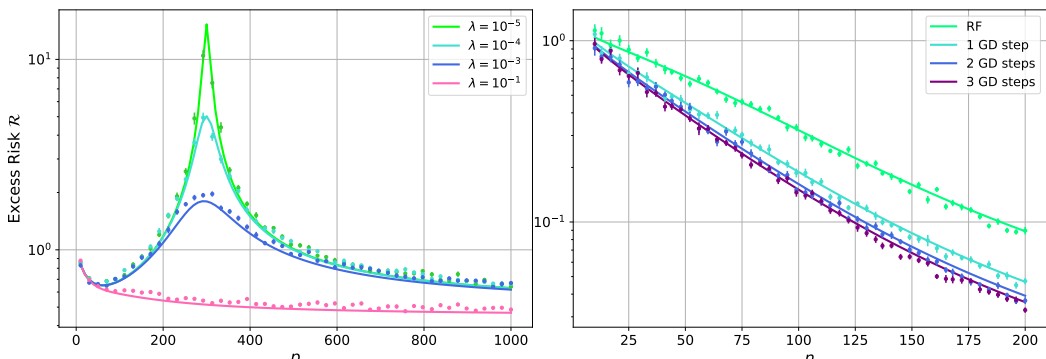

Figure 7: (**Left**) Excess risk eq. (6) of random features ridge regression. Solid lines are obtained from the deterministic equivalent in Theorem 3.3, and points are numerical simulations, with the different curves denoting different regularization strengths $\lambda \geq 0$. Training data $(\boldsymbol{x}_i, y_i)_{i \in [n]}$, $n = 300$, sub-sampled from the MNIST data set Lecun et al. [1998], with feature map given by $\varphi(\boldsymbol{x}, \boldsymbol{w}) = \mathrm{erf}(\langle \boldsymbol{w}, \boldsymbol{x} \rangle)$ and $\mu_w = \mathcal{N}(0, d^{-1} \boldsymbol{I}_d)$. (**Right**) Excess risk eq. (6) of random features ridge regression. Solid lines are obtained from the deterministic equivalent in Theorem 3.3, and points are numerical simulations, with the different curves denoting different number of total iterations of gradient descent on the weight of the first layer with learning rate $\eta = 10^{-2}$, before training the second layer with regularization strength $\lambda = 10^{-4}$ (details in Appendix C.5). Zero iterations corresponds to random feature regression (RF). Training data $(\boldsymbol{x}_i, y_i)_{i \in [n]}$, sampled from a teacher-student model $y_i = \langle \boldsymbol{\beta}, \boldsymbol{x}_i \rangle$, with random feature map $\varphi(\boldsymbol{x}, \boldsymbol{w}) = \mathrm{ReLU}(\langle \boldsymbol{w}, \boldsymbol{x} \rangle)$ and $p = 8000$ fixed. Both covariates $\{\boldsymbol{x}_i\}$ and initialization weights $\{\boldsymbol{w}_i\}$ are uniformly sampled from the $d$-dimensional spheres respectively with radius $\sqrt{d}$ and 1.

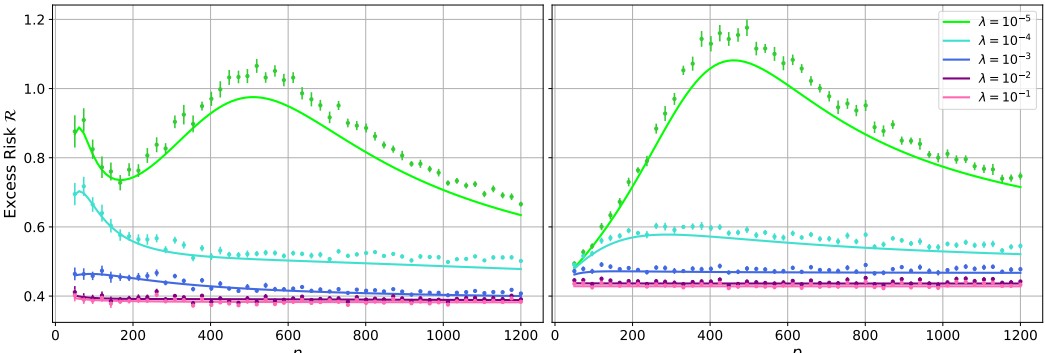

Figure 8: Excess risk eq. (6) of random features ridge regression. Solid lines are obtained from the deterministic equivalent in Theorem 3.3, and points are numerical simulations, with the different curves denoting different regularization strengths $\lambda \geq 0$. Training data $(\boldsymbol{x}_i, y_i)_{i \in [n]}$, sampled from a teacher-student model $y_i = \tanh(\langle \boldsymbol{\beta}, \boldsymbol{x}_i \rangle) + \varepsilon_i$, $\sigma_\varepsilon^2 = 0.1$, with random feature map given by the convolutional features with global average pooling $\varphi(\boldsymbol{x}, \boldsymbol{w}) = {}^1/d \sum_{\ell=1}^{d} \mathrm{ReLU}(\langle \boldsymbol{w}, g_\ell \cdot \boldsymbol{x} \rangle)$ where $g_\ell \cdot \boldsymbol{x} = (x_{\ell+1}, \dots, x_d, x_1, \dots, x_\ell)$ is the $\ell$-shift operator with cyclic boundary conditions. Both covariates $\{\boldsymbol{x}_i\}$ and weights $\{\boldsymbol{w}_i\}$ are uniformly sampled from the $d$-dimensional spheres respectively with radius $\sqrt{d}$ and 1. (**Left**) Excess risk as a function of $n$, with $p = 500$ fixed. (**Right**) Excess risk as a function of $p$, with $n = 500$ fixed. The discrepancy between theoretical results and numerical experiments is $\approx {}^\mathcal{R}/\sqrt{n \wedge p}$, compatible with the approximation rate in eq. (32).

# D   Derivation of the rates

In this appendix we present a sketch of the derivation of the decay rates for the excess error given in Section 4. We choose

$$p = n^q, \quad \lambda = n^{-(\ell-1)}, \quad \text{with } q, l \geq 0$$

and we assume

$$\eta_k = k^{-\alpha}, \quad \beta_{*k} = k^{-\frac{1+2\alpha r}{2}}, \quad \text{with } \alpha > 1, \ r > 0,$$

which follows the source and capacity conditions stated in (37). In order to simplify the derivation of the rates, we introduce the following notation:

$$T_{\delta\gamma}^s(\nu) := \sum_{k=1}^\infty \frac{k^{-s-\delta\alpha}}{(k^{-\alpha} + \nu)^\gamma}, \qquad\qquad s \in 0,1, \ 0 \leq \delta \leq \gamma.$$

For $s + \alpha(\delta - \gamma) < 1$ and in the limit $\nu \to 0$, this term can be written as a Riemann sum as follows

$$T_{\delta\gamma}^s(\nu) = \nu^{-\gamma+\delta+s/\alpha} \sum_{k=1}^\infty \frac{(k\nu^{1/\alpha})^{-s-\delta\alpha}}{((k\nu^{1/\alpha})^{-\alpha} + 1)^\gamma} \tag{173}$$

$$\overset{\nu\to 0}{\approx} \nu^{-(\gamma-\delta)-(1-s)/\alpha} \int_{\nu^{1/\alpha}}^\infty \frac{x^{-s-\delta\alpha}}{(x^{-\alpha} + 1)^\gamma} = O\left(\nu^{-(\gamma-\delta)-(1-s)/\alpha}\right). \tag{174}$$

Otherwise, if $s + \alpha(\delta - \gamma) > 1$, we can write:

$$T_{\delta\gamma}^s(\nu) = \sum_{k=1}^{\lfloor \nu^{1/\alpha} \rfloor} \frac{k^{-s-\delta\alpha}}{(k^{-\alpha} + \nu)^\gamma} + \sum_{k=\lfloor \nu^{1/\alpha} \rfloor+1}^\infty \frac{k^{-s-\delta\alpha}}{(k^{-\alpha} + \nu)^\gamma}$$

$$\overset{\nu\to 0}{\approx} O(1) + \nu^{-(\gamma-\delta)-(1-s)/\alpha} \int_{1+\nu^{1/\alpha}}^\infty \frac{x^{-s-\delta\alpha}}{(x^{-\alpha} + 1)^\gamma} = O(1).$$

Hence, for $\nu \to 0$,

$$T_{\delta\gamma}^s(\nu) = O\left(\nu^{1/\alpha[s-1+\alpha(\delta-\gamma)]\wedge 0}\right). \tag{175}$$

Rewriting (18-19) as follows, we study the dependence of the positive parameters $\nu_1$ and $\nu_2$ with $n$:

$$\begin{cases} \nu_2 = \frac{\nu_2}{p} T_{11}^0(\nu_2) + \nu_1 \\ \nu_1 = \frac{\nu_1}{n} T_{11}^0(\nu_2) + \frac{\lambda}{n} \end{cases}$$

. In the limit $n \to \infty$, we can distinguish the following regimes

$$\begin{cases} T_{11}^0(\nu_2) \ll n, p \\ \nu_1 = n^{-1}\lambda\left(1 + O\left(T_{11}^0(\nu_2)n^{-1}\right)\right) \\ \nu_2 = n^{-1}\lambda\left(1 + O\left(T_{11}^0(\nu_2)(n \wedge p)^{-1}\right)\right) \end{cases} \overset{\nu_2=o(1)}{\Longrightarrow} \begin{cases} \nu_2^{-1/\alpha} \ll n^{1\wedge q} \implies \ell < \alpha(1 \wedge q) \\ \nu_1 \approx \nu_2 \approx n^{-\ell} \end{cases}$$

$$\begin{cases} T_{11}^0(\nu_2) \ll p \\ \nu_1 \gg \lambda n^{-1} \\ T_{11}^0(\nu_2) = n - (n\nu_1)^{-1}\lambda \\ \nu_2 = \nu_1(1 + O\left(T_{11}^0(\nu_2)p^{-1}\right)) \end{cases} \overset{(a)}{\Longrightarrow} \begin{cases} \nu_2^{-1/\alpha} \ll n^q \\ \nu_1 \gg n^{-\ell} \\ \nu_2^{-1/\alpha} = O(n + o(n)) \\ \nu_2 = \nu_1(1 + o(1)) \end{cases} \implies \begin{cases} q > 1, \ \ell > \alpha \\ \nu_1 \approx \nu_2 \propto n^{-\alpha} \end{cases}$$

$$\begin{cases} T_{11}^0(\nu_2) \ll n \\ \nu_1 \ll \nu_2 \\ \nu_1 = n^{-1}\lambda\left(1 + O\left(T_{11}^0(\nu_2)n^{-1}\right)\right) \\ T_{11}^0(\nu_2) = p - \nu_1\nu_2^{-1} \end{cases} \overset{(a)}{\Longrightarrow} \begin{cases} \nu_2^{-1/\alpha} \ll n \\ \nu_1 \ll \nu_2 \\ \nu_1 = n^{-\ell}(1 + o(1)) \\ \nu_2^{-1/\alpha} = n^q - o(1) \end{cases} \implies \begin{cases} q < \left(1 \wedge \ell\alpha^{-1}\right) \\ \nu_1 \approx n^{-\ell} \\ \nu_2 \propto n^{-\alpha q} \end{cases}$$

In $(a)$ we have used the fact that, for $\nu_2$ constant or diverging with $n$, $T_{11}^0(\nu_2)$ is respctively $O(1)$ or infinitesimal, while we have that $T_{11}^0(\nu_2)$ is diverging in both cases. Hence, $\nu_2$ must be infinitesimal,

allowing us to use (173).

In conclusion, as $n \to \infty$,

$$\nu_1 \approx \begin{cases} O\left(n^{-\alpha}\right), & \text{for } q > 1 \text{ and } \ell > \alpha, \\ n^{-\ell}, & \text{otherwise} \end{cases} \tag{176}$$

$$\nu_2 \approx O\left(n^{-\alpha(1 \wedge q \wedge \ell/\alpha)}\right), \tag{177}$$

in particular

$$\frac{\nu_1}{\nu_2} = \begin{cases} 1 - O\left(\nu_2^{-1/\alpha} n^{-q}\right), & \text{for } q > 1 \wedge \ell/\alpha \\ O\left(n^{-\ell + \alpha q}\right) = o(1) & \text{otherwise} \end{cases}. \tag{178}$$

## D.1 Variance term

Considering the results (176-178), we can write eq. (20) as

$$\Upsilon(\nu_1, \nu_2) = \frac{p}{n}\left[\left(1 - \frac{\nu_1}{\nu_2}\right)^2 + \left(\frac{\nu_1}{\nu_2}\right)^2 \frac{T_{22}^0(\nu_2)}{p - T_{22}^0(\nu_2)}\right] \tag{179}$$

$$= \begin{cases} n^{-1}O(\nu_2^{-1/\alpha}) = O\left(n^{-(1-(1\wedge \ell/\alpha))}\right), & \text{for } q > 1 \wedge \ell/\alpha \\ n^{-(1-q)}(1 + o(1)) & \text{otherwise} \end{cases} \tag{180}$$

One could notice, using the integral approximation of the Riemann sum $T_{22}^0(\nu_2)$ given in (173), that $1 - \Upsilon(\nu_1, \nu_2) = O(1)$ for any choice of $\ell$ and $q$. Hence, the variance term given by (23) decays with $n$ with rate

$$\gamma_{\mathcal{V}}(\ell, q) = 1 - \left(\frac{\ell}{\alpha} \wedge q \wedge 1\right).$$

## D.2 Bias term

Using again (176-178), we can compute the rate of $\chi(\nu_2)$ defined in eq. (21), as $n \to \infty$:

$$\chi(\nu_2) = \frac{T_{12}^0(\nu_2)}{p - T_{22}^0(\nu_2))} = n^{-q}O\left(\nu_2^{-1-1/\alpha}\right)\left(1 + n^{-q}O\left(\nu_2^{-1/\alpha}\right)\right)$$

$$= n^{-q}O\left(\nu_2^{-1-1/\alpha}\right)$$

Using the integral approximation given in (173), one could verify that $p - T_{22}^0(\nu_2) = O(p)$ for any choice of $\ell$ and $q$.

The deterministic equivalent for the bias term, given in eq. (22), can be written as

$$\mathsf{B}_{n,p}(\boldsymbol{\beta}_*, \lambda) = \frac{\nu_2^2}{1 - \Upsilon(\nu_1, \nu_2)}\left(T_{2r,2}^1(\nu_2) + \chi(\nu_2)T_{2r+1,2}^1(\nu_2)\right) \tag{181}$$

$$= O\left(\nu_2^2\right)O\left(\nu_2^{2(r-1 \wedge 0)} + n^{-q}\nu_2^{-1-1/\alpha+(2r-1)\wedge 0}\right) \tag{182}$$

$$= O\left(\nu_2^{2(r \wedge 1)} + n^{-q}\nu_2^{-1/\alpha+2(r\wedge1/2)}\right) \tag{183}$$

where we have used (175) to compute the scalings of the terms $T^1\delta\gamma$ and the fact that $1 - \Upsilon(\nu_1, \nu_2) = O(1)$.

From eq. (183), and using the result (177), it is straightforward to see that the decay rate of the bias term is given by

$$\gamma_{\mathcal{B}} = \left[2\alpha\left(\frac{\ell}{\alpha} \wedge q \wedge 1\right)(r \wedge 1)\right] \wedge \left[\left(2\alpha\left(r \wedge \frac{1}{2}\right) - 1\right)\left(\frac{\ell}{\alpha} \wedge q \wedge 1\right) + q\right]. \tag{184}$$

Examples of the results of Theorem 4.1 and Corollary 4.2 are shown in Fig. 3 and 9.

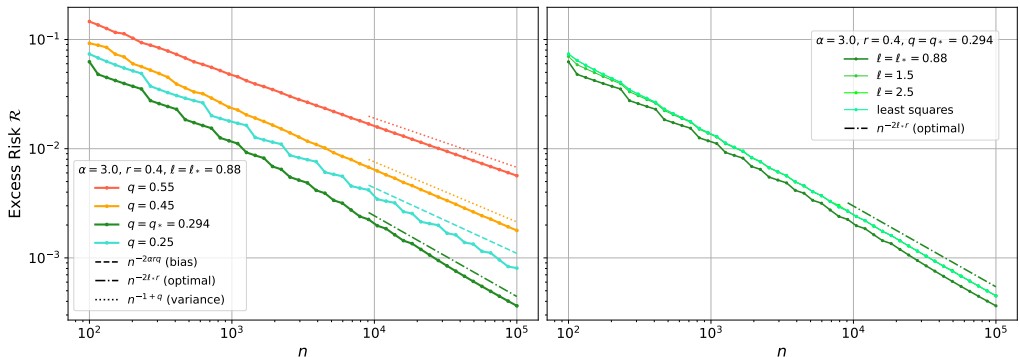

Figure 9: Excess risk eq. (6) of random features ridge regression as a function of the number of samples $n$ under source and capacity conditions eq. (37) and power-law assumptions $\lambda = n^{-(\ell-1)}$, $p = n^q$, with noise variance $\sigma_\varepsilon^2 = 1$, obtained from the deterministic equivalent Theorem 3.3. Dashed and dotted lines are the analytical rates from Theorem 4.1, stated in the legend. The colour scheme is the following: variance dominated region: orange and brown for the slow decay regime; cyan for the bias dominated region; shades of green for the optimal decay (red lines in Fig. 2 (right). In particular we show: (**left**) the crossover between the orange and teal regions in Fig. 2 at fixed regularization and $r < 1/2$; (**right**) the optimal decay rate along the horizontal red line line in Fig. 2 at $q = q_*$ and $r < 1/2$, for any $\lambda \leq \lambda_*$, included the non regularized case.

### D.3 Details of Remark 4.1

In order to extend the results of Theorem 4.1 and Corollary 4.2 to the excess risk defined in (6), in this section we compute the intervals for $\ell$ and $q$ such that the assumptions of Theorem 3.3 hold and the approximation rates $\mathcal{E}(n,p)$ are vanishing, under source and capacity conditions.

Given $n$, $p = n^q$, $\lambda^{-(\ell-1)}$ and $\nu_2$ as in (177) assumption 3.2 is verified for $\mathrm{m} = n^{q+\ell}$ and $\mathsf{C}_* = O(\nu_2^{-1})$. In fact

$$\sum_{k=\mathrm{m}+1}^{\infty} \xi_k^2 > \int_{\mathrm{m}+1}^{\infty} x^{-\alpha} = \frac{(\mathrm{m}+1)^{1-\alpha}}{\alpha-1} \tag{185}$$

and the inequality in (16) holds if

$$n^{2q}(\mathrm{m}+1)^{-\alpha} \leq n^{q-\ell} \frac{(\mathrm{m}+1)^{1-\alpha}}{\alpha-1} \implies \mathrm{m} \geq n^{q+\ell}(\alpha-1) - 1. \tag{186}$$

The inequalities in (17) can be written as

$$\mathsf{C}_* \geq \frac{T_{11}^0(\nu_2)}{T_{22}^0(\nu_2)} = O(1), \tag{187}$$

$$\mathsf{C}_* \geq \frac{T_{2r,1}^1(\nu_2)}{\nu_2 T_{2r,2}^1(\nu_2)} = O\left(\nu_2^{-((0\vee(2r-1))\wedge 1)}\right). \tag{188}$$

Then, introducing $\eta_* \in (0, 1/2)$, we can compute the following quantities of interest (introduced in eqs. (25) to (27)):

$$r_{\boldsymbol{\Sigma}}(\lfloor \eta_* \cdot k \rfloor) = \frac{\text{Tr}(\boldsymbol{\Sigma}_{\geq \lfloor \eta_* \cdot k \rfloor})}{||\boldsymbol{\Sigma}_{\geq \lfloor \eta_* \cdot k \rfloor}||_{\text{op}}} = O(\lfloor \eta_* \cdot k \rfloor) = O(k) \tag{189}$$

$$M_{\boldsymbol{\Sigma}}(k) := 1 + \frac{r_{\boldsymbol{\Sigma}}(\lfloor \eta_* \cdot k \rfloor) \vee k}{k} \log\left(r_{\boldsymbol{\Sigma}}(\lfloor \eta_* \cdot k \rfloor) \vee k\right) = 1 + O(\log k), \tag{190}$$

$$\rho_\kappa(p) := 1 + \frac{p \cdot \xi^2_{\lfloor \eta_* \cdot p \rfloor}}{\kappa} M_{\boldsymbol{\Sigma}}(p) = 1 + O\left(\frac{n^{q(1-\alpha)}}{\kappa} \log n\right), \tag{191}$$

$$\widetilde{\rho}_\kappa(n, p) := 1 + \mathbb{1}[n \leq p/\eta_*] \cdot \left\{\frac{n\xi^2_{\lfloor \eta_* \cdot n \rfloor}}{\kappa} + \frac{n}{p} \cdot \rho_\kappa(p)\right\} M_{\boldsymbol{\Sigma}}(n) \tag{192}$$

$$\overset{n \geq \eta_*^{1/(1-q)}}{=} 1 + \mathbb{1}[q \geq 1]O\left(\frac{n^{1-\alpha}}{\kappa} \log n\right). \tag{193}$$

Similarly, considering $\nu_1$ scaling as in eq. (176), we have that eq. (31)

$$\gamma_\lambda = \frac{p\lambda}{n} + \sum_{k=\mathsf{m}+1}^{\infty} \xi_k^2 = O\left(n^{q-\ell} + n^{(q+\ell)(1-\alpha)}\right), \tag{194}$$

$$\gamma_+ = p\nu_1 + \sum_{k=\mathsf{m}+1}^{\infty} \xi_k^2 = O\left(\mathbb{1}[q \geq 1]n^{q-(\ell \wedge \alpha)} + \mathbb{1}[q < 1]n^{q-\ell} + n^{(q+\ell)(1-\alpha)}\right) \tag{195}$$

$$= O\left(\mathbb{1}[q \geq 1]n^{q-(\ell \wedge \alpha)} + \mathbb{1}\left[\frac{\ell}{\alpha}(2-\alpha) \leq q < 1\right]n^{q-\ell} + \mathbb{1}\left[q < \frac{\ell}{\alpha}(2-\alpha)\right]n^{(q+\ell)(1-\alpha)}\right). \tag{196}$$

The last step is a consequence of

$$q - (\ell \wedge \alpha) > (q+\ell)(1-\alpha) \implies q > \frac{\ell}{\alpha}\underbrace{(1-\alpha)}_{<0} + \left(\frac{\ell}{\alpha} \wedge 1\right), \tag{197}$$

$$q - \ell > (q+\ell)(1-\alpha) \implies q > \frac{\ell}{\alpha}(2-\alpha) \tag{198}$$

Fixing $K > 0$ and considering the , we consider condition (28):

$$\lambda \geq n^{-K} \impliedby \ell \leq 1 + K, \tag{199}$$

$$\gamma_\lambda \geq p^{-K} \impliedby \begin{cases} \ell \leq q(1+K) \\ q > \frac{(2-a)}{a}\ell \end{cases} \vee \begin{cases} \ell \leq \frac{q(1+K-\alpha)}{\alpha-1} \\ q < \frac{(2-a)}{a}\ell \end{cases}, \tag{200}$$

$$\widetilde{\rho}_\lambda(n, p)^{5/2} \cdot \log^{3/2}(n) \leq K\sqrt{n} \impliedby \ell > \mathbb{1}[q \geq 1]\left(\alpha + \frac{1}{5}\right), \tag{201}$$

and, similarly, condition (29) is satisfied if, for $q \geq 1$

$$\left(1 + O\left(n^{\ell-\alpha}\log n\right)\right)^2 \left(O\left(1 + n^{(\ell \wedge \alpha)-q\alpha}\log n\right)\right)^8 q\log^4 n \leq Kn^{q/2} \tag{202}$$

$$\implies 2(\ell-\alpha) \vee 0 < \frac{q}{2} \implies \ell < \frac{q}{4} + \alpha, \tag{203}$$

while, for $q < 1$, if

$$\begin{cases} 1 > q \geq \frac{\ell}{\alpha}(2-\alpha) \\ \left(1 + O\left(n^{\ell-q\alpha}\right)\right)^8 q\log^4 n \leq Kn^{q/2} \end{cases} \vee \begin{cases} q < \frac{\ell}{\alpha}(2-\alpha) \\ \left(1 + O\left(n^{\ell(\alpha-1)}\right)\right)^8 q\log^4 n \leq Kn^{q/2} \end{cases} \tag{204}$$

$$\implies \begin{cases} 1 > q \geq \frac{\ell}{\alpha}(2-\alpha) \\ \ell < \alpha q + \frac{q}{16} \end{cases} \vee \begin{cases} q < \frac{\ell}{\alpha}(2-\alpha) \\ \ell < \frac{q}{16(\alpha-1)} \end{cases} \tag{205}$$

$$\implies \ell < q\left(\left(\alpha + \frac{1}{16}\right) \vee \frac{1}{16(\alpha-1)}\right),^4 \tag{206}$$

|  | This work | Bahri et al. [2024] | Maloney et al. [2022] | Atanasov et al. [2024] |
|---|---|---|---|---|
| Input dimension | $d$ | $d$ | $M$ | $D$ |
| Number of features | $p$ | $P$ | $N$ | $N$ |
| Number of samples | $n$ | $D$ | $T$ | $P$ |
| Capacity | $\alpha$ | $1 + \tilde{\alpha}$ | $1 + \tilde{\alpha}$ | $\alpha$ |
| Source | $r$ | $\nicefrac{1}{2}(1 - \nicefrac{1}{(\tilde{\alpha}+1)})$ | $\nicefrac{1}{2}(1 - \nicefrac{1}{(\tilde{\alpha}+1)})$ | $r$ |
| Target decay (in $L_2$) | $\alpha r + \nicefrac{1}{2}$ | $\nicefrac{1}{2}(1 + \tilde{\alpha})$ | $\nicefrac{1}{2}(1 + \tilde{\alpha})$ | $\alpha r + \nicefrac{1}{2}$ |

Table 1: Dictionary of notation between the source and capacity conditions defined in eq. (38) and the scalings in different neural scaling laws works. Note that since Bahri et al. [2024], Maloney et al. [2022] also employ the greek letter "$\alpha$", we denote theirs by $\tilde{\alpha}$ to avoid confusion.

where the last step is a consequence of

$$\frac{\alpha}{2-\alpha} \leq \alpha + \frac{1}{16} \leq \frac{1}{16(\alpha-1)}, \qquad \text{for } \alpha \geq \overline{\alpha} := \frac{15 + \sqrt{353}}{32} \approx 1.05588, \qquad (207)$$

$$\frac{\alpha}{2-\alpha} > \alpha + \frac{1}{16} > \frac{1}{16(\alpha-1)}, \qquad \text{for } \alpha < \overline{\alpha}. \qquad (208)$$

Finally, the approximation rate defined in remark 4.1 is

$$\mathcal{E}(n,p) = \left( n^{-\nicefrac{1}{2}} + \mathbb{1}[q \geq 1]\tilde{O}\left( n^{6(\ell-\alpha)-\nicefrac{1}{2}} \right) \right) + \qquad (209)$$

$$\left( n^{-\nicefrac{q}{2}} + \mathbb{1}[q \geq 1]\tilde{O}\left( n^{2(\ell-\alpha)-\nicefrac{q}{2}} \right) \right) \left( 1 + \tilde{O}\left( \frac{n^{8q(1-\alpha)}}{\gamma_+^8} \right) \right), \qquad (210)$$

where the second term vanishes under the conditions in (203) and (206), and the first term vanishes by further assuming, for $q \geq 1$

$$\ell < \alpha + \frac{1}{12}. \qquad (211)$$

# E   Comparison with neural scaling laws

In this appendix we discuss the relationship between our results and the recent literature of the theory of neural scaling laws with linear models. We adopt a notation close to ours, with dictionary to their notation given in Table 1 and Table 2.

Bahri et al. [2024] and Maloney et al. [2022] have considered a model where with Gaussian input data and linear target function:

$$f_\star(\boldsymbol{x}_i) = \langle \boldsymbol{\beta}_\star, \boldsymbol{x}_i \rangle, \qquad \boldsymbol{x}_i \sim \mathcal{N}(0, \boldsymbol{\Lambda}), \qquad i \in [n] \qquad (212)$$

The covariance matrix $\boldsymbol{\Lambda} = \mathrm{diag}(\lambda_k)_{k \in [d]}$ is taken to be diagonal, with eigenvalues following a power-law scaling:

$$\lambda_k \sim \left( \frac{d}{k} \right)^\alpha, \qquad k \in [d] \qquad (213)$$

with $\alpha > 1$ and the target weights are assumed to be random Gaussian vectors $\boldsymbol{\beta}_\star \sim \mathcal{N}(0, \nicefrac{1}{d}I_d)$. In particular, note that $\mathrm{Tr}\boldsymbol{\Lambda} \sim d$ for $d \to \infty$. Given the training data, they consider least-squares regression in the class of linear random features predictor:

$$\hat{f}(\boldsymbol{x}; \boldsymbol{a}) = \langle \boldsymbol{a}, \boldsymbol{W}\boldsymbol{x} \rangle \qquad (214)$$

where $\boldsymbol{W} \in \mathbb{R}^{p \times d}$ is a Gaussian random matrix elements in $\mathcal{N}(0, \nicefrac{1}{d}I_d)$.[5]

---

[4]Under this last condition, if $K \geq \alpha - 1 + \nicefrac{1}{16}$, both inequalities (199) and (200) are satisfied.

[5]In [Maloney et al., 2022], $\boldsymbol{\beta}_\star$ has variance $\sigma_w/d$, the spectrum has a scale $\lambda_-$ and the random projection $\boldsymbol{W}$ has variance $\sigma_u/d$. Here we take $\sigma_w = \lambda_- = \sigma_u = 1$ since it is irrelevant to the discussion.

|  | This work | Bordelon et al. [2024] | Lin et al. [2024] | Paquette et al. [2024] |
|---|---|---|---|---|
| Input dimension | $d$ | $D$ | $d$ | $v$ |
| Number of features | $p$ | $N$ | $M$ | $d$ |
| Number of samples | $n$ | $P$ | $N$ | $r$ |
| Capacity | $\alpha$ | $b$ | $a$ | $2\tilde{\alpha}$ |
| Source | $r$ | $(a-1)/2b$ | $(b-1)/2a$ | $(2\tilde{\alpha}+2\beta-1)/4\tilde{\alpha}$ |
| Target decay (in $L_2$) | $\alpha r + 1/2$ | $a/2$ | $b/2$ | $\tilde{\alpha} + \beta$ |

Table 2: Dictionary of notation between the source and capacity conditions defined in eq. (38) and the scalings in different neural scaling laws works. Note that since Paquette et al. [2024] also employs the greek letter "$\alpha$", we denote theirs by $\tilde{\alpha}$ to avoid confusion.

This setting is a particular case of the one introduced in Section 2. In particular, it satisfies particular source and capacity conditions eq. (38). To see this, note that the feature population covariance is identical to the input data covariance:

$$\mathbb{E}[\boldsymbol{W}\boldsymbol{x}\boldsymbol{x}^\top\boldsymbol{W}^\top] = 1/d\boldsymbol{\Lambda} \tag{215}$$

Therefore, we can identify $\boldsymbol{\Sigma} = 1/d\boldsymbol{\Lambda}$ which has $\text{Tr}\boldsymbol{\Sigma} < \infty$ for $\alpha > 1$ in the limit $d \to \infty$. Therefore, the features satisfy a capacity condition with scaling $\alpha$. Moreover, the asymptotic kernel is simply the linear kernel:

$$K(\boldsymbol{x}, \boldsymbol{x}') = \mathbb{E}_{\boldsymbol{w}}[\langle\boldsymbol{w}, \boldsymbol{x}\rangle\langle\boldsymbol{w}, \boldsymbol{x}'\rangle] = \frac{\langle\boldsymbol{x}, \boldsymbol{x}'\rangle}{d}. \tag{216}$$

Since the target variance is constant, this is equivalent to a source condition with:

$$r = \frac{1}{2}\left(1 - \frac{1}{\alpha}\right) \tag{217}$$

Since $\alpha \in (1, \infty)$, we are always in the hard regime $r \in (0, 1/2)$ where the target does not belong to the RKHS $\mathcal{H} = \mathbb{R}^d$. Indeed, since $\boldsymbol{\beta}_\star \sim \mathcal{N}(0, 1/dI_d)$, we have:

$$||f_\star||_{\mathcal{H}}^2 = \sum_{k=1}^{d} \beta_{\star,k}^2 \lambda_k \sim d^{\alpha-1} \tag{218}$$

which indeed diverges as $d \to \infty$. Moreover, note that least-squares regression correspond to the case $\ell = \infty$.

From the discussion above, the bias term scalings from Bahri et al. [2024] (resolution limited regime) and Maloney et al. [2022] (underparametrized regime $n \gg p$, i.e. $q \ll 1$, and overparametrized regime $n \ll p$, i.e. $q \gg 1$), correspond to a vertical cross-section on the large $\ell$ region of Fig. 2 (Right). Indeed, we recover exactly the rate of the *label term* in eqs. (167)-(168) of Maloney et al. [2022]:

$$\mathcal{B}(f_\star, \boldsymbol{X}, \boldsymbol{W}, \boldsymbol{\varepsilon}, \lambda) = O\left(n^{-2\alpha r q}\right) = O\left(p^{-(\alpha-1)}\right), \qquad n \gg p, \tag{219}$$

$$\mathcal{B}(f_\star, \boldsymbol{X}, \boldsymbol{W}, \boldsymbol{\varepsilon}, \lambda) = O\left(n^{-2\alpha r}\right) = O\left(n^{-(\alpha-1)}\right), \qquad n \ll p. \tag{220}$$

Similarly, it is possible to recover the rates for the *noise term* (first two results in eq. (86) of Maloney et al. [2022]) as the vertical cross-section on the large $\ell$ region of Figure 2 for the rates of the variance term. In particular:

$$\mathcal{V}(f_\star, \boldsymbol{X}, \boldsymbol{W}, \boldsymbol{\varepsilon}, \lambda) = \begin{cases} O\left(\sigma_\varepsilon^2 n^{-(1-q)}\right) = O\left(\sigma_\varepsilon^2 \frac{p}{n}\right), & n \gg p \\ O\left(\sigma_\varepsilon^2 n^0\right), & p \gg n \end{cases}. \tag{221}$$

**Comparison with the SGD rates from Paquette et al. [2024], Lin et al. [2024] —** Furthermore, in the linear noiseless target setting, lifting the condition in eq. (217), it is possible to compare our results to the compute-optimal rates for the risk obtained through stochastic gradient descent in Paquette et al. [2024]. In particular, we consider unitary batch-size and the correspondence between the number of iterations of stochastic gradient descent and the number of samples $n$ in ridge regression. Then, defining $\hat{\gamma}$ the decay rate of the compute-optimal curves for the risk $\hat{\mathcal{R}} \asymp n^{-\hat{\gamma}}$ in Paquette et al. [2024], corresponding to the compute-optimal number of features $\hat{p} \asymp \hat{p} =: n^{\hat{q}}$,[6]

---

[6]Note that this quantity is denoted $d_\star$ in Paquette et al. [2024].

coincides with $\gamma_\mathcal{B}(\ell = 1, \hat{q})$ in Theorem 4.1, *i.e.* with fixed regularization parameter $\lambda = 1$ ($\ell = 1$). In particular, consider the following regions in the phase diagram provided in their work:[7]

- Phase Ia ($r < 1/2$):

$$\hat{q} = \frac{1}{\alpha}, \tag{222}$$

$$\hat{\gamma} = 2r = 2\alpha\hat{q}r = \gamma_\mathcal{B}(1, \hat{q}); \tag{223}$$

- Phase II ($r > 1/2$ and $r < 1 - 1/2\alpha < 1$):

$$\hat{q} = \frac{1 + 2\alpha r - \alpha}{\alpha} < 1, \tag{224}$$

$$\hat{\gamma} = 2r = \frac{\alpha - 1}{\alpha} + \hat{q} = \gamma_\mathcal{B}(1, \hat{q}); \tag{225}$$

- Phase III ($r > 1/2$ and $r > 1 - 1/2\alpha$):

$$\hat{q} = 1, \tag{226}$$

$$\hat{\gamma} = \frac{2\alpha - 1}{\alpha} = \frac{\alpha - 1}{\alpha} + \hat{q} = \gamma_\mathcal{B}(1, \hat{q}). \tag{227}$$

We emphasize that, in Phases Ia and II, $\hat{\gamma} = \max_q \gamma_\mathcal{B}(1, q)$, while, in Phase III, $\hat{\gamma} \leq \max_q \gamma_\mathcal{B}(1, q)$. Hence, the compute-optimal decay rate of the risk for stochastic gradient descent is equal or smaller than the largest rate achievable by RFRR with fixed regularization $\lambda = 1$ and therefore always smaller than the optimal one in Corollary 4.2.

A similar setting has been investigated by the recent work Lin et al. [2024], providing scaling laws for the excess risk obtained by stochastic gradient descent with stepsize schedule $\eta_t = \eta/2^{t \log n/n}$, for $t = 1, ..., n$.[8] Under the same source and capacity conditions, assuming $r \in (0, 1/2)$ and $\eta = O(1)$, the result in their Theorem 4.2 may be rephrased as follows:

$$\mathbb{E}_{\boldsymbol{X},\boldsymbol{\varepsilon}} \mathcal{R}(f_\star, \boldsymbol{X}, \boldsymbol{W}, \boldsymbol{\varepsilon}, \eta) \asymp n^{-\gamma_{\mathrm{SGD}}(\eta, p)}, \tag{228}$$

$$\gamma_{\mathrm{SGD}}(\eta, p) = \left[ 2\alpha r \left( \frac{1 + \log_n \eta}{\alpha} \wedge \log_n p \right) \right] \wedge \left[ 1 - \left( \frac{1 + \log_n \eta}{\alpha} \wedge \log_n p \right) \right]. \tag{229}$$

Hence, choosing $p \asymp n^q$ and $\eta \asymp n^{\ell-1}$, *i.e.* $\eta \asymp \lambda^{-1}$, provided $\eta = O(1) \implies \ell < 1$, this result recovers precisely the same rates as in our Theorem 4.1.

---

[7]The Phases Ib, Ic and IV correspond to $\alpha < 1$, *i.e.* to an activation $\sigma \notin L_2$.

[8]The stepsize in Lin et al. [2024] is denoted by the greek letter $\gamma$, which we changed to $\eta$ to avoid confusion with the symbol we use for the risk decay rate.

