# OpenReview forum: "Dimension-free deterministic equivalents and scaling laws for random feature regression"
_NeurIPS.cc/2024/Conference — NeurIPS 2024 spotlight_

### Official Review · Reviewer_gFLF · 2024-07-02

**Soundness:** 4
**Presentation:** 3
**Contribution:** 2
**Rating:** 6
**Confidence:** 4

**Summary:**

This paper provides a non-asymptotic bound (Theorem 3.3) on the test error of random feature ridge regression (RFRR) using dimension-free (in the sense of the feature space) deterministic equivalents, which are the solutions to some self-consistent equations, under a concentration assumption (Assumption 3.1) on the features. As a result, they recover (Corollary 3.5) the deterministic equivalents result on the test error of kernel ridge regression (KRR) from previous literature, and they prove the exact decay rate of the neural scaling law (Corollary 4.1), filling the gap from previous literature.

**Strengths:**

This paper improves the results from previous literature in multiple ways: non-asymptotic over asymptotic bounds [Loureiro2022], RFRR over KRR [Misiakiewicz2024], more precise neural scaling law [Rudi2017, Cui2022]. The phase diagram (Figure 2) is novel, as far as I know, and it offers great insights for the test error in RFRR.

Reference:
- *Bruno Loureiro, Cédric Gerbelot, Hugo Cui, Sebastian Goldt, Florent Krzakala, Marc Mézard, and Lenka Zdeborová. Learning curves of generic features maps for realistic datasets with a teacher student model. Journal of Statistical Mechanics: Theory and Experiment, 2022(11):114001, nov 2022.*
- *Theodor Misiakiewicz and Basil Saeed. A non-asymptotic theory of kernel ridge regression: deterministic equivalents, test error, and gcv estimator, 2024.*
- *Alessandro Rudi and Lorenzo Rosasco. Generalization properties of learning with random features. In I. Guyon, U. Von Luxburg, S. Bengio, H. Wallach, R. Fergus, S. Vishwanathan, and R. Garnett, editors, Advances in Neural Information Processing Systems, volume 30. Curran Associates, Inc., 2017.*
- *Hugo Cui, Bruno Loureiro, Florent Krzakala, and Lenka Zdeborová. Generalization error rates in kernel regression: the crossover from the noiseless to noisy regime. Journal of Statistical Mechanics: Theory and Experiment, 2022(11):114004, nov 2022.*

**Weaknesses:**

My main concern, however, is the significance of the theoretical contribution of this paper. From the first sight, this paper is merely an extension of the paper [Misiakiewicz2024] with same techniques of deterministic equivalents.

Also, the result holds only under the strong concentration Assumption 3.1. As mentioned in the paper (line 119-121), I think that this paper could have been more significant if Assumption 3.1 had been relaxed in this paper.

**Questions:**

Regarding the above comments,

1. Could the authors explain more how different it is to implement the deterministic equivalents techniques in RFRR than in KRR?

2. How difficult it would be to relax Assumption 3.1 as in [Misiakiewicz2024]? And what datasets and random feature model could satisfy Assumption 3.1 empirically?

3. Despite not essential, but it could help the readers a lot if the authors could explain more on the quantities and their intuitions in the paper, like Eq (21-27). From my point of view, Theorem 3.3 is not the easiest to read.

Besides the questions, I found some potential typos in the appendix:

- A paragraph (line 536 - 540) seems misplaced: they should be directly after Eq (50).

- The equation below line 587 should be $\mathbf{f}\_j = ((\xi\_k\phi\_k(\mathbf{w}\_j)))\_{j\geq1}$ instead of $\mathbf{f}\_j = ((\xi\_k\psi\_k(\mathbf{w}\_j)))\_{j\geq1}$.

- a ``\rangle`` is missing in the equation under line 590: it should be $\sigma(\langle \mathbf{w}\_p,\mathbf{x}\_i \rangle)$ instead of  $\sigma(\langle \mathbf{w}\_p,\mathbf{x}\_i )$.

**Limitations:**

This paper is theoretical paper and the authors have addressed the assumptions and conditions explicitly in the paper.

---

> ### Author Rebuttal · Authors · 2024-08-06
>
> We thank the reviewer for their appreciation of our work and for the suggestions which will help to improve it. The typos in the appendix raised in the review will be corrected in the revised version of the manuscript.
>
> > *My main concern, however, is the significance of the theoretical contribution of this paper. From the first sight, this paper is merely an extension of the paper [Misiakiewicz2024] with same techniques of deterministic equivalents.*
>
> Indeed, our paper uses the deterministic equivalents for the functionals listed in Theorem A.2 from [Misiakiewicz, Saeed, 2024]. However, we politely disagree that this is ``merely an extension with same techniques''. It requires a significant amount of work to ensure that we can indeed achieve multiplicative bounds that applies to the regime with optimal rates under source and capacity conditions. As noted by reviewer Gqjy, the proof is 28 pages despite not repeating previous results.
>
> We further emphasize that it was not clear a priori that such a multiplicative bound---with optimal rates---for general $n$ and $p$ is achievable. Indeed, compared to the kernel case, the covariance over $z$ is a finite-rank random matrix $FF^T/p$, which does not concentrate (in particular, it has rank $p < n$ in the underparametrized case) and the model presents a double descent with a diverging peak at $n = p$ as $\lambda \to 0$.
>
> [Misiakiewicz, Saeed, 2024] only considers kernel ridge regression, while we consider random feature ridge regression which involves many more terms.  In particular, the following parts only appear in the RF case:
>
> - Controlling the covariance matrix $\hat \Sigma_F$ of the feature $z$ (Lemma B.2).
>
> - Showing concentration of the fixed points to the deterministic fixed point of Definition 1 (Proposition B.4, which requires a careful perturbation analysis and a uniform bound).
>
> - While the deterministic equivalent over $z$ corresponds to the same computation as in the kernel ridge regression case with covariance $\hat \Sigma_F$ (with an added projection term), the deterministic equivalents over $F$ involves many more terms that are not covered by Theorem A.2. These terms need to be decomposed carefully in order to show that we can indeed obtain multiplicative bounds with optimal rates and control the dependencies on $\Sigma,\lambda,\gamma_+$.
>
> > *Also, the result holds only under the strong concentration Assumption 3.1.*
>
> We refer the reviewer to the general rebuttal for a more detailed discussion.
>
> Besides Theorem 3.3, our paper offers the following contributions: 1) the statement of the general non-asymptotic deterministic equivalents which unifies existing asymptotic predictions, 2) the complete and precise decay rates under source and capacity conditions, which improves over [Rudi, Rosasco, 2017], 3) numerical simulations in various settings that show that Theorem 3.3 apply beyond its formal assumptions.
>
> As discussed above and in the response to reviewer b7dx, we consider Theorem 3.3 to be an interesting and challenging result to obtain even under Assumption 3.1. It is the first non-asymptotic (dimension-free) result with multiplicative approximation bounds for random feature regression. We acknowledge that Assumption 3.1 is restrictive, but such assumptions are common in the theoretical literature (e.g., it corresponds to the assumption in [Cheng, Montanari, 2022] for the infinite dimension case, or to [Bach, 2023] for linear asymptotics).
>
> We consider relaxing this assumption to be an important direction, which we leave to future research. For instance, we believe that the approach in [Misiakiewicz, Saeed, 2024] could be applied here (see response below). However, this would significantly increase the complexity of the proof---which is already 28 pages---while marginally contributing to the main message of the paper (Definition 1 and Corollary 4.1).
>
> - *How difficult it would be to relax Assumption 3.1 as in [Misiakiewicz2024]? And what datasets and random feature model could satisfy Assumption 3.1 empirically?*
>
> For the first question, [Misiakiewicz, Saeed, 2024] considers a relaxation of Assumption 3.1 by dividing the features into a low and a high-frequency components. The low-frequency features follow a Hanson-Wright type inequality similar to Assumption 3.1, while the high-frequency features are only assumed to be nearly orthogonal. Such a relaxation can be applied in the case of random feature ridge regression to both $(\psi_k)$ and $(\phi_k)$.  However, note that the random feature matrix corresponds to the product $FG^T$ and therefore it is not straightforward to separate these two parts. Although we believe an analysis differentiating between the underparametrized and overparametrized regimes will work, we believe that it is beyond what can be expected from a conference paper.
>
> See general rebuttal for a discussion of the second question.
>
> > *Despite not essential, but it could help the readers a lot if the authors could explain more on the quantities and their intuitions in the paper, like Eq (21-27). From my point of view, Theorem 3.3 is not the easiest to read.*
>
> We will add a longer discussion on these terms in the revised version of the paper.

---

> > ### Comment · Reviewer_gFLF · 2024-08-09
> >
> > Thank you for your detailed answer. I acknowledge the contribution of this paper that it unifies asymptotic prediction using deterministic equivalents despite restrictive assumptions. I will therefore keep my score unchanged and tend to accept this paper in the conference, under the condition that the authors include the above discussion in the revised version of the paper.

---

### Official Review · Reviewer_Gqjy · 2024-07-11

**Soundness:** 3
**Presentation:** 2
**Contribution:** 3
**Rating:** 5
**Confidence:** 3

**Summary:**

EDIT: updated my score after rebuttal.

The authors investigate the excess risk in random feature ridge regression. They give a deterministic expression that approximates the excess risk, with a controlled relative error. The dependence of the deterministic "equivalent" w.r.t. to key quantities in the problem is examined.

**Strengths:**

* Understanding the generalization ability of simple models is important, especially as our intuition of generalization has been challenged by recent results on overparametrized neural networks. Therefore, the paper is potentially impactful.
* The proposed result generalizes and refines a string of previous results in a single deterministic equivalent, and the dependence of this equivalent to problem parameters yields interesting conclusions, e.g. on the minimal number of features needed to achieve a statistically satisfying rate.
* There is clearly a lot of work that has gone in obtaining such a general result.

**Weaknesses:**

## General comments
My main concern is the format of the paper, which I think is more suited to a mathematical-minded ML/stats journal than a conference. The main paper is very dense but misses key details such as proofs or proof outlines, while the full proof of the main result is 28 pages long, which makes the full paper 50 pages long (i.e., appendices included). It is not reasonable to expect reviewers to go through such a long technical proof in the context (short time span and heavy workload) of NeurIPS reviews. As a journal submission, I would have the time required to go through the proofs and get to the heart of the proposed (interesting) result.

## Major
1. Eqn 4: don't you want $\epsilon$ to be zero-mean as well?
2. p3 L104; "we define wlog $\mathcal{V} = Im(T)$": if this is indeed a definition, I don't see why we need to add "wlog". If this is a statement about the image of $T$ being the whole of $\mathcal{V}$, we need to define $\mathcal{V}$ beforehand.
3. Eqn 13 is only true in $L^2(\mu_x)$ so I would avoid involving $\mathbf{x}$ in the statement of the equation, which suggests you mean a pointwise equality and requires a quantifier. If you actually meant a pointwise equality, then this should be explained, and further assumptions should likely be made on $\varphi$.
4. Assumption 3.1. I am unsure what is meant by an infinite matrix $A$. Especially when one needs to talk about the trace of $\Sigma A$, so that I assume we want to guarantee that $\Sigma A$ is a trace-class operator. Can you rephrase the assumption in terms of operators?
Similarly, the Frobenius norm for operators should be defined.
5. p4 L117: can you give more details on why cases 1) and 2) are covered by your assumption? Same for p4 L128: can you detail why these power decays satisfy Assumption 3.2?
6. Definition 1: For easier reading, I would define $\nu_1, \nu_2$ first, then $\Upsilon$, and then only $B$ and $R$. Otherwise, the reader has to wait until Eqn (24) to understand Eqn (18), which requires a lot of buffer memory from the reader's brain.
7. p5 L156 is there an implicit dependence on the feature map dimension? Can you explain where?
8. Figure 1: I would keep the caption short and descriptive, and move the definition of the data generating process to the main text. This would allow to explain more, for instance, what you mean by $v$ has a fixed overlap with the teacher vector".
9. The bibliography needs to be harmonized. There are many missing journal/conference names (if it's an arxiv preprint, say so and give the arxiv number), and a few initials mixed with full first names.
10. p9 A short discussion section summarizing the main points and limitations of the paper would be a good addition.

## Minor
* p1 L23: here and in the rest of the paper, you use the notation $\mu_w(\mathcal{W})$ to indicate that $\mu_w$ is a probability measure on $\mathcal{W}$. I would say that is not standard notation, and I would rather keep $\mu_w(\mathcal{W})$ for the measure of the whole space $\mathcal{W}$, i.e. 1.
* p1 L25: I think $\sigma$ has not been defined yet.
* p2 L46 "demystifying phenomena such as double descent and benign overfitting". I would give a reference for each concept.
* p2 L55 no need to boldface "our main contributions".
* Eqn 9: I would write $\mathrm{Var}$ instead of $\mathrm{Cov}$.
* p4 L127: settings
* p4 L135: what is a "self-consistent" equation?
* p5 L142 "we use $a_i = '*'$ to denote [...]": I don't understand the exact meaning of this statement.
* Eqn 30: I would remind the reader that $R_{n,p}$ has been defined in Definition 1.
* p5 L151 "in place" reads strangely. "In order", maybe?
* p5 L152 what does "fully" mean in "fully non-asymptotic"?
* p6 L171 what do you mean by "single-index"?
* p6 L183 span, not spam!
* p6 L183 infinite-dimensional
* p7 L216 understanding
* p8 L255 decays
* Figure 3 is too small to read.

**Questions:**

* Item 4 in my major comments.

**Limitations:**

This is fundamental work and does not have any immediate potentially harmful impact.

---

> ### Author Rebuttal · Authors · 2024-08-06
>
> We thank the reviewer for their appreciation of our work and for the suggestions which will help to improve it.  The issues (9, 10) and the typos raised in the minor comment section will be addressed and/or corrected in the revised version. Due to the space constraint in the rebuttals, will answer your questions in the "Minor" in a comment.
>
> Below, we address the "General" and "Major" comments and questions.
>
> > *My main concern is the format of the paper, which I think is more suited to a mathematical-minded ML/stats journal than a conference.*
>
> Random feature is a major research topic for the theory community at NeurIPS, with many papers about it every year. Our work is in direct dialogue with previous NeurIPS works, e.g. [Rahimi and Recht 2007, 2008; Rudi and Rosasco 2017; Cui et al. 2021; Xiao et al., 2022], and therefore our choice of venue simply reflects the interest we believe our results can rouse in this community.
>
> Moreover, the main message of this work is simple, and we believe is well conveyed in 9 pages: we derive a deterministic, non-asymptotic characterization of the test error which generalizes previous results and can be used to derive new insights, such as the different power-law scalings of interest to both the kernel and neural scaling law communities. Indeed, due to the generality of the results the proof involve long and technical arguments with are left in the Appendix, but we politely disagree that this stands out from the NeurIPS practice, see e.g. proofs in the Appendices of [Rudi and Rosasco 2017; Loureiro et al., 2021; Xiao et al., 2022].
>
> > *Eqn 4: don't you want $\epsilon$ to be zero-mean as well?*
>
> Indeed, the noise is intended to be zero-mean. We will explicitly state this assumption in the revised version of the paper
>
> > *p3 L104; "we define wlog $\mathcal{V} = {\rm Im}(\mathbb{T})$ ": if this is indeed a definition, I don't see why we need to add "wlog". If this is a statement about the image of $\mathbb{T}$ being the whole of $\mathcal{V}$, we need to define $\mathcal{V}$ beforehand.*
>
> Indeed, we will remove ``wlog'' here, as we simply define $mathcal{V} = {\rm Im}(\mathbb{T})$.
>
> > *Eqn 13 is only true in $L^2(\mu_x)$*
>
> Following the reviewer's suggestion, we will avoid any ambiguity by rewriting $f_\star = \sum_{k\geq 1}\beta_{\star,k}\psi_k$ in the revised version.
>
> > *Assumption 3.1. I am unsure what is meant by an infinite matrix $A$.
>
> The matrix notation $A\in\mathbb{R}^{\infty\times\infty}$ is here used to represent a linear operator $A$ acting on an infinite-dimensional Hilbert space $\mathcal{H}$. In particular, given a basis ($\psi_k$) of $\mathcal{H}$, we define the $kk'-$th element of $A$ as $\langle \psi_k, A \psi_{k'}\rangle_{\mathcal{H}}$. The expression $Tr(\Sigma A)<\infty$ ensures that $\Sigma A$ is trace-class, as correctly stated by the reviewer. More generally, the infinite dimensional matrices that appears in the paper (e.g., $\Sigma$, $F^T F$, or $ff^T$) are understood to be linear operators $\mathcal{H} \to \mathcal{H}$.
>
> Following the reviewer's suggestion, we will rephrase Assumption 3.1 in order to improve its clarity and add a remark on our notations in terms of linear operator for completeness.
>
> > "*p4 L117: can you give more details on why cases 1) and 2) are covered by your assumption? Same for p4 L128: can you detail why these power decays satisfy Assumption 3.2?*"
>
> Assumption 3.1 is a slight relaxation of Hanson-Wright inequality, which is satisfied by 1) and 2), see [Adamczak, 2014] (note that the inequality is stated in finite dimension, however it also holds for $d = \infty$, as noted in Remark 2.1 in [Cheng, Montanari, 2022]).
>
> Concerning the second part of the question, we consider the power decays $\xi_k^2 \asymp k^{-\alpha}$ and $\beta_{\star,k}\asymp k^{-\beta}$. Since we are considering square-integrable target function and feature map, we have that $2\beta, \alpha > 1$.
>
> We notice that:
> $$
>     \sum_{k={\rm m}+1}^\infty \xi^2_k >  \int_{\rm m+1}^\infty x^{-\alpha} = \frac{({\rm m}+1)^{1-\alpha}}{\alpha - 1}
> $$
>     and the inequality in (16) holds if
> $$
>     p^2({\rm m}+1)^{-\alpha}\leq \frac{p\lambda}{n}\cdot\frac{({\rm m+1})^{1-\alpha}}{\alpha - 1} \implies {\rm m} \geq \frac{pn}{\lambda}(\alpha - 1) - 1.
> $$
>     The inequalities in (17) may be written as
> $$
>         \frac{\sum_{k\geq 1} k^{-\alpha}(k^{-\alpha} + \nu_2)^{-1}}{\sum_{k\geq 1} k^{-2\alpha}(k^{-\alpha} + \nu_2)^{-2}}\leq C_*, \qquad \frac{\sum_{k\geq 1} k^{-2\beta}(k^{-\alpha} + \nu_2)^{-1}}{\sum_{k\geq 1} k^{-2\beta}(k^{-\alpha} + \nu_2)^{-2}}\leq C_*.
> $$
>     Therefore, by applying the integral test for convergence, it is easy to show that all the series involved in the inequalities are bounded by a positive constant term dependent on $\alpha, \beta, \nu_2$ (where the latter depends itself on $\alpha, \beta$ and $n, p, \lambda$). For instance
> $$\sum_{k\geq 1} \frac{k^{-\alpha}}{(k^{-\alpha} + \nu_2)^{-1}} < \int_{0}^\infty \frac{x^{-\alpha}}{(x^{-\alpha} + \nu_2)^{-1}}{\rm d}x = \nu_2^{-1/\alpha}\frac{\pi}{\alpha}\csc\left(\frac{\pi}{\alpha}\right) =: C_{\alpha,\nu_2}.$$
>
> [Adamczak, 2014] *A note on the Hanson-Wright inequality for random vectors with dependencies.*
>
> > *Definition 1: For easier reading, I would define $\nu_1,\nu_2$ first, then $\Upsilon$, and then only $B$ and $R$.*
>
> We thank the reviewer for raising this issue, which will be addressed in the revised version of the paper.
>
> > *p5 L156 is there an implicit dependence on the feature map dimension?*
>
> Indeed, this remark is confusing and we will rephrase it. There is no dependency on the feature map dimension. The covariance $\Sigma$ might depend indirectly on the feature map dimension (however the key quantity is the intrinsic dimension $r_\Sigma$ and our results hold for infinite dimensional feature maps).

---

> ### Comment · Reviewer_Gqjy · 2024-08-10
>
> Thanks for the clarifications! I am still in two minds: one the one hand, I agree that the 9-pager already conveys an interesting technical contribution, with clarity, provided the authors implement the minor changes recommended by the reviewers. On the other hand, I would have preferred having had the time to proofread the proof carefully, in a journal submission like JMLR. But, after having read the other reviews, and anticipating over the reviewer discussion period, I am willing to increase my score and not argue for rejection.

---

### Official Review · Reviewer_tRLP · 2024-07-25

**Soundness:** 4
**Presentation:** 4
**Contribution:** 4
**Rating:** 7
**Confidence:** 4

**Summary:**

Prior work on random feature (ridge) regression study the test error in the high-dimensional asymptotic. However, ideally one would hope for a non-asymptotic deterministic characterization of the test error.   In this paper, the authors tackle this problem and show that under a concentration assumption,  the test error is well approximated by a closed-form expression that only depends on the feature map eigenvalues. They use this result to study various problem in random features regression.

**Strengths:**

- This paper rigorously solves an important problem in the analysis of random features regression. I think Theorem 3.3 will be of independent interest as well.
- The main result of the paper does not require the random regression coefficient assumption and hold for deterministic beta_star.
- Theorem 3.3 has a very clean form. The bounds in Theorem 3.3 are multiplicative. Thus, they scale correctly with the risk. Taking particular limits (e.g., p \to \infty or \lambda \to zero), we easily recover already known phenomenon.
-  The authors derive sharp excess error rates under power-law assumptions and provide a tight result on the smallest number of features needed to achieve optimal minimax rate.
- The proofs seem correct and rigorous.

All in all, I really enjoyed reading this paper and I recommend acceptance.

**Weaknesses:**

I suggest the authors expand the discussion around Assumption 3.1 and provide more detailed examples for which this assumption holds.

**Questions:**

The paper is very well written and I have no particular question.

**Limitations:**

The authors have adequately addressed the limitations.

---

> ### Author Rebuttal · Authors · 2024-08-06
>
> We thank the reviewer for their appreciation of our work and for the suggestion which will help improving it.
>
>
> > "*I suggest the authors expand the discussion around Assumption 3.1 and provide more detailed examples for which this assumption holds.*"
>
> We will expand the discussion around Assumption 3.1 (see the general rebuttal for a discussion).

---

### Official Review · Reviewer_b7dx · 2024-07-28

**Soundness:** 4
**Presentation:** 3
**Contribution:** 3
**Rating:** 6
**Confidence:** 4

**Summary:**

The paper studied the non-asymptotic generalization error for random feature ridge regression (RFRR) models. By considering the eigendecomposition of random features with respect to data distribution and weight distribution, the authors proved a feature-dimension-free deterministic equivalence for the generalization error of RFRR, where the sample size and number of features bound the approximation error. From this deterministic equivalence, fixing the number of data points, this paper presented the minimal number of features for the optimal decay rate of the excess risk of RFRR when considering power law assumptions for data covariance and target. This analysis provides a clear picture of the generalization error scaling law for source and capacity conditions of RFRR.

**Strengths:**

1. The paper is well-motivated, and the mathematics appears correct to me. The writing is clear, and the authors do a good job of presenting results with many discussions and comparisons of related works.

2. The authors provide various empirical simulations to justify the theorems, including random synthetic data, real-world data, random weights, and weights trained by gradient descents. This offers more insights into theoretic results and the generality of the results in this paper.

**Weaknesses:**

1. One concern is how we can check Assumptions 3.1 and 3.2 for specific nonlinear feature models with some concrete data and weight distributions. The results of this paper rely on the eigen decomposition in (11) but can we get some specific examples of eigenvectors $\psi_k$ and $\phi_k$? For instance, in a classical high-dim statistics setting, if we consider a nonlinear RFRR model with i.i.d. sub-Gaussian dataset and weight matrix and a certain nonlinear teacher model, can we justify Assumptions 3.1 and 3.2 in this case?

2. There should be more clarification for the notations and conditions of the main results, Theorem 3.3. For instance, what would be the meaning of (25-27) and (28-29)? Are these bounds and approximation rates necessary or due to some technical reasons? If we apply the bounds in (28-29) to (32) for the error bound of the deterministic equivalence, the bound of $\mathcal{E}(n,p)$ seems to be loose and cannot get $\tilde O(n^{-1/2}+p^{-1/2})$.

**Questions:**

1.  In the simulation, the authors provide an empirical diagonalization method for estimating $\Sigma$ and $\beta$ by considering $N=P\gg n,p$ and computing the eigendecomposition of the empirical feature covariance. Is there any theoretical guarantee for this approximation? This approximation seems to indicate that we can still apply the asymptotic results for empirical feature covariance under the proportional or polynomial scaling regime for $n, p$, e.g. [Mei and Montanari, 2022, Gerace et al., 2021, Dhifallah and Lu, 2020, Hu and Lu, 2023, Hu et al., 2024], to analyze the deterministic equivalence and the generalization errors.

2. In (10), do you need to assume the subsets $\mathcal{X}$ and $\mathcal{W}$ in $\mathbb{R}^d$ are compactly supported?

3. Typo (7): $h_*\to f_*$

4. What is $\nu_2$ in (17) in Assumption 3.2? You did not introduce this notation until Definition 1.

5. You did not introduce intrinsic dimension around (25).

6. In line 147, why do you assume assumption 3.2 again?

7. Line 183, typo: spam; Line 230: (37) $\to$ (38)?

8. In Definition 3, typo in the definition $r_\Gamma(k)$: $p \to q$. Line 546, $\Sigma\to \Gamma$.

9. In Theorem A.2, what is the typical order of $\rho_\lambda (n)$? From (53), we cannot claim the error term in (54) will be vanishing.

10. How do you prove the last equation on page 16, for the upper bound the $||S_i||_{op}$? Can you explain it? And in the same proof, how do you apply the matrix Bernstein inequality for infinite dimension $\tilde S$?

11. Below Line 587, typo in $f_j$: $j\ge 1\to k\ge 1$; typo in (60): $A\to B$.

12. In Line 634, you consider $\eta_*\in (0,1/4)$ but in the main results, you have $\eta_*\in (0,1/2)$. Can you clarify it?

13. How do you prove $||\hat \Sigma_F||_{op}\ge 1/2$?

14. Below Line 800 and in Line 1033, you use $G$ to denote the resolvent which has been used as the data feature matrix as well.

---

> ### Author Rebuttal · Authors · 2024-08-06
>
> We thank the reviewer for their appreciation of our work and for the suggestions which will help improving it. The typos raised (2, 5, 6, 7, 8, 11, 14) will be corrected in the revised version of the manuscript.
>
> Below, we address the other specific comments and questions.
>
> > "*One concern is how we can check Assumptions 3.1 and 3.2*"
>
> Please see general rebuttal.
>
> > "*[... ] the bound of $\mathcal{E}(n,p)$ seems to be loose and cannot get $\tilde{O}(n^{-1/2}+p^{-1/2})$*"
>
> The bounds in (28-29) are technical conditions used in intermediate steps of the proof to ensure the multiplicative bounds. If conditions (28-29) are verified, then the approximation guarantee (30) is satisfied with $\mathcal{E} (n,p)$ rate given in (32). Conditions (28-29) are not meant to ensure that $\mathcal{E}(n,p)$ is small, but that the bound (30) holds (and indeed, $\mathcal{E}(n,p)$ might not be vanishing).
>
> The conditions (28-29) and rate $\mathcal{E}(n,p)$ depend explicitly on $n,p,\lambda,\Sigma$ and should be applied to specific settings. We will add the details of how to compute these bounds in the case of source and capacity conditions in Corollary 4.1., to illustrate the use of this theorem (and provide a full proof of Remark 4.1).
>
> Some dependency in $\Sigma$ and $\lambda$ in the multiplicative bounds are unavoidable, and therefore are not technical. However, in most relevant examples (e.g. regularly varying spectrum), $\rho_\kappa (p)$ and $\tilde \rho_\kappa (n,p)$ will be of order $\log (\max (n,p))^C/\kappa$, and $\mathcal{E} (n,p)$ will indeed be of order $p^{-1/2} + n^{-1/2}$ up to log factors. We further believe that the dependency on $\lambda$ could be improved, at the cost of a worst dependency $p^{-c} + n^{-c}$, $c <1/2$, and further assumptions (see for example [Cheng, Montanari, 2022]). We will add this discussion to the revised version.
>
> > "*Is there any theoretical guarantee for this approximation?*"
>
> First, we would like to clarify that Figures 1 and 2, 5, 6 are intended as an illustration of the scope of the theory to settings that might go beyond the technical assumptions. The ``empirical diagonalization'' procedure is therefore only a tool to obtain the spectrum in cases which it is not available analytically (e.g. real data), and has been inspired from other works in the literature, e.g. [Bordelon et al., 2020; Loureiro et al., 2020; Simon et al., 2023a].
>
> Indeed, when estimating the spectrum empirically  the quality of the theoretical predictions will depend on the choice $N,P$. The convergence rates for the empirical estimation of the spectrum were studied in [Koltchinskii and Giné, 2000; Braun, 2006], and in the worst-case are given by the usual CLT rates. In practice (e.g. Figs. 1 and 2), we observe that $N,P = 10^{4}$ was enough for a very good agreement with finite size simulations in the range of $n,p$ considered.
>
> However, it is important to stress that this is unrelated to the proportional asymptotics (after estimating $\Sigma,\beta_*$, we use the deterministic equivalents in Definition 1). It just implies that to go to larger $n,p$ (e.g. a polynomial scaling) ones requires a finer estimation of the spectrum.
>
> -[Koltchinskii and Giné, 2000] *Random matrix approximation of spectra of integral operators*.
>
> -[Braun, 2006] *Accurate error bounds for the eigenvalues of the kernel matrix*.
>
> > "*In (10), do you need to assume the subsets $\mathcal{X}$ and $\mathcal{W}$ in $\mathbb{R}^{d}$ are compactly supported?*"
>
> We introduce an abstract random feature model $\varphi : \mathcal{X} \times \mathcal{W} \to \mathbb{R}$, but we only require that it is diagonalizable. It is sufficient for $\mathcal{X}$ and $\mathcal{W}$ to be Polish probability spaces and $\varphi$ to be square integrable, so that $\varphi$ is diagonalizable (by the spectral theorem of compact operators). For Theorem 3.3, we only use Assumption 3.1 that is an assumption directly on the eigenfunctions of the activation.
>
> > *What is $\nu_{2}$ in (17) in Assumption 3.2? You did not introduce this notation until Definition 1.*
>
> Indeed, the parameter $\nu_2$ is the solution of (23) in Definition 1. Following the reviewer's remark, we will move Assumption 3.2 after Definition 1 in the revised version.
>
> > "*In line 147, why do you assume assumption 3.2 again?*"
>
> This is a typo. Thank you for pointing it out!
>
> > "*In Theorem A.2, what is the typical order of $\rho_{\lambda}(n)$? From (53), we cannot claim the error term in (54) will be vanishing.*"
>
> For most cases of interest, $\rho_{\lambda}(n)$ will be of order $\log (n)^C/\lambda$ (see response above). We further insist that these bounds are non-asymptotic (they depend explicitly on finite $n$) and do not need to be vanishing: if they are equal to $1/2$, it is enough to pinpoint the scale of the functionals.
>
> > "*How do you prove the last equation on page 16? How do you apply the matrix Bernstein inequality for infinite dimension $\tilde{S}$?*"
>
> $||S_i|| = ||x_{+,i}||^{2}=  x_{+,i}^T I x_{+,i}$. We can therefore apply Assumption A.1 with $A= I$ and do an union bound on $i \in [n]$, which results in the $\log(n)$ factor.
>
> $S_i$ can be seen as a trace class operator, and many tools from matrix theory extend to this setting. The matrix Bernstein inequality only depends on the operator norm and the intrinsic dimension of the matrix, and not on the size of the matrix,  and  its proof extends to infinite dimensional operators. See for example [Rudi, Rosasco, 2017] and references therein.
>
> > "*In Line 634, you consider $\eta\in(0,1/4)$ but in the main results, you have $\eta\in(0,1/2)$?*"
>
> This is a typo. Thank you for pointing it out!
>
> > "*How do you prove $||\hat{\Sigma}||_{op}\geq 1/2$*"
>
> In the proof in section B.7.1, we have $|| \hat{\Sigma} || \geq || F_1 F_1^T /p || $ and by equation (124), we proved that the top eigenvalue of $F_1 F_1^T /p$ (i.e., $k_1 = 1$) is lower bounded by $\hat \xi_1^2/2$, where $\hat\xi_1^2 = || \Sigma || = 1$ by assumption.

---

> > ### Comment · Reviewer_b7dx · 2024-08-13
> > **Official Comment by Reviewer b7dx**
> >
> > Many thanks for the authors' response and explanations. The rebuttal has resolved most of my questions. I tend to accept this paper and expect the authors to include more discussions in the revision, especially for Assumption 3.1 and Remark 4.1.

---

### Author Rebuttal · Authors · 2024-08-06

**Assumption 3.1**:  We acknowledge Assumption 3.1 can be restrictive, as mentioned in the paragraph 116-122. In fact, it will not be satisfied by some standard examples of random feature models, such as $\varphi (x,w) = \sigma ( \langle x, w\rangle)$ with non-linear activation $\sigma :\mathbb{R} \to \mathbb{R}$ and $x,w $ Gaussian vectors.

This is a general challenge when studying deterministic equivalents for non-linear random feature and kernel models. The eigenfunctions are not independent and contain functions of arbitrary high frequency (heavy tailed), which are far from the standard setting of RMT (see discussions in [Misiakiewicz, Saeed, 2024]). Hence, most existing results for kernel methods have either been restricted to linear asymptotics (thanks to the linearization trick [El Karoui, 2010], [Hu, Lu,2020], [Bartlett, Montanari, Rakhlin, 2021]), or polynomial asymptotics for restricted settings (namely inner-product kernels or activations on the sphere [Xiao, Hu, Misiakiewicz, Lu, Pennington, 2022], [Hu, Lu, Misiakiewicz, 2024]). The notable exception is [Misiakiewicz, Saeed, 2024], which provides abstract assumptions (satisfied by inner-product kernels on the sphere) under which non-asymptotic multiplicative bounds can be proven (see response to reviewer gFLF).

We consider the primary motivation of our work to be the rigorous derivation of the excess risk rates in Corollary 4.1, which requires infinite dimensional features, general target functions (not restricted to the RKHS), and multiplicative bounds.

In order to make progress on this question, we made the following choice: 1) introduce an abstract random feature model $\varphi (x, w)$; 2) present the general non-asymptotic deterministic equivalents; and 3) show a sufficient assumption (Assumption 3.1) where we can show tight multiplicative approximation bounds.

We believe our choice is justified by the following:

- These deterministic equivalents recover all the known asymptotic results for random feature models as special limits. We believe there is value in stating these general formulas along with a proof, albeit under a restricted assumption.

- This is further motivated by our numerical simulations which show these theoretical predictions are remarkably accurate across various synthetic and real datasets. Note that the simulations include the case of $\sigma (\langle x, w\rangle)$ with $w,x$ Gaussian vectors. This indicates that the validity of Theorem 3.3 extends way beyond Assumption 3.1.

- These deterministic equivalents allow to derive tight rates (both lower and upper bounds) under source and capacity conditions, which display a phase diagram richer than previously known (Figure 2). In particular, our derived optimal parametrization improves on [Rudi and Rosasco, 2017].

- Assumption 3.1 is satisfied by some toy models that are popular (and often necessary) in theoretical investigations. For example, if $x$ and $w$ are (possibly infinite dimensional) vectors with independent sub-Gaussian entries (by Hanson-Wright inequality, which extends to the infinite dimensional case) and $\varphi (x,w) = x^T w$ (e.g., infinite dimensional linear regression [Cheng, Montanari, 2022]). In this model, our results vastly extends the asymptotic results in the linear scaling $n \asymp p \asymp d$ of [Hastie, Montanari, Rosset, Tibshirani, 2022] and [Bach, 2023], to $d = \infty$, general $n,p$, and multiplicative approximation bounds. Another example (with non-independent entries) is $\varphi (x,w) = f(x)^T g(w)$ with $x,w$ Gaussian random vectors and $f,g$ Lipschitz functions (by Lipschitz concentration of Gaussian vectors).






Note that already obtaining Theorem 3.3 under current assumptions is an interesting (and challenging) result: the features are infinite dimensional, the covariance does not have bounded conditioning number, and there is no reason to expect that the deterministic equivalents will remain accurate under the scalings considered here (source and capacity conditions, vanishing regularization, no restriction on the scaling between $p$ and $n$). We further expect the obtained multiplicative approximation rates $\tilde O (p^{-1/2} + n^{-1/2})$ to be optimal, based on the local law fluctuations of the empirical feature matrix.

We believe that showing deterministic equivalents under more realistic assumptions is an important and challenging direction, which we leave to future research.

We will add further discussions on Assumption 3.1 in the main text, with added references and the examples described above.

- [El Karoui, 2010] *The spectrum of kernel random matrices*.

- [Hu, Lu,2020] *Universality Laws for High-Dimensional Learning with Random Features*.

- [Bartlett, Montanari, Rakhlin, 2021] *Deep learning: a statistical viewpoint.*

- [Hastie, Montanari, Rosset, Tibshirani, 2022] *Surprises in high-dimensional ridgeless least squares interpolation.*

- [Bach, 2023] *High-dimensional analysis of double descent for linear regression with random projections.*

---

### Decision · Program_Chairs · 2024-09-25

**Decision:**

Accept (spotlight)

**Comment:**

The paper investigates the generalization performance of random feature-based ridge regression by providing a deterministic equivalent for the test error. The key contribution is that this approximation result is non-asymptotic, multiplicative, and independent of the feature map dimension allowing for infinite-dimensional features. As an application, this result is employed in random feature-based kernel ridge regression to provide a tight bound on the smallest number of random features needed to achieve the optimal minimax error rate, thereby improving the result in the literature. The paper is well-written and the contribution is solid. All the reviewers are positive about the paper's novelty and contributions, which I agree with. Based on the discussion with the reviewers, I think the paper is a nice contribution to the literature and therefore I am recommending acceptance. I strongly urge the authors to incorporate all the reviewer comments along with any clarifications they provided in the rebuttal into the camera-ready version.